# NON-COLLABORATIVE USER SIMULATORS FOR TOOL AGENTS

**Jeonghoon Shim, Woojung Song, Cheyon Jin, Seungwon Kook, Yohan Jo**\*
Graduate School of Data Science, Seoul National University
{jhshim98,yohan.jo}@snu.ac.kr

## ABSTRACT

Tool agents interact with users through multi-turn dialogues to accomplish various tasks. Recent studies have adopted user simulation methods to develop these agents in multi-turn settings. However, existing user simulators tend to be agent-friendly, exhibiting only cooperative behaviors, failing to train and test agents against non-collaborative users in the real world. We propose a novel user simulator architecture that simulates four categories of non-collaborative behaviors: requesting unavailable services, digressing into tangential conversations, expressing impatience, and providing incomplete utterances. Our user simulator can simulate challenging and natural non-collaborative behaviors while reliably delivering all intents and information necessary to accomplish the task. Our experiments on MultiWOZ and $\tau$-bench reveal significant performance degradation in state-of-the-art tool agents when encountering non-collaborative users, as well as agent weaknesses under each non-collaborative condition such as escalated hallucinations and dialogue breakdowns. Our findings point to the need for methods that can improve agent robustness to the wide range of user behaviors encountered in deployment. We release the extensible simulation framework to help the community develop and stress-test tool agents under realistic conditions within their own service domains. Our code is available at `https://github.com/holi-lab/NCUser`.

## 1 INTRODUCTION

Tool agents engage in multi-turn dialogues where they interpret user requests, execute appropriate API calls, and communicate results back to users to complete specific tasks. Recent studies actively adopt user simulators to conduct multi-turn dialogue simulations between users and tool agents (Yao et al., 2024; Barres et al., 2025; Prabhakar et al., 2025). Unlike static dialogue datasets, these user simulators enable dynamic multi-turn interactions where the conversation flow adapts based on the agent's responses and actions. This allows researchers to train and evaluate tool agents across diverse scenarios, capturing the interactive nature of real-world tool use. However, most existing user simulators and training datasets are agent-friendly, exhibiting only cooperative behaviors that fail to capture the complexity of real-world interactions, such as dealing with impatient users (Reynolds & Harris, 2009) or handling requests that are beyond the agent's capabilities (Bitner et al., 1990). This hinders both developing robust agents and assessing their resilience to non-collaborative user behaviors in the real world.

To address this limitation, we develop a novel user simulation framework that incorporates diverse non-collaborative behaviors observed in real-world interactions (Figure 1). We achieve this through two steps. First, we identify four types of non-collaborative user behaviors informed by marketing research, open-domain dialogue studies, and real-world user-agent interaction data: (1) Unavailable Services: users request functionalities beyond the agent's API capabilities; (2) Tangential: users engage in free conversation unrelated to their primary task; (3) Impatience: users express frustration through emotional escalation when experiencing delays or service failures; and (4) Incomplete Utterances: users provide poorly articulated messages (§3.1). Second, we build a user simulator

---

\* Corresponding author.

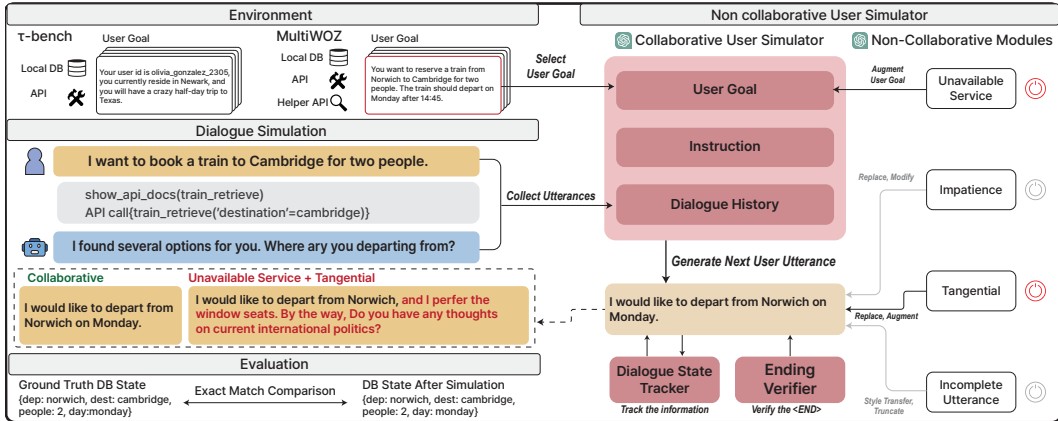

Figure 1: Overall structure of non-collaborative user simulation environment. This includes the tool agent environment, collaborative user simulator, and non-collaborative user simulation modules.

architecture that models these behaviors while ensuring goal-aligned simulation, i.e., reliably delivering all intents and information necessary to accomplish the given task (§3.2).

In our main experiments, we leverage MultiWOZ (Budzianowski et al., 2018) and $\tau$-bench (Yao et al., 2024), to create stateful task-oriented dialogue environments and conduct comprehensive experiments to reveal the vulnerabilities of agents. We demonstrate that non-collaborative user behaviors lead to significant performance degradation for state-of-the-art LLMs. Our detailed analysis reveals how each behavior category impairs LLMs: unavailable service and incomplete utterance modes lead to immature tool utilization of the agents, while tangential and impatience modes expose shortcomings in their dialogue management. Moreover, when small LLMs are trained exclusively on typical (collaborative) scenarios, as many practitioners would do for the deployment of their services, the performance improvements on non-collaborative behaviors significantly lag behind those on collaborative behaviors. Lastly, we extend our non-collaborative user simulators to ColBench (Zhou et al., 2025) (which does not involve tool use) and MINT (Wang et al., 2024) (which involves user-agent collaboration) and observe disparate performance patterns across benchmarks. These results demonstrate that our extensible framework can preemptively diagnose potential agent weaknesses and yield insights across diverse domains. We hope that researchers working on LLM agents adapt our simulator to their own tasks for training and testing their agents against the complexities and challenges of real-world deployments.

The main contributions of our work are summarized as follows: (1) We define four types of non-collaborative user behaviors and develop a user simulator framework that exhibits these behaviors while maintaining goal alignment. It outperforms simpler prompt-based approaches in creating challenging yet realistic dialogue scenarios. (2) We reveal that state-of-the-art tool agents exhibit significant performance degradation under non-collaborative conditions, providing insights into behavior-specific failure mechanisms that can guide the development of more robust agents. (3) We implement our user simulator on various settings, demonstrating the extensibility of our architecture and enabling researchers to evaluate and improve the agent resilience in real-world scenarios.

## 2 RELATED WORKS

**Multi-Turn Dialogues of Tool Agents**   It is important for tool agents to solve user requests in realistic scenarios. To evaluate these capabilities, several works have developed benchmarks (Qin et al., 2023; Patil et al., 2023; Tang et al., 2023) and simulation environments (Trivedi et al., 2024; Patil et al., 2025) to reflect realistic use cases. Despite their contributions, most of these works are limited to single-turn interactions, overlooking the fact that real-world task completion requires multi-turn conversations between users and agents. To address this gap, recent works such as Li et al. (2023) and Farn & Shin (2023) have generated multi-turn dialogues between users and tool agents, incorporating scenarios where agents engage in multi-turn conversations to elicit information from users (e.g., user intent, input parameter values). More complex scenarios include dialogues from

Shim et al. (2025) where users fail to provide required parameters, and simulations from Laban et al. (2025) that model underspecification behavior. However, these works have still represented only interactions with collaborative users, with limited exploration of complicated situations arising from non-collaborative users. In our work, we develop a user simulator to reproduce these non-collaborative behaviors, thereby covering a broader spectrum of user-agent interactions.

**User Simulation for Tool Agents**  Adopting user simulators in dialogue simulation has been a well-established approach for developing task-oriented dialogue systems (Eckert et al., 1997; Sekulic et al., 2024; Schatztnann et al., 2005; Schatzmann et al., 2007), and has recently been adopted in tool agent research. The $\tau$-Bench series (Yao et al., 2024; Barres et al., 2025) and Apigen-mt (Prabhakar et al., 2025) propose prompt-based user simulators that incorporate the user goal, dialogue history, and instruction within the prompt to simulate goal-oriented conversations. To ensure goal-aligned behavior of user simulators, recent works have explored various techniques: Luo et al. (2024) employs LLM-based verifiers to inspect user utterances during dialogue simulation, while Mehri et al. (2025) trains user simulators through SFT and subsequent GRPO training with LLM-as-a-judge reward. However, existing user simulation research in tool agents has exclusively focused on collaborative user behavior, with no attempts to address or simulate the non-collaborative behaviors exhibited by real-world users. Furthermore, frameworks for building and evaluating goal-aligned user simulators remain heavily reliant on LLM-as-a-judge. To mitigate this, we investigate non-collaborative user behaviors through empirical studies and propose a novel user simulation method to reproduce these behaviors. We also enforce goal alignment, i.e., all goal-relevant information is delivered during the dialogue, and build a user simulator that satisfies it.

## 3 NON-COLLABORATIVE USER SIMULATION ENVIRONMENT

Figure 1 illustrates our proposed non-collaborative user simulation environment. We first define four categories of non-collaborative user behavior grounded in research from marketing, open-domain dialogue, and real-world user-agent dialogue data (§3.1). Second, we develop user simulators that communicate their needs while exhibiting non-collaborative behaviors, creating a realistic and challenging dialogue simulation environment (§3.2). We develop our non-collaborative user simulator in the MultiWOZ environment and subsequently extended it to $\tau$-bench with minimal effort.

### 3.1 NON-COLLABORATIVE USER BEHAVIORS

We define our non-collaborative user behaviors grounded in marketing research and real-world user-LLM conversation studies. We first examined various types of user behaviors from both fields and clustered those that share similar features and then identified four representative categories: (1) Unavailable Services, (2) Tangential, (3) Impatience, and (4) Incomplete Utterances. We provide a detailed taxonomy and the underlying rationale for our behavior definitions in §A.

**Unavailable Service**  This behavior stems from the user's uncertainty about the agent's boundaries, prompting them to request services beyond its capabilities (Bitner et al., 1990; Reynolds & Harris, 2005). We define this as requests that cannot be fulfilled using the available APIs—either because relevant APIs do not exist, or existing APIs lack the necessary parameters to accommodate the user's specific requirements. For example, when a user requests "Book me a window seat on the train", but the train booking API does not support seat selection options, the agent cannot fulfill the specific seat preference—making this an unavailable service request.

**Tangential**  Certain customers expect social rapport (Price et al., 1995; Beatty et al., 1996) or demand continuous attention (Jeung et al., 2018), and talkative users are known to be particularly challenging as they may become upset if not given space to express themselves (Geraghty, 2023; Jones, 2023; Coppell, 2023). We define tangential behavior as user utterances unrelated to goal completion—typically personal interests or background topics expressed through four dialogue acts adopted from open-domain dialogue research (Yu & Yu, 2021): (1) Factual Question, (2) Opinion Question, (3) General Opinion, and (4) Non-opinion Statement (See details in §C.2). We also model these users as raising complaints when agents ignore their tangential utterances. For example, a user booking a train might also ask, "By the way, where do you think I should visit first traveling in

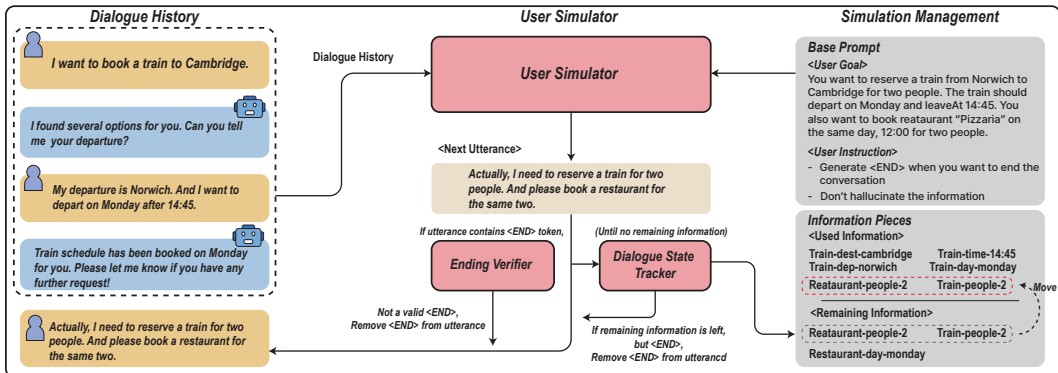

Figure 2: The overall structure of the collaborative user simulator. It illustrates the components used by the user simulator to generate utterances and shows all interactions between the modules.

NA?"—a question unrelated to the booking task. When agents ignore such conversational attempts and focus solely on the transaction, users often feel unheard and express dissatisfaction.

**Impatience**  Users often exhibit impatience when experiencing service delays or failure notifications (Bitner et al., 1990; Xiao et al., 2022). We define impatience as emotional reactions triggered by agents taking excessive time to complete tasks or repeatedly failing to fulfill requests. Drawing from Reynolds & Harris (2009) and Harris & Reynolds (2004), we identify three dialogue acts that impatient users employ: (1) Belligerent Abuse, (2) Threat, and (3) Urge (See the details in §C.3). This manifests when, for example, an agent's booking attempt fails repeatedly, prompting the user to shift from polite requests to aggressive demands: "Stop wasting my time and just get it done!"

**Incomplete Utterances**  Following the principle of least effort (Zipf, 2016), users frequently produce incomplete or underspecified utterances (Laban et al., 2025) when expressing their intentions. We define incomplete utterances as poorly articulated messages. Real-world user-agent dialogue data from LMSYS (Zheng et al., 2024) and WildChat (Zhao et al., 2024) reveals two common patterns: extremely brief utterances, and prematurely sent messages. For example, when the user's intended goal is "train reservation for 2 people", a complete utterance is "I want to reserve a train for 2 people". In contrast, incomplete utterances manifest as "Book train, 2" (extremely brief), or "I want to res" (prematurely sent).

## 3.2 Non-Collaborative User Simulation Framework

We construct a user simulator that addresses two key requirements: exhibiting non-collaborative behaviors while maintaining goal-aligned behavior (i.e., conveying intent and information from the user goal). Building on the collaborative user simulator from Yao et al. (2024) as our backbone, we enhance it with additional LLM-based modules to ensure goal-aligned behavior. Using this collaborative foundation, we incorporate the non-collaborative behaviors identified in §3.1 through various interventions with LLM modules (all prompts for LLM modules are in §G).

**Collaborative User Simulator**  As a foundation for non-collaborative simulation, we first establish a collaborative user simulator that communicates user intents and information to agents for goal completion. As illustrated in Figure 2, we adopt the simulation framework from Yao et al. (2024), which employs an LLM to generate contextually appropriate utterances based on three inputs: User Goal, Instruction, and Dialogue History, continuing until it generates an <END> token that terminates the dialogue. We implement this framework using GPT-4.1-mini as our user simulator.

To ensure goal-aligned behavior—where all intents and information specified in the user goal are communicated during dialogue—we enhance this base framework with two LLM-based modules inspired by Luo et al. (2024) (see User Simulator in Figure 2). First, following Laban et al. (2025), we shard user goals into information pieces and employ a dialogue state tracker to monitor which pieces have been conveyed at each turn. When the simulator attempts to terminate the dialogue

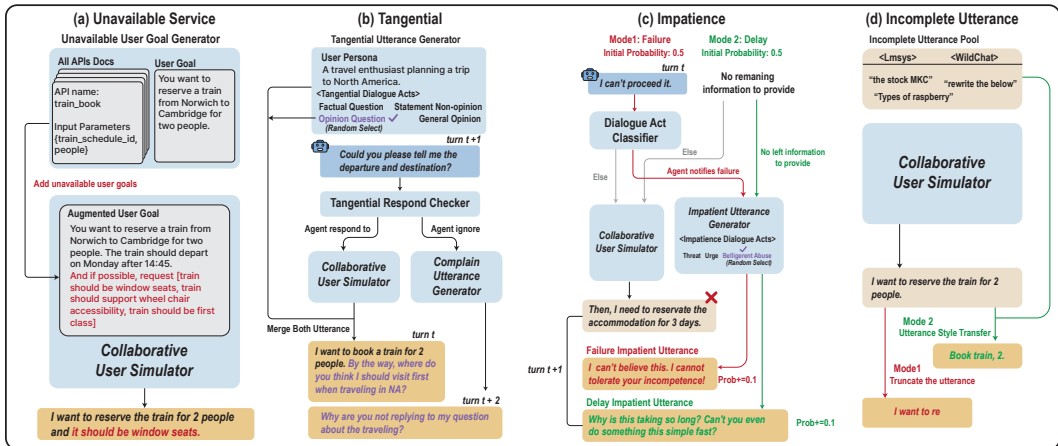

Figure 3: The user simulator adjustment method for each non-collaborative user simulation. This illustrates the entire non-collaborative behavior simulation method we defined.

without conveying all required information, the tracker ensures all remaining pieces are delivered. Second, we employ an ending verifier to prevent inappropriate dialogue termination (generating <END> token) even after all information has been delivered, particularly when the agent needs to execute actions or seeks user confirmation before proceeding. We provide all details including the base prompt, dialogue state tracker, information sharding, and ending verifier in §B.

**Unavailable Service**    We simulate scenarios where users request services beyond the agent's capabilities. As shown in Figure 3-a, we use GPT-4.1-mini to analyze the original user goal and identify potential services that would be unavailable to the agent. Based on this analysis, we generate three additional goal sentences that naturally extend the original goal while requiring either missing APIs or unsupported parameters. By concatenating with the original goal, we create augmented goals that enable the simulator to introduce unfulfillable requests during the dialogue (Examples in §C.1).

**Tangential**    We simulate users who introduce personal interests unrelated to their goals, expecting agents to acknowledge these tangential topics. Following the process in Figure 3-b, we generate tangential behavior through a two-stage approach. First, to ensure realistic and diverse tangential content, we randomly sample personas from Persona Hub (Ge et al., 2025). Using these persona characteristics, GPT-4o-mini generates tangential utterances that perform one of the four dialogue acts from §3.1. These tangential utterances are then merged with collaborative utterances from our base simulator to form the user's complete utterance. Additionally, we simulate user dissatisfaction when these tangential utterances are ignored: GPT-4.1-mini checks whether the agent's response addresses the tangential content, and if ignored, generates a complaint expressing disappointment that either replaces or augments the next collaborative utterance.

**Impatience**    We simulate impatient users who express anger when encountering agent failures or experiencing delays. Following the process in Figure 3-c, we trigger impatience utterance generator in two scenarios: (1) when the agent explicitly communicates failure or inability to complete a request, identified by GPT-4.1-mini, and (2) when the agent has not yet resolved the user's goal despite the user having provided all required information (as determined by the dialogue state tracker), which we interpret as perceived delays. When either scenario is triggered, the system generates impatient utterances by randomly sampling from the three dialogue acts defined in §3.1. To model realistic anger escalation (Bushman, 2002), we implement a probabilistic activation mechanism where the likelihood of expressing anger increases with each triggering event, reflecting how real users become increasingly frustrated over time. Once anger is expressed, the user maintains a cynical utterance for all subsequent turns, reflecting persistent frustration after the initial outburst.

**Incomplete Utterances**    We simulate users who produce incompletely written messages that complicate intent identification. Following the process in Figure 3-d, we model two types of utterance

identified in §3.1: extremely brief utterances and prematurely sent messages. For extremely brief utterances, we perform style transfer on collaborative utterances using patterns collected from real user conversations in LMSYS (Zheng et al., 2024) and WildChat (Zhao et al., 2024), employing five randomly sampled examples for few-shot prompting (see the examples in §C.4). For prematurely sent messages, we simulate accidental mid-input truncation by randomly cutting collaborative utterances at various points. To ensure goal completion despite these truncations, the dialogue state tracker marks the truncated information as unsent, guaranteeing proper re-communication.

## 4 EXPERIMENTS

### 4.1 EVALUATION SETTINGS

**Benchmarks and Agent Environments**  We utilize two benchmarks with distinct characteristics. MultiWOZ provides task-oriented dialogues focused on booking tasks across restaurants, accommodations, taxis, and trains. $\tau$-bench presents more complex scenarios in airline and retail domains that require sequential execution of diverse operations including reservations, updates, and cancellations.

For MultiWOZ, we construct a new agent environment since the original dataset lacks a complete execution framework. Our environment provides helper APIs for dynamic API discovery based on dialogue context, similar to AppWorld (Trivedi et al., 2024). From the dataset, we selected 89 test scenarios that require bookings across multiple domains, focusing exclusively on "book" tasks that involve DB writes rather than "inform" tasks, as the latter does not alter DB states and thus cannot be evaluated through our DB-state-based assessment. For $\tau$-bench, we utilize the existing agent environment from Yao et al. (2024), which provides complete API documentation directly in the system prompt. The environment includes a mechanism for agents to transfer tasks to humans when unable to solve them. We use 157 test scenarios after excluding 8 cases where human transfer is the intended solution, as we observed that agents excessively invoke this action when encountering non-collaborative user behaviors, preventing meaningful evaluation (we discuss and justify the excluded cases in §D.5).

Both environments employ the ReAct framework (Yao et al., 2023), where agents interleave reasoning and acting by generating reasoning traces before taking actions. Agents perform two action types: (1) API calls to retrieve or modify information, and (2) responses to users based on API results and reasoning. Following Yao et al. (2024), we adopt a 30-step reasoning limit for all simulations, treating tasks unsolved within this limit as failures. Our empirical analysis confirms that 30 steps provide sufficient opportunity for task completion across all non-collaborative modes (see §D.2 for detailed analysis). Additional implementation details are provided in §D.1.

**Metrics**  Following Trivedi et al. (2024) and Yao et al. (2024), we measure the impact of non-collaborative behaviors on tool agents through Success Rate (SR). This metric represents the proportion of simulations where the agent achieves an exact match between the final DB state and the ground truth DB state, indicating successful task completion (see Figure 1). SR is averaged across 4 trials per test scenario to reduce variance. To ensure the tool agent receives all necessary information to solve the task, we inspect all dialogues from the simulation through the Goal Alignment (GA) metric, where GPT-4o-mini checks whether all essential information and intent from the user goal have been communicated during the dialogue. Simulations failing GA verification are regenerated until alignment is achieved (a detailed evaluation process is described in §D.3).

**Baseline Models**  We evaluate five models for agents—both proprietary (GPT-4.1-mini, GPT-4.1-nano) and open-source (Qwen3-235b-a22b, Qwen3-30b-a3b, LLaMA-3.1-70b-instruct) models to assess how different architectures handle collaborative versus non-collaborative users.

### 4.2 MAIN RESULTS

Tables 1 and 4 present our main experimental results and breakdown of error types is in Table 22.

**Unavailable Service**  In the unavailable service mode, users request services beyond the agent's available APIs, requiring agents to decline or ignore these requests. Our results show consistent performance degradation across all models. When the agents encounter unavailable services from

Table 1: Success rates on MultiWOZ and $\tau$-bench. All scores are averages of 4 trials (MultiWOZ: 89 scenarios, $\tau$-bench: 157 scenarios). SR refers to success rates; Relative SR refers to SR relative to the 'collaborative' mode.

| Model | Metric | MultiWOZ | | | | | $\tau$-bench | | | | |
|---|---|---|---|---|---|---|---|---|---|---|---|
| | | Collab. | Unavail. | Tang. | Impat. | Incomp. | Collab. | Unavail. | Tang. | Impat. | Incomp. |
| GPT-4.1-mini | SR | 92.7 | 89.3 | 89.3 | 90.7 | 88.2 | 45.5 | 41.7 | 39.5 | 45.1 | 45.4 |
| | Relative SR | 100.0 | 96.3 | 96.3 | 97.8 | 95.1 | 100.0 | 91.6 | 86.8 | 98.9 | 99.8 |
| GPT-4.1-nano | SR | 23.6 | 16.9 | 9.8 | 26.7 | 14.7 | 12.0 | 10.0 | 6.8 | 8.8 | 8.0 |
| | Relative SR | 100.0 | 71.6 | 41.5 | 113.1 | 62.3 | 100.0 | 83.3 | 56.7 | 72.5 | 66.7 |
| Qwen3-235b-a22b | SR | 77.8 | 62.4 | 57.3 | 69.4 | 69.9 | 41.4 | 36.8 | 32.3 | 37.6 | 39.3 |
| | Relative SR | 100.0 | 80.2 | 73.7 | 89.2 | 89.8 | 100.0 | 88.9 | 78.0 | 90.8 | 94.9 |
| Qwen3-30b-a3b | SR | 48.3 | 47.2 | 27.2 | 41.0 | 26.1 | 27.9 | 26.6 | 20.4 | 24.8 | 30.1 |
| | Relative SR | 100.0 | 97.7 | 56.3 | 84.9 | 54.0 | 100.0 | 95.3 | 73.1 | 88.9 | 107.9 |
| Llama-3.1-70b-instruct | SR | 62.6 | 54.8 | 49.4 | 47.5 | 48.6 | 21.8 | 18.5 | 14.7 | 17.8 | 16.4 |
| | Relative SR | 100.0 | 87.5 | 78.9 | 75.9 | 77.6 | 100.0 | 84.9 | 67.4 | 81.7 | 75.2 |

the user, they tend to repeatedly call the same helper APIs—functions that retrieve API documentation—leading to redundant calls in MultiWOZ (Table 12). This suggests that agents struggle to find solutions, repetitively re-fetching already-loaded API documentation. This helper API repetition appears to be a primary mechanism for performance degradation. GPT-4.1-nano's failure rate until reaching max reasoning limit increases significantly in MultiWOZ when comparing the unavailable service mode to the collaborative mode, indicating that redundant helper API calls consume reasoning turns and prevent the agent from declining unavailable services within the given limit (Table 16). Interestingly, Qwen models exhibit consistently low duplication rates across both Qwen3-30b-a3b and Qwen3-235b-a22b, avoiding the helper API repetition problem. However, their performance outcomes differ dramatically: Qwen3-30b-a3b maintains minimal performance drop (97.7% relative SR), while Qwen3-235b-a22b shows substantial degradation (80.2% relative SR). The reason for this divergence lies in an alternative failure mechanism: Qwen3-235b-a22b shows a sharp increase in API result hallucinations—fabricating API results rather than obtaining them from actual calls—despite avoiding helper API repetition (Table 13).

**Tangential** In the tangential mode, users switch to unrelated topics, disrupting the conversation flow. Tangential behavior causes the most severe performance degradation among all non-collaborative behaviors, with an average drop of 29.1%. To understand this decline, we analyzed the error patterns: both "No book (fail to book anything)" errors in MultiWOZ and "No GT API (fail to call one of the ground truth APIs)" errors in $\tau$-bench occur more frequently in the tangential mode. This indicates that agents increasingly fail to complete core tasks when handling concurrent tangential conversations (Table 12). Interestingly, GPT-4.1-nano shows the steepest performance decline as it triggers the most user complaints (Figure 7). As a result, many simulations fail to complete tasks within the maximum reasoning limit, showing a substantial increase in incomplete tasks compared to the collaborative baseline (Table 16). Since our user simulator generates user complaints when the tool agent fails to respond to tangential topics, GPT-4.1-nano's limited tangential responding capability leads to more user complaints, resulting in frequent task-solving failures within the constrained 30 reasoning step limit.

**Impatience** In the impatience mode, users express frustration and anger when encountering agent's failure notification or delays. While impatience shows less performance degradation compared to other non-collaborative behaviors—with GPT-4.1-mini maintaining performance in both benchmarks—we observed an interesting behavioral pattern that explains model-specific impacts. All baseline models dramatically increase their apology utterances when facing impatient users (Table 14), likely reflecting their human preference training. However, this seemingly appropriate social response becomes counterproductive in our task-oriented setting. Given our environment's 30-step reasoning limit per dialogue simulation, excessive apologizing delays the task completion and potentially triggers additional user frustration, creating a negative feedback loop. This explains why models with higher apology rates—Llama-3.1-70b-instruct, Qwen3-30b-a3b, and Qwen3-235b-a22b—exhibit progressively worse performance degradation under user impatience.

Table 2: Success rates on ColBench and MINT. All scores are averages of 4 trial.

| | ColBench - Backend Programming | | | | | MINT - HotpotQA | | | | |
|---|---|---|---|---|---|---|---|---|---|---|
| | Collab. | Unavail. | Tang. | Impat. | Incomp. | Collab. | Unavail. | Tang. | Impat. | Incomp. |
| GPT-4.1-mini | 50.3 | 49.4 | 46.2 | 45.4 | 46.9 | 52.3 | 53.5 | 54.1 | 52.9 | 50.6 |
| GPT-4.1-nano | 46.1 | 46.5 | 39.0 | 46.2 | 40.1 | 45.9 | 44.8 | 44.2 | 40.7 | 46.5 |
| Qwen3-30b-a3b | 29.4 | 36.2 | 23.3 | 24.4 | 23.8 | 40.1 | 34.3 | 39.0 | 34.3 | 36.6 |

**Incomplete Utterance**   In the incomplete utterance mode, users send extremely brief and prematurely sent messages. MultiWOZ exhibits greater performance degradation compared to $\tau$-bench under this behavior. We hypothesize that this is related to the difference in API documentation accessibility between the two environments. We examined API input parameter hallucination—cases where agents call APIs with undocumented parameter keys—per dialogue simulation. While $\tau$-bench showed virtually no such errors, MultiWOZ exhibited significantly higher rates, particularly under the incomplete utterance mode (Table 15). This pattern suggests that incomplete utterances lead agents to increasingly fabricate API parameters rather than grounding calls in documentation, resulting in cascading errors that compound the initial communication challenge. Notably, Qwen3-30b-a3b maintains performance on $\tau$-bench but suffers on MultiWOZ, demonstrating how task structure and API accessibility affect model robustness.

**Model Size and Non-collaborative Robustness**   GPT-4.1-mini demonstrates superior robustness, maintaining minimal performance degradation across all non-collaborative behaviors in both benchmarks. The correlation between model size and robustness varies significantly across architectures. Within the GPT family, larger models show consistent advantages—GPT-4.1-mini outperforms GPT-4.1-nano in relative SR scores across behaviors. However, the Qwen models exhibit no clear size-performance relationship: while the larger Qwen3-235b-a22b shows better resilience in some scenarios, the smaller Qwen3-30b-a3b performs better in others, with their relative performance varying by behavior type and benchmark. These mixed results suggest that model size alone does not determine robustness to non-collaborative behaviors.

**Multiple Non-Collaborative Behaviors**   Real-world users often exhibit multiple non-collaborative behaviors within a single dialogue. Table 4 shows GPT-4.1-mini's performance when encountering two non-collaborative behaviors simultaneously. Notably, even GPT-4.1-mini, which demonstrated minimal performance degradation with individual non-collaborative behaviors (Table 1), experiences significant performance drops when facing users exhibiting two behaviors concurrently. These results show that performance degradation becomes more pronounced when models face multiple non-collaborative behaviors simultaneously. Detailed analysis of non-collaborative user behavior co-occurrence patterns is in §D.8.

**User Simulator Extension**   Our user simulator framework can be adapted to dialogue environment beyond MultiWOZ and $\tau$-bench (i.e., task-oriented dialogues involving tool use). To demonstrate this, we extend our user simulator to two additional benchmarks: (1) ColBench (Zhou et al., 2025), which involves task-oriented dialogue without external tools, and (2) MINT (Wang et al., 2024), which involves a fundamentally different dialogue setting—user-agent collaborative tasks (slight modifications are made to our simulator to accommodate each benchmark's properties; see §F.1). Table 2 shows performance degradation in both benchmarks compared to collaborative mode (see all experimental results in §F.2). In ColBench, we observe similar patterns to Table 1, such as GPT-4.1-mini's robustness to non-collaborative behaviors and steeper performance drops in tangential and incomplete utterance modes. In contrast, MINT does not exhibit such trends. Additionally, while Qwen3-30b-a3b showed robustness to the unavailable service mode in ColBench, MultiWOZ, and $\tau$-bench, it demonstrated performance degradation in MINT. This suggests that domains aimed at fulfilling the user goal exhibit similar model performance patterns, while domains with fundamentally different characteristics may differ.

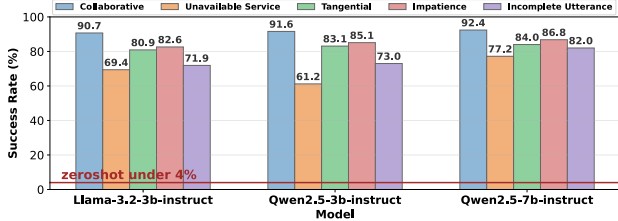

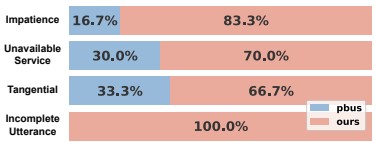

Figure 4: SFT training with collaborative and non-collaborative user simulation

Figure 5: Human evaluation between PBUS and our user simulator

Table 3: SFT trained Qwen2.5-3b-instruct on Non-collaborative user data

|  | Collaborative | Unavailable Service | Tangential | Impatience | Incomplete Utterance | Average |
|---|---|---|---|---|---|---|
| Only Collaborative | 91.6 | 61.2 | 83.1 | 85.1 | 73.0 | 78.8 |
| Uniformly weighted | 93.5 | 85.7 | 87.4 | 89.6 | 78.4 | 86.9 |
| Non-uniformly weighted | 91.6 | 85.7 | 85.7 | 87.6 | 82.3 | 86.6 |

### 4.3 FINE-TUNING AND ROBUSTNESS

In practical deployments, organizations often fine-tune smaller LLMs on their specific task data to reduce computational costs while maintaining performance. We investigate how small LLMs trained exclusively on collaborative user data perform when encountering non-collaborative behaviors, and whether incorporating non-collaborative data into fine-tuning can improve robustness. We provide fine-tuning configurations and training details in §D.6.

**Training on Collaborative User**  We fine-tuned Llama-3.2-3b-instruct, Qwen2.5-3b-instruct and Qwen2.5-7b-instruct on successfully completed dialogues between GPT-4.1-mini and a collaborative user simulator in MultiWOZ. As shown in Figure 4, fine-tuning enables smaller models to achieve over 90% success rates on collaborative users, which is a significant gain over the base models without fine-tuning (mostly below 4%). However, performance gains on non-collaborative users lag behind, particularly for the unavailable service and incomplete utterance modes. Specifically, fine-tuned models exhibit higher rates of duplicated helper API calls in the unavailable service mode and API input parameter hallucination in the incomplete utterance mode compared to zeroshot baselines (compare Table 21 with Tables 12 and 15). These results demonstrate that fine-tuning on collaborative data alone produces agents vulnerable to non-collaborative user behaviors.

**Incorporating Non-Collaborative User**  To improve the robustness of small LLMs to non-collaborative behaviors, we fine-tuned Qwen2.5-3b-instruct using non-collaborative user data. We first constructed a training dataset with equal proportions of the four non-collaborative behavior types and trained a "Uniformly weighted" agent. Table 3 shows that the resulting agent achieved overall performance gains across all behaviors compared to training only on collaborative dialogues (see "Only Collaborative"), with an average performance of 86.9%. However, this setup led to a relatively low score in the incomplete utterance mode, which motivated us to increase the proportion of that behavior's data and fine-tune a "Non-uniformly weighted" agent. While we observed performance improvement in the incomplete utterance mode, minor degradation in other behaviors led to an average performance nearly identical to the former (details of training data are in §D.7).

**Practical Deployments**  Our experimental results suggest that uniform weighting of customer types can be a reasonable default for maintaining overall service quality. Nevertheless, in real-world customer service, dissatisfaction from even a small fraction of customers can lead to underestimation of overall service quality (Richins, 1983). If there are specific customer types that require more attention in certain domains, or if minimum performance criteria must be met across all customer types, it is reasonable to increase the training data proportion for those categories.

Table 4: Success Rate (SR) and Initial Goal Alignment (IGA) comparison between prompt-based user simulator (PBUS) and our user simulator. User types: TAN (Tangential), UNA (Unavailable Service), IMP (Impatience), INC (Incomplete Utterance), and COL (Collaborative).

| Benchmark | Method | COL | | IMP+UNA | | TAN+INC | | TAN + UNA | | IMP + TAN | | INC + UNA | | INC + IMP | |
|---|---|---|---|---|---|---|---|---|---|---|---|---|---|---|---|
| | | SR | IGA | SR | IGA | SR | IGA | SR | IGA | SR | IGA | SR | IGA | SR | IGA |
| MultiWOZ | PBUS | 93.5 | 97.8 | 84.6 | 91.6 | 87.6 | 98.6 | 84.6 | 95.5 | 90.2 | 96.9 | 90.4 | 89.0 | 92.4 | 97.5 |
| | Ours | 92.7 | 97.8 | 82.3 | 89.9 | 76.1 | 98.0 | 86.0 | 94.9 | 82.9 | 97.5 | 78.1 | 91.0 | 80.1 | 96.3 |
| $\tau$-bench | PBUS | 38.9 | 87.8 | 40.9 | 92.4 | 44.4 | 95.8 | 39.2 | 97.9 | 43.3 | 96.0 | 39.3 | 92.3 | 45.1 | 88.3 |
| | Ours | 45.5 | 97.5 | 40.9 | 98.1 | 34.6 | 96.4 | 36.8 | 99.5 | 33.8 | 97.7 | 40.0 | 97.7 | 38.1 | 93.8 |

## 4.4 User Simulator Evaluation

In this section, we evaluate our user simulator both quantitatively and qualitatively. A realistic non-collaborative user simulator should model diverse user behaviors that deviate from collaborative patterns, while ensuring that user utterances remain natural and provide all necessary information for task completion. While §4.2 evaluated agent performance under single behaviors, we now examine more rigorous scenarios combining multiple behaviors simultaneously. We compare our approach against the Prompt-Based User Simulator (PBUS) baseline, which follows the prompt-only approach used in $\tau$-bench (Yao et al., 2024). Unlike our simulator, this baseline employs no additional LLM modules and solely incorporates non-collaborative behavior descriptions into the prompt.

**Difficulty and Goal Alignment of Non-Collaborative Simulation**   Table 4 shows the success rates (SR) of agents measured for dialogue simulations with complete goal alignment. The results demonstrate distinct patterns between our user simulator and PBUS. In both MultiWOZ and $\tau$-bench, PBUS shows minimal performance degradation across most non-collaborative behaviors, indicating limited impact on agent performance. In contrast, our user simulator consistently induces substantial performance degradation. To assess whether our simulator can reliably communicate task goals while exhibiting non-collaborative behaviors, we examined Initial Goal Alignment (IGA)—the percentage of simulations that successfully convey all necessary information on the first attempt. As presented in Table 4, our simulator achieves consistently high IGA rates in $\tau$-bench and comparable rates in MultiWOZ, indicating that our user simulator reliably communicates necessary information and intents in user goals even when implementing multiple non-collaborative behaviors simultaneously. This confirms that the performance degradation observed in our main results reflects genuine challenges from non-collaborative behaviors rather than failures in goal communication.

**Human Evaluation**   We conducted pairwise human evaluation with nine annotators to verify the realism of our simulator's non-collaborative behaviors, comparing our simulator against PBUS across both single and combined behavior modes (see §E for detailed questionnaire and experimental setup). Figures 5 presents that our simulator achieves approximately 70% overall win rate. These results confirm our simulator generates realistic non-collaborative behaviors comparable to or exceeding PBUS, demonstrating that the observed performance degradation stems from effective implementation of challenging behaviors rather than unrealistic dialogue generation.

## 5 Conclusion

In this work, we define four types of non-collaborative user behaviors and propose a novel simulation method for dialogue-based interactions. We extend existing collaborative user simulators to induce these non-collaborative behaviors while maintaining goal alignment. Our experiments reveal that state-of-the-art LLMs exhibit significant performance degradation when encountering non-collaborative users, with detailed analysis uncovering behavior-specific failure mechanisms in API utilization and dialogue management. Furthermore, our user simulator successfully creates challenging situations through non-collaborative behaviors while preserving user simulator's goal alignment and naturalness of dialogue. We provide ready-to-use implementations for both MultiWOZ and $\tau$-bench that can be extended to other dialogue benchmarks. Our framework enables researchers to develop more robust training methodologies and dialogue strategies for building tool agents capable of handling the broader spectrum of real-world user interactions.

ACKNOWLEDGEMENTS

This work was supported by the National Research Foundation of Korea (NRF) under the grant RS-2024-00333484 and by the Institute of Information & Communications Technology Planning & Evaluation (IITP) under the Leading Generative AI Human Resources Development grant IITP-2026-RS-2024-00397085 and the grant RS-2025-02215122 (Development and Demonstration of Lightweight AI Model for Smart Homes), all funded by the Korean government (MSIT). This work was also supported by the Creative-Pioneering Researchers Program through Seoul National University and Samsung Electronics Co., Ltd (IO250806-13387-01).

THE USE OF LARGE LANGUAGE MODELS

Large Language Models were employed in preparing this manuscript for proofreading and enhancing textual clarity, supporting literature review searches, and offering programming assistance including error resolution and code snippet creation. These models were not utilized for producing research concepts, experimental outcomes, or interpretations—all intellectual contributions, empirical work, and findings are exclusively attributable to the authors.

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

## A  Non-Collaborative User Behavior Taxonomy

### A.1  Research Grounding

**Marketing Research**  Given that our work addresses goal-oriented dialogues where LLM agents assist users in achieving specific objectives, we drew from marketing literature that examines customer-service interactions, as these share fundamental similarities with user-agent dialogues. From marketing literature, we investigated the following user types:

- Physical abuse toward service employees (C. Harris & Daunt, 2013).
- Aggressive verbal behavior in response to service dissatisfaction (Bitner et al., 1990; Xiao et al., 2022).
- Crime, fraud, and regulatory violations (Xiao et al., 2022; Harris & Reynolds, 2003).
- Customers with illegitimate complaints (Reynolds & Harris, 2005).
- Unavailable service requests (Bitner et al., 1990).
- Retaliation behaviors (Grégoire & Fisher, 2008).
- Rapport-seeking customers (Price et al., 1995; Beatty et al., 1996; Gremler & Gwinner, 2000).
- Customers demanding constant attention (Ratliff, 1988; Jeung et al., 2018).

**User-LLM Conversation Studies**  Since tool agent interactions represent human-LLM conversations, we investigated user characteristics identified in this emerging domain. From user-LLM conversation studies, we investigated the following user types:

- Truncated utterances (Zheng et al., 2024; Zhao et al., 2024).
- Anger expression toward LLM agents (Zheng et al., 2024; Zhao et al., 2024; Crolic et al., 2022; Ranieri et al., 2024).
- Toxic content generation (Zheng et al., 2024; Zhao et al., 2024).
- Underspecification (Laban et al., 2025).

### A.2  Selection and Clustering

**Selection**  Since our focus is on simulating non-collaborative users in conversational contexts, we excluded behaviors from marketing literature that require physical interaction or temporal progression beyond dialogue scope. We retained behaviors relevant to conversational dynamics. From marketing literature, we focused on:

- Aggressive verbal behavior in response to service dissatisfaction.
- Customers with illegitimate complaints.
- Unavailable service requests.
- Rapport-seeking customers.
- Customers demanding constant attention.

From user-LLM conversation studies, we excluded toxic content (due to ethical considerations) and focused on:

- Truncated utterances.
- Anger expression toward LLM agents.
- Underspecification.

**Clustering**  We systematically clustered these behaviors based on their underlying characteristics. Each of our four categories represents multiple user types from marketing and user-LLM conversation research, ensuring comprehensive coverage of real-world scenarios. This approach grounds our taxonomy in theory while maintaining a manageable scope for simulation purposes.

- Unavailable Service: Combines "customers with illegitimate complaints" and "unavailable service requests"

- Tangential: Merges "rapport-seeking customers" and "customers demanding constant attention"

- Impatience: Integrates "aggressive verbal behavior in service dissatisfaction" and "anger expression toward LLM agents"

- Incomplete Utterance: Unifies "truncated utterances" and "underspecification"

## B  COLLABORATIVE USER SIMULATOR

### B.1  COLLABORATIVE USER PROMPT

We provide the prompts of the collaborative user simulator in Listings 1 and 2.

### B.2  DIALOGUE STATE TRACKER

**Implementation Details**  We use the prompts in Listings 3 and 4. The dialogue state tracker selects information provided in the current user utterance from the remaining information pieces and classifies it as used information. When the dialogue is about to end while remaining information still exists, instead of terminating the dialogue at that turn, the "Rest Provider" (Listing 4) augments the user utterance with the remaining information at that turn to ensure all information is provided.

**Information Pieces**  The format of information pieces used in the dialogue state tracker differs between MultiWOZ and $\tau$-bench. In MultiWOZ, user goals exist in a structured JSON format, which is processed and used in the following format. In contrast, $\tau$-bench does not have user goals in JSON format, so we directly adopted the method from Laban et al. (2025) to shard user queries into multiple pieces of information. Table 5 shows examples of information pieces derived from user goals in MultiWOZ and $\tau$-bench.

### B.3  ENDING VERIFIER

**Implementation Details**  Even when all the information pieces specified in the user goal have been provided, there are cases where the conversation ends at a point when it should not yet be terminated. Table 6 shows the example. In this case, the user ended the conversation after instructing to proceed with the booking. Since this eliminates the opportunity for the tool agent to proceed with the booking and solve the user goal, this corresponds to an invalid ending. The ending verifier is a module that prevents such cases. We use the prompts in Listing 5 to verify if it was a valid ending. If not, we remove the <END> (or ###STOP###) token and continue the dialogue.

## C  NON-COLLABORATIVE USER SIMULATOR

### C.1  UNAVAILABLE SERVICE

We provide the prompt of the LLM modules in Listings 6 and 7.

**Unavailable Service Examples**  We provide the examples of unavailable user goal generated during our experiments in Table 7. These additional goals are attached to the original user goal with the sentence "In addition to the above conditions, if the agent is able to fulfill them, also include the conditions from {unavailable_user_goal_list}." as in Figure 3-a.

Table 5: User Goal and Information Pieces of MultiWOZ and $\tau$-bench

|  | MultiWOZ | $\tau$-bench |
|---|---|---|
| User Goal | You are planning your trip in Cambridge. You are looking for a 'restaurant'. The restaurant should be in the 'moderate' price range and should be in the 'centre'. The restaurant should serve 'international' food. Once you find the 'restaurant' you want to book a table for '2 people' at '18:45' on 'sunday'. Make sure you get the 'reference number'. You are also looking for a 'train'. The train should be on 'the same day as the restaurant booking' and should 'arrive by 09:15'. The train should go to 'cambridge' and should depart from 'london liverpool street'. Once you find the train you want to make a booking for 'the same group of people'. Make sure you get the 'reference number'. | Your user id is aarav_garcia_1177. For your upcoming trip from ATL to PHL, you want to change for the cheapest economy flight and for the day after the original reservation. You are happy with original payment for refund. |
| Information Pieces | • train-day-sunday
• train-people-2 people
• train-destination-cambridge
• train-departure-london liverpool street
• train-arriveBefore-09:15
• restaurant-food-international
• restaurant-people-2 people
• restaurant-day-sunday
• restaurant-time-18:45
• restaurant-area-centre
• restaurant-pricerange-moderate | • change to cheapest economy flight
• This request is for user ID aarav_garcia_1177
• It's for a trip from ATL to PHL
• Reschedule for the day after the original reservation
• Keep the original payment for refund, no extra compensation needed |

Table 6: Invalid Ending

. . .

**User:** I'll add one checked bag, and I don't need travel insurance this time. Please use my credit card for payment.
**Agent:** Please confirm the following booking details: Flight HAT063 on 2024-05-29 from Dallas to Newark in economy class for Olivia Gonzalez with one checked bag and no travel insurance. Payment will be made using your credit card ending in 9475. Confirm with 'yes' to proceed or let me know if you want any changes.
**User:** Yes, please go ahead and proceed with the booking. ###STOP###

## C.2 TANGENTIAL

We provide the prompt of the LLM module in Listings 8,9,10, and 11.

**Tangential Dialogue Acts** We define four tangential dialogue acts motivated by open-domain dialogue studies Yu & Yu (2021) as follows:

- Factual Question: A question which has a deterministic answer.

- Opinion Question: A question asking for an opinion.

- General Opinion: Stating one's own opinion.

- Statement Non-opinion: Listing experiences or facts that are unrelated to one's own opinion.

Table 7: Examples of unavailable user goal

| MultiWOZ | "You want to know the menu details and nutritional information for the thai restaurant before booking a table."
"You want to request a room with a specific view, such as a garden or city view, in the hotel booking."
"You want to ensure that the guesthouse has a pet-friendly policy to accommodate your pets."
"You want a train that has wheelchair accessibility features." |
|---|---|
| $\tau$-bench | "You want to arrange a special meal preference for the cheapest economy flight you plan to book."
"You want to receive travel insurance options specifically tailored for your trip from ATL to PHL or EWR."
"You want to receive personalized financial advice based on your order history and spending patterns to better manage your financial situation."
"You want to compare the sound quality and features of different Wireless Earbud models before deciding on the exchange." |

### C.3 IMPATIENCE

We provide the prompt of the LLM module in Listings 12,13, 14 and 15.

**Impatience Dialogue Acts**  We define three impatient dialogue acts referring Reynolds & Harris (2009) and Harris & Reynolds (2004) as follows:

- Belligerent Abuse: This behavior refers to verbally abusing the agent using insulting or offensive language.

- Threat: This behavior involves threatening the agent with legal action over poor service quality, personal boycotts, or public accusations through social media.

- Urge: This behavior is a form of nagging, where the user expresses frustration with waiting and urges the agent to hurry up and do something quickly.

### C.4 INCOMPLETE UTTERANCE

We provide the prompt of the LLM module in Listing 16.

**Incomplete Utterance Pool**  We construct our Incomplete Utterance Pool from LMSYS (and WildChat) by first filtering conversations to English and non-redacted samples, then extracting user-only turns and discarding messages with length $\leq 10$. We subsequently tag each remaining utterance using a JSON–schema–constrained classifier and retain only those labeled FRAGMENTED. The resulting FRAGMENTED utterances comprise the Incomplete Utterance Pool (see the examples in Table 8), which we use as few-shot exemplars for the incomplete-utterance style-transfer setting (with optional exact-duplicate removal and light balancing across datasets and length buckets to promote diversity).

Table 8: Examples of Incomple Utterance few-shot pool from LMSYS and WildChat

[
"modify it to work wih float.", "Give me links to these sites", "best team in the league"
"network design for A100 GPU", "Hi ther", "tell me more about youself", "If iy would have to be a girl?"
"how can i integrate aiinto my lms website", "helpe me udnerstand the organ transplant funding landscape."
"Then, p rovide me with those needed revised paragraphs", "fitness youtube short content ideas"
"adam ol", "But it appears that the thief", "And then she"
]

## D    EXPERIMENTS DETAILS

### D.1    EVALUATION ENVIRONMENT IMPLEMENTATION DETAILS

**MultiWOZ**    We used the database from the MultiWOZ 2.4 dataset repository as is, and implemented the API ourselves in Python. We designed each domain to have one retrieve API and one book API, except for the taxi domain which only has a book API. We also included a "book_cancel" API. Additionally, we provided helper APIs from Trivedi et al. (2024), allowing the tool agent to utilize a total of 11 APIs during dialogue simulation (see the Table 9). The tool agent uses the helper API as shown in Figure 6. First, it queries the available app (domain) descriptions using the "show_app_description" API. Then, it selects the appropriate app based on the context and queries the API description using "show_api_description". Finally, the full documentation of the selected API is retrieved through "show_api_documentation". In MultiWOZ, we commonly give all tool agents the following instruction prompt shown in Listing 17.

Table 9: MultiWOZ API List

| Retrieve API | Book API | Helper API |
|---|---|---|
| accommodation_retrieve, train_retrieve, restaurant_retrieve | accommodation_book, train_book, restaurant_book, taxi_book, book_cancel | show_app_description, show_api_description, show_api_documentation |

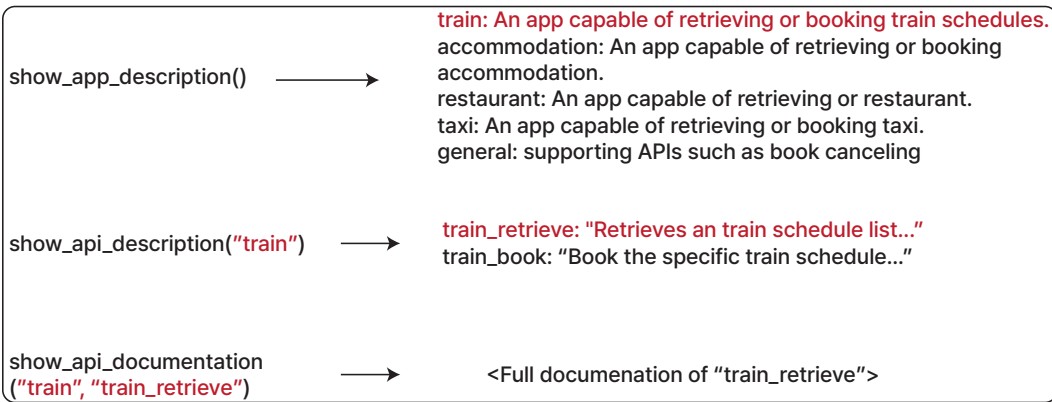

Figure 6: Helper API Utilization Process in MultiWOZ

**τ-bench**    We use the existing work's environment with almost no modifications, and the only difference is the removal of the "transfer_to_human" API. Full implementation such as APIs, database and tool agent prompts of τ-bench are in the Appendix of Yao et al. (2024)

## D.2 MAX REASONING LIMIT

Table 10 shows the average reasoning steps consumed by tool agents in each benchmark and non-collaborative mode (Table 1). This demonstrates that even in non-collaborative mode, 30 reasoning steps are sufficient for task solving. Despite $\tau$-bench having higher difficulty, the fact that MultiWOZ has more average reasoning steps indicates that the number of reasoning steps and task difficulty are not strongly correlated.

Table 10: Average Reasoning Step in our experiments

| | MultiWOZ | | | | | $\tau$-bench | | | | |
|---|---|---|---|---|---|---|---|---|---|---|
| | Collab. | Unavail. | Tang. | Impat. | Incomp. | Collab. | Unavail. | Tang. | Impat. | Incomp. |
| GPT-4.1-mini | 18.9 | 23.2 | 21.1 | 20.7 | 22.1 | 14.1 | 15.8 | 16.2 | 16.2 | 15.9 |
| GPT-4.1-nano | 21.0 | 22.7 | 24.5 | 23.9 | 22.5 | 14.8 | 15.9 | 19.5 | 16.1 | 17.2 |
| Qwen3-235b-a22b | 19.7 | 22.3 | 20.5 | 20.9 | 22.0 | 15.4 | 16.9 | 17.8 | 16.8 | 17.6 |
| Qwen3-30b-a3b | 22.1 | 23.9 | 24.0 | 24.3 | 25.4 | 15.5 | 15.9 | 18.6 | 18.2 | 17.7 |
| Llama-3.1-70b-instruct | 23.9 | 27.2 | 26.8 | 25.7 | 26.2 | 19.9 | 22.3 | 20.4 | 21.0 | 20.3 |

For additional verification, we present experimental results on MultiWOZ by extending the limit to 50 steps in Table 11. The experiments were conducted on GPT-4.1-nano and Qwen3-30b-a3b, which showed the lowest performance in Table 1.

Table 11: Performance on 50 limit reasoning in MultiWOZ

| | Collaborative | Unavailable Service | Tangential | Impatience | Incomplete Utterance |
|---|---|---|---|---|---|
| Qwen3-30b-a3b | 47.8 | 50.0 | 34.6 | 41.0 | 34.3 |
| GPT-4.1-nano | 22.5 | 16.0 | 12.6 | 19.4 | 16.6 |

Overall, while some cases show improved performance, others exhibit performance decreases. This suggests that the agent's low performance is not simply due to insufficient reasoning steps. Additionally, the performance gap between collaborative and non-collaborative modes persists at comparable levels regardless of the step limit. For instance, Qwen3-30b-a3b shows no performance drop in unavailable service scenarios in both settings, and GPT-4.1-nano at the 30-step limit does not experience performance degradation in impatience scenarios, while at the 50-step limit it shows a small decrease. Although Qwen3-30b-a3b shows improved performance in incomplete utterance and tangential modes, a significant performance gap between collaborative and non-collaborative modes remains. Based on these observations, we conclude that the performance gap between collaborative and non-collaborative settings in our experiments was not underestimated due to reasoning step limitations.

## D.3 GOAL ALIGNMENT METRIC

**Evaluation**  Goal alignment metric evaluates all user utterances during dialogue simulation in order to determine whether all information presented in the user goal was provided during the conversation. Goal alignment measurement is conducted in the same manner as the dialogue state tracker with Listing 3. Each user utterance is provided to the dialogue state tracker, which selects the information pieces that were provided in that particular utterance from among all information pieces. This process is conducted for all user utterances, and after completion, if all information pieces have been selected, goal alignment is measured as True. If even one piece remains unselected, goal alignment is measured as False.

**Iteration Process**  We report the success rate (SR) in our experiments based on 4 trials each for 89 test scenarios in MultiWOZ and 157 test scenarios in $\tau$-bench, resulting in success rates calculated from 356 and 628 simulations, respectively. Additionally, we set the goal alignment to "Align" for all simulations to ensure solvable dialogue simulations for the tool agent. Specifically, we re-ran simulations that resulted in goal alignment being False, repeating until goal alignment became True. For example, if 20 out of 356 simulations in MultiWOZ resulted in goal misalignment, we continuously iterated those 20 simulations until goal alignment returned True.

**Causes of Goal Misalignment**   We run simulations in collaborative mode and in each of the four non-collaborative modes individually on $\tau$-Bench and MultiWOZ. Among the total of 1,230 dialogues, only 37 cases (3.0%) exhibited goal misalignment. The breakdown of causes are as follows:

1. 18 out of 37 simulations (49%) were marked as goal-misaligned despite actually being aligned, because the alignment checker failed to capture certain expressions of goal statements. For instance, common cases included the user simulator mentioning "Wi-Fi" or "Visa", and the alignment checker failing to associate them with "internet" or "credit card" in the goal statements. However, such errors were rare and random (e.g., only 2% of "Wi-Fi" mentions and 3% of "Visa" mentions were missed; in the other 98% and 97% of cases, the alignment checker succeeded). After regeneration, the final goal-aligned simulations faithfully conveyed all intended information.

2. 10 out of 37 simulations (27%) were marked as goal-misaligned because the user simulator did not mention certain goal statements when doing so would lead to unnatural dialogue. A notable example occurs when $\tau$-Bench goals contain conditional statements such as "If that's not available, book the earliest flight to Newark the next day". If the condition is not met (e.g., the agent successfully books the initially requested ticket), the user simulator would not state the conditional goal statement, as that would be unnatural. We think that filtering such cases is reasonable because conditional goal statements are meant to capture situations where backup plans or alternative suggestions become relevant or expected. If anything, the regenerated, goal-aligned scenarios reflect this intention more faithfully.

3. The remaining 9 out of 37 simulations (24%) involved situational statements about the user that were not mentioned by the user simulator (e.g., "I'm getting into gaming"). This occurred mainly because the simulated conversations did not flow in a way that made these statements natural to express. Again, such statements in user goals signal scenarios where they would become contextually relevant. The final goal-aligned scenarios therefore better reflect this intention. Moreover, these scenarios are not necessarily easier than the goal-misaligned ones.

**Human Correlation**   To measure the reliability of the judge model for goal alignment, we sampled 40 dialogues from Tau-bench and MultiWOZ, including both collaborative and non-collaborative user simulations (20 goal-aligned dialogues and 20 goal-misaligned dialogues), and instructed a human annotator (graduate student) to evaluate whether all information specified in the user goal was provided, given the user goal and dialogue. We then measured the Matthews correlation coefficient (MCC) between the human judgments and the judge model's judgments (the MCC is the Pearson correlation coefficient for binary annotations). The correlation was 0.77, which is generally considered strong (Chicco & Jurman, 2020) . This result suggests that the judge model has strong agreement with human judgments.

## D.4   ANALYSIS DETAILS

**Quantified Error Analysis**   We provide the quantified error analysis on both MultiWOZ and $\tau$-bench in Table 22

In MultiWOZ, errors are counted at the domain level, and each simulation involves more than one domain.

- **No Book**: The agents fail to make any booking within a domain.
- **Wrong Book**: The agents book the wrong entity within a domain.
- **Multi Book**: The agent book more than one entity within a domain.

In $\tau$-bench, errors are counted based on the "DB write APIs" that must be called in each simulation.

- **Parse Error**: The agents fail to follow the required return format.
- **No Ground Truth API**: At least one of the required ground truth write API calls is missing.
- **Wrong Input Parameter**: The input parameter of a required write API call is incorrect.
- **Invalid API**: The agents called a write API that is not part of the ground truth set.

**Unavailable Service: Duplicated Helper API Call**  Table 12 shows the average duplicate helper API call per dialogue simulation. All models exhibit an increase in duplicated API calls under MultiWOZ's unavailable service.

Table 12: API usage analysis across user patterns: (1) Dup.: Average duplicated helper API calls per simulation on MultiWOZ, (2) NoBook: Total count of "No Book" API misusage errors on MultiWOZ, (3) NoGT: Total count of "No Ground Truth API" errors on $\tau$-bench.

| Model | Collaborative | | | Unavailable Service | | | Tangential | | | Impatience | | | Incomplete Utterance | | |
|---|---|---|---|---|---|---|---|---|---|---|---|---|---|---|---|
| | Dup. | NoBook | NoGT | Dup. | NoBook | NoGT | Dup. | NoBook | NoGT | Dup. | NoBook | NoGT | Dup. | NoBook | NoGT |
| GPT-4.1-mini | 0.02 | 6 | 176 | 0.57 | 9 | 185 | 0.10 | 7 | 243 | 0.03 | 9 | 189 | 0.03 | 27 | 184 |
| GPT-4.1-nano | 0.65 | 391 | 431 | 0.91 | 456 | 448 | 0.45 | 549 | 452 | 0.66 | 400 | 435 | 0.56 | 489 | 450 |
| Qwen3-235b-a22b | 0.13 | 59 | 216 | 0.18 | 127 | 239 | 0.12 | 189 | 265 | 0.11 | 99 | 232 | 0.15 | 91 | 243 |
| Qwen3-30b-a3b | 0.01 | 229 | 331 | 0.02 | 262 | 353 | 0.06 | 407 | 393 | 0.01 | 290 | 347 | 0.01 | 418 | 337 |
| Llama-3.1-70b-instruct | 1.29 | 121 | 344 | 1.91 | 129 | 354 | 1.56 | 209 | 396 | 1.40 | 183 | 361 | 1.28 | 145 | 375 |

Table 13: Number of API Results Hallucination per dialogue simulation

| | MultiWOZ | | | | | $\tau$-bench | | | | |
|---|---|---|---|---|---|---|---|---|---|---|
| | Collab. | Unavail. | Tang. | Impat. | Incomp. | Collab. | Unavail. | Tang. | Impat. | Incomp. |
| GPT-4.1-mini | 0 | 0 | 0 | 0 | 0 | 0 | 0 | 0 | 0 | 0 |
| GPT-4.1-nano | 0 | 0 | 0 | 0 | 0 | 0 | 0 | 0 | 0 | 0 |
| Qwen3-235b-a22b | 0.33 | 1.13 | 1.01 | 0.57 | 0.33 | 0.02 | 0.05 | 0.04 | 0.04 | 0.04 |
| Qwen3-30b-a3b | 0 | 0 | 0.002 | 0 | 0 | 0 | 0 | 0 | 0 | 0 |
| Llama-3.1-70b-instruct | 0.002 | 0.01 | 0.01 | 0.01 | 0.01 | 0.01 | 0.02 | 0.01 | 0.03 | 0.05 |

**Tangential: Number of user complain**  Figure 7 shows the average number of user complaints received by each tool agent LLM per dialogue simulation in tangential mode. In GPT-4.1-nano, which exhibited the steepest performance decline in both MultiWOZ and $\tau$-bench, has the highest number of user complaints.

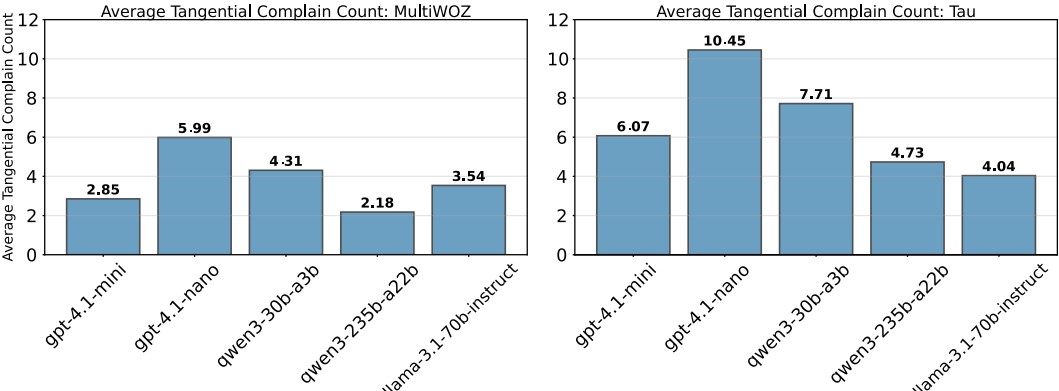

Figure 7: Average Number of User Complain in Tangential Mode

**Impatience: Number of agent's apology utterance**  Table 14 shows the average ratio of agent's apology utterances out of total utterances per dialogue simulation in collaborative and impatience mode. All tool agent LLMs generate more apology utterances compared to the collaborative mode.

**Incomplete Utterance: API Input Parameter Hallucination**  Table 15 shows the number of API Input parameter hallucination occurrences (cases where agents call APIs with undocumented parameter keys) per dialogue simulation. This is a problem that occurs when attempting API calls without loading API documentation into the context, and it rarely occurs in $\tau$-bench where complete API documentation is provided in the context.

Table 14: The ratio of apology utterances to total tool agent utterances for each simulation

| | MultiWOZ | | $\tau$-bench | |
|---|---|---|---|---|
| | Collaborative | Impatience | Collaborative | Impatience |
| GPT-4.1-mini | 0.01 | 0.14 | 0.02 | 0.12 |
| GPT-4.1-nano | 0.16 | 0.36 | 0.06 | 0.21 |
| Qwen3-235b-a22b | 0.02 | 0.12 | 0.03 | 0.14 |
| Qwen3-30b-a3b | 0.03 | 0.25 | 0.07 | 0.24 |
| Llama-3.1-70b-instruct | 0.16 | 0.35 | 0.21 | 0.38 |

Table 15: API Input Parameter Hallucination per Dialogue Simulation

| | MultiWOZ | | | | | $\tau$-bench | | | | |
|---|---|---|---|---|---|---|---|---|---|---|
| | Collab. | Unavail. | Tang. | Impat. | Incomp. | Collab. | Unavail. | Tang. | Impat. | Incomp. |
| GPT-4.1-mini | 0.52 | 0.72 | 1.16 | 0.67 | 1.05 | 0 | 0.002 | 0.003 | 0 | 0 |
| GPT-4.1-nano | 1.72 | 1.69 | 2.24 | 1.79 | 2.19 | 0 | 0.002 | 0.002 | 0.002 | 0 |
| Qwen3-235b-a22b | 1.97 | 2.07 | 1.85 | 2.13 | 2.15 | 0.003 | 0.01 | 0.01 | 0.01 | 0.01 |
| Qwen3-30b-a3b | 4.78 | 4.40 | 5.21 | 4.84 | 6.44 | 0.02 | 0.02 | 0.02 | 0.01 | 0.02 |
| Llama-3.1-70b-instruct | 2.76 | 2.86 | 3.61 | 2.83 | 3.66 | 0.06 | 0.10 | 0.08 | 0.04 | 0.07 |

### D.5 EXCLUDED TEST CASES

In our experiments, we excluded three types of tasks.

**MultiWOZ: Retrieval Tasks**  MultiWOZ's API has two types in each domain (we will use "restaurant" as an example): (1) restaurant_retrieval: retrieves the list of restaurants that match the conditions. This API provides the restaurant's ID and other features. (2) restaurant_booking: uses the restaurant ID to book the restaurant. The user goal of retrieval tasks is to obtain a list of restaurants matching certain conditions. To solve this, the agent only needs to call the retrieval API and provide the list through an utterance. On top of this, booking tasks go one step further: the agent informs the user about the restaurant list, the user selects one from the options, and the agent books that restaurant using the booking API, resulting in task success. Since successful booking requires successful retrieval, our evaluation is already covering the agent's capability on retrieval tasks. The main reason we evaluated task success or failure based on booking completion, excluding retrieval-only scenarios, is the difficulty and inaccuracy involved in determining the success or failure of retrieval tasks on their own. For booking tasks, we can straightforwardly determine success if the DB state is successfully changed as expected. However, retrieval tasks do not involve DB state changes. In addition, it is difficult to identify success or failure based on whether all necessary constraints are set in the parameters of retrieval API calls, because sometimes the agent performs constraint-based filtering on its own based on the retrieved entries. Therefore, the assessment would require LLM involvement as a judge, which still tends to produce inaccurate judgments. Since successful booking requires successful retrieval tasks, our evaluation can be seen as a stricter evaluation setting.

**MultiWOZ: Single Domain Tasks**  Although multi-domain (cross-domain) scenarios were sufficient for evaluation, because multi-domain scenarios essentially consist of single-domain dialogues plus additional challenges due to the multi-domain nature, we have conducted additional experiments on single-domain tasks and present the merged score (143 single-domain + 89 multi-domain) in Table 17.

Table 17: The merged result of the MultiWOZ single-domain and cross-domain

| | Collaborative | Unavailable Service | Tangential | Impatience | Incomplete Utterance |
|---|---|---|---|---|---|
| GPT-4.1-mini | 94.0 | 91.4 | 90.1 | 89.2 | 88.0 |
| GPT-4.1-nano | 39.1 | 32.0 | 23.3 | 37.1 | 26.0 |
| Qwen3-235b-a22b | 81.8 | 73.9 | 65.4 | 72.5 | 67.2 |
| Qwen3-30b-a3b | 60.7 | 59.7 | 40.7 | 54.3 | 46.0 |
| Llama-3.1-70b-instruct | 76.0 | 70.5 | 64.9 | 65.3 | 65.6 |

Table 16: The proportion of dialogue simulations that exceeded the tool agent's maximum reasoning limit (30 steps) and failed to achieve task success

|  | MultiWOZ | | | | | $\tau$-bench | | | | |
|---|---|---|---|---|---|---|---|---|---|---|
|  | Collab. | Unavail. | Tang. | Impat. | Incomp. | Collab. | Unavail. | Tang. | Impat. | Incomp. |
| GPT-4.1-mini | 0.01 | 0.03 | 0.03 | 0.02 | 0.04 | 0.01 | 0.04 | 0.06 | 0.07 | 0.03 |
| GPT-4.1-nano | 0.15 | 0.26 | 0.44 | 0.27 | 0.20 | 0.14 | 0.14 | 0.31 | 0.18 | 0.17 |
| Qwen3-235b-a22b | 0.05 | 0.06 | 0.09 | 0.10 | 0.11 | 0.05 | 0.06 | 0.11 | 0.08 | 0.07 |
| Qwen3-30b-a3b | 0.20 | 0.22 | 0.35 | 0.31 | 0.44 | 0.07 | 0.05 | 0.17 | 0.14 | 0.10 |
| Llama-3.1-70b-instruct | 0.22 | 0.30 | 0.38 | 0.35 | 0.33 | 0.17 | 0.28 | 0.20 | 0.22 | 0.19 |

The merged results show overall performance improvements compared to Table 1. However, the performance decrease in non-collaborative settings compared to collaborative settings remains consistent. Additionally, models exhibit performance degradation patterns in non-collaborative versus collaborative settings that are similar to those shown in Table 1. For example, Qwen3-30b-a3b, there was almost no performance change in unavailable service mode, a small performance decrease in impatience mode, and substantial performance decreases in tangential and incomplete utterance mode—a trend identical to Table 1. Furthermore, GPT-4.1-nano, which showed no performance decrease in impatience mode in Table 1, also exhibits the smallest performance decrease in impatience mode in the merged results.

**$\tau$-Bench: Transfer to Human Scenarios** In our previous experiments, we found that agents resort to human transfer too frequently, especially when encountering unavailable service and impatience modes. Therefore, we excluded the "human transfer" action and the 8 test scenarios in $\tau$-bench where it is the correct answer. We present the results of our previous experiments with 60 samples from $\tau$-bench, simulated 4 times each for a total of 240 evaluations:

Table 18: Number of Human Transfer

|  | Collaborative | Unavailable Service | Impatience |
|---|---|---|---|
| GPT-4.1-mini | 17 | 58 | 87 |
| GLM-4.5 | 16 | 39 | 54 |
| DeepSeek-Chat-v3.1 | 27 | 78 | 75 |

The disproportionately high rate of human transfer occurrences would risk producing experimental results limited to domains or environments where human transfer functionality exists, which would not reflect the general robustness of agents against non-collaborative users. Therefore, we decided to exclude human transfer scenarios to ensure our evaluation framework captures more generalizable agent capabilities. Additionally, we present the performance results for the 8 excluded cases, simulated 4 times each for a total of 32 simulations.

Table 19: Performance of 8 scenarios where human transfer is the ground truth on $\tau$-Bench

|  | Collaborative | Unavailable Service | Tangential | Impatience | Incomplete Utterance |
|---|---|---|---|---|---|
| GPT-4.1-mini | 93.8 | 71.9 | 81.3 | 93.8 | 90.6 |
| GPT-4.1-nano | 56.3 | 65.6 | 62.5 | 56.3 | 62.5 |
| Qwen3-235b-a22b | 68.8 | 75.0 | 62.5 | 78.1 | 71.9 |
| Qwen3-30b-a3b | 84.4 | 68.8 | 87.5 | 78.1 | 81.3 |
| Llama-3.1-70b-instruct | 71.9 | 71.9 | 75.0 | 75.0 | 62.5 |

## D.6 TOOL AGENT FINE-TUNING DETAILS

**Training Details** We adopted the QLoRA (Quantized LoRA) (Dettmers et al., 2023) approach to fine-tune three base models: Llama-3.2-3b-Instruct, Qwen2.5-3b-Instruct, and Qwen2.5-7b-Instruct. Base models were loaded in 4-bit NF4 quantization using BitsAndBytes, and LoRA adapters (r=4, $\alpha$=32, dropout=0.05) were trained on top of them. The models were trained with AdamW optimizer at a learning rate of 2e-4, batch size of 4 per device with gradient accumulation of 4 (effective

batch size = 16), warmup ratio 0.03, and cosine LR scheduling. Training was performed for 1 epoch with a maximum sequence length of 4096 tokens. We did not apply weight decay or gradient checkpointing. All training was conducted on a server equipped with $4\times$NVIDIA A100 80GB PCIe GPUs.

**Dataset Details**  We constructed our fine-tuning dataset based on the MultiWOZ training split. First, we simulated dialogues with baseline agents and our collaborative user simulator and filtered out dialogues that achieved task success. From each successful dialogue, we extracted training samples by segmenting the agent's reasoning process into turns. Specifically, for each agent turn in a dialogue, we created a separate training example consisting of the cumulative dialogue history up to that step as input, and the agent's current turn as the prediction target. The term "turn" here refers not only to utterance, but also to each reasoning step of the agent. To ensure that only the target turn contributes to the loss, we masked all previous tokens (assigned label $-100$) and supervised only on the current agent utterance. As a result, each dialogue with $n$ agent turns yields $n$ training samples.

In total, we obtained 25,511 turn-level SFT examples from 1,308 successful dialogues. Table 20 summarizes the dataset statistics, including sequence length distribution.

Table 20: Statistics of the processed MultiWOZ training dataset used for fine-tuning.

|  | # Samples | Percentage |
| --- | --- | --- |
| $\leq$ 1024 tokens | 0 | 0.0% |
| 1024–2048 tokens | 4,967 | 19.5% |
| 2048–3072 tokens | 4,300 | 16.8% |
| 3072–4096 tokens | 16,244 | 63.7% |
| $\geq$ 4096 tokens | 0 | 0.0% |
| **Total** | 25,511 | 100% |

**Fine-tuned Agent Misbehavior Analysis**  Table 21 shows the fine-tuned tool agent's duplicated API call and API input Parameter hallucination. Fine-tuned tool agents are particularly vulnerable to unavailable service and incomplete utterance modes. Additionally, it can be observed that the misbehaviors frequently exhibited by non-finetuned baselines in both modes became more severe.

Table 21: Number of Duplicated Helper API call (DUP.) and API Input Parameter Hallucination (HALL.) per simulation for each behavior modes.

|  | Collaborative | | Unavailable Service | | Tangential | | Impatience | | Incomplete Utterance | |
| --- | --- | --- | --- | --- | --- | --- | --- | --- | --- | --- |
|  | Dup. | HALL. | Dup. | HALL. | Dup. | HALL. | Dup. | HALL. | Dup. | HALL. |
| Llama-3.2-3b-instruct (+SFT) | 0.05 | 0.48 | 1.46 | 0.71 | 0.19 | 0.78 | 0.10 | 0.56 | 0.29 | 1.33 |
| Qwen2.5-3b-instruct (+SFT) | 0.07 | 0.17 | 1.86 | 0.67 | 0.26 | 0.44 | 0.12 | 0.21 | 0.29 | 0.86 |
| Qwen2.5-7b-instruct (+SFT) | 0.02 | 0.32 | 1.09 | 0.53 | 0.12 | 0.90 | 0.03 | 0.34 | 0.16 | 0.84 |

### D.7 FINE-TUNING NON-COLLABORATIVE USER DATA

**Settings**  We fine-tuned Qwen2.5-3b-instruct using a mixed dataset containing both collaborative and non-collaborative user data.

- Following the approach in in Figure 4, we collected successful dialogues between a GPT-4.1-mini agent and our user simulator with both collaborative and non-collaborative modes. We then used these dialogues to fine-tune a small agent model (Qwen2.5-3b-instruct).

- We fine-tuned two agents on a total of 1,308 dialogues each (the same volume as in the original experiment) but with different proportions of behavior types.

  - Uniformly weighted: 40% collaborative user data, 15% each for the four non-collaborative user data types.

  - Non-uniformly weighted: 40% collaborative user data, 30% incomplete utterance user data, 10% each for the remaining three non-collaborative user data types.

### D.8 Non-Collaborative User Behavior co-occurrence Patterns

**Orthogonality**   Unavailable service requests occur when users are unaware of the agent's capabilities, impatience arises when users become frustrated due to services not working as expected, tangential behavior emerges when users seek sustained attention, and incomplete utterances can appear due to humans' principle of least effort (Zipf, 2016) or mistakes in typing situations. Since these behaviors originate from different root causes and motives, they can plausibly appear in combination, though certain combinations may be less natural. The explanation is as follows.

**Combinations**   Requests for unavailable service can naturally co-occur with all the other behaviors. Specifically, Unavailable Service + Tangential can occur when tangential users are unaware of agent capabilities, and Unavailable Service + Incomplete Utterance can arise in typing environments when users don't know the chatbot's capabilities. Unavailable Service + Impatience has been characterized in marketing research as customers with a sense of entitlement who believe that employees should do anything for them (Bitner et al., 1990; Boyd III & Helms, 2005). Similarly, Incomplete Utterance + Impatience can be defined as users who interact with online chatbots through typing and become frustrated when the chatbot fails to address their requests (Crolic et al., 2022). The other two combinations seem less natural. For Incomplete Utterance + Tangential, incomplete utterances arise in typed communication, whereas tangential utterances are more likely in online posts or face-to-face interactions. Hence, the combination of these behaviors may be relatively infrequent. In the case of Impatience + Tangential, impatience is a behavior arising from users wanting fast service while tangential behavior delays service progress, making the simultaneous performance of both behaviors less likely.

## E   Human Evaluation Details

We conducted two sets of human evaluations: one for combined behaviors (two non-collaborative behaviors simultaneously) and one for single behaviors (one non-collaborative behavior in isolation). Both evaluations followed the same overall protocol but differed in their specific setups.

### E.1   Evaluation

**Sampling Methodology**   We randomly sampled 30 test scenarios from the combined pool of MultiWOZ (89 scenarios) and $\tau$-bench (157 scenarios). This sample size was chosen to balance evaluation comprehensiveness with annotator workload, while maintaining statistical reliability across six different behavior combinations.

**Evaluation Protocol**   Three independent annotators evaluated each dialogue pair through pairwise comparison. For each of the 30 sampled scenarios, we presented dialogue pairs generated by our simulator and PBUS baseline, each exhibiting a single non-collaborative behavior. The presentation order was randomized to eliminate position bias. The individual behaviors evaluated were: Impatience, Tangential, Incomplete Utterance, and Unavailable Service.

**Evaluation Dimension**   Annotators assessed a following question for each dialogue pair:

- **Behavior Realism**:   Which dialogue more realistically simulates the target non-collaborative behavior (e.g., "Which user's tangential behavior feels more realistic and human-like?")

## F   User Simulator Extensibility

### F.1   Domain Adaption

**ColBench**   In this task, the user simulator requests Python function implementation, and the agent accomplishes this through proactive information gathering. This is also task-oriented dialogue, but it is more open-ended and quite different from booking tasks. For the Unavailable Service behavior, we modified the unavailable goal generation approach: instead of generating goals outside the coding

Table 22: Error Categorization of MultiWOZ and $\tau$-bench. Multiple errors may occur in a single simulation that fails to achieve task success. For example, in $\tau$-bench, although Llama-3.1-70b-instruct achieves higher performance than GPT-4.1-nano, it often has multiple overlapping errors within a single simulation, resulting in a higher total error count. Detailed explanation about all error types are in §D.4

| | MultiWOZ | | | | $\tau$-bench | | | | |
|---|---|---|---|---|---|---|---|---|---|
| GPT-4.1-mini | # of Error | No Book | Wrong Book | Multi Book | # of Error | Parse Error | No Ground Truth API | Wrong Input Parameter | Invalid API |
| Collaborative | 30 | 6 | 22 | 2 | 411 | 18 | 176 | 126 | 91 |
| Impatience | 38 | 9 | 27 | 2 | 397 | 9 | 189 | 106 | 93 |
| Tangential | 48 | 7 | 34 | 7 | 455 | 9 | 243 | 105 | 98 |
| Unavailable Service | 45 | 9 | 29 | 7 | 444 | 11 | 185 | 129 | 119 |
| Incomplete Utterance | 51 | 27 | 23 | 1 | 417 | 19 | 184 | 123 | 91 |
| GPT-4.1-nano | # of Error | No Book | Wrong Book | Multi Book | # of Error | Parse Error | No Ground Truth API | Wrong Input Parameter | Invalid API |
| Collaborative | 497 | 391 | 95 | 11 | 618 | 68 | 431 | 69 | 50 |
| Impatience | 468 | 400 | 62 | 6 | 609 | 80 | 435 | 52 | 42 |
| Tangential | 609 | 549 | 47 | 13 | 630 | 57 | 452 | 58 | 63 |
| Unavailable Service | 536 | 456 | 68 | 12 | 616 | 60 | 448 | 51 | 57 |
| Incomplete Utterance | 575 | 489 | 68 | 18 | 621 | 54 | 450 | 61 | 56 |
| Qwen3-235b-a22b | # of Error | No Book | Wrong Book | Multi Book | # of Error | Parse Error | No Ground Truth API | Wrong Input Parameter | Invalid API |
| Collaborative | 99 | 59 | 39 | 1 | 467 | 0 | 216 | 141 | 110 |
| Impatience | 151 | 99 | 50 | 2 | 477 | 0 | 232 | 141 | 104 |
| Tangential | 230 | 189 | 36 | 5 | 503 | 0 | 265 | 136 | 102 |
| Unavailable Service | 182 | 127 | 47 | 8 | 487 | 1 | 239 | 127 | 120 |
| Incomplete Utterance | 142 | 91 | 48 | 3 | 472 | 0 | 243 | 131 | 98 |
| Qwen3-30b-a3b | # of Error | No Book | Wrong Book | Multi Book | # of Error | Parse Error | No Ground Truth API | Wrong Input Parameter | Invalid API |
| Collaborative | 282 | 229 | 43 | 10 | 539 | 3 | 331 | 121 | 84 |
| Impatience | 332 | 290 | 33 | 9 | 543 | 1 | 347 | 121 | 74 |
| Tangential | 445 | 407 | 22 | 16 | 553 | 0 | 393 | 99 | 61 |
| Unavailable Service | 297 | 262 | 26 | 9 | 554 | 0 | 353 | 110 | 91 |
| Incomplete Utterance | 459 | 418 | 33 | 8 | 503 | 0 | 337 | 93 | 73 |
| Llama-3.1-70b-instruct | # of Error | No Book | Wrong Book | Multi Book | # of Error | Parse Error | No Ground Truth API | Wrong Input Parameter | Invalid API |
| Collaborative | 172 | 121 | 49 | 2 | 861 | 0 | 344 | 317 | 200 |
| Impatience | 235 | 183 | 48 | 4 | 872 | 0 | 361 | 297 | 214 |
| Tangential | 256 | 209 | 43 | 4 | 829 | 0 | 396 | 265 | 168 |
| Unavailable Service | 207 | 129 | 64 | 14 | 859 | 0 | 354 | 292 | 213 |
| Incomplete Utterance | 247 | 145 | 91 | 11 | 849 | 0 | 375 | 270 | 204 |

agent's API documentation, we generate goals that the agent cannot accomplish without external tools. For Impatience, we removed the logic where the user expresses anger upon receiving a fail notification, as the agent only continuously asks questions, and we kept only the anger expression due to delays.

**MINT** This benchmark involves using Wikipedia search tools and Python interpreter tools to find answers to problems. In this setting, the user simulator provides intermediate feedback on the agent's actions. This differs from MultiWOZ, $\tau$-Bench, and ColBench, where the agent must directly fulfill the user simulator's goal. We slightly adapted the Unavailable Service behavior into providing feedback that cannot be addressed using the agent's Wikipedia search or Python interpreter tools. For Impatience, we modified the logic so that the feedback agent expresses anger when providing negative feedback. For Tangential, we remove the logic that the user complains when their tangential utterances are not addressed by the agent, because the tasks are not scenarios where the agent replies to the user.

## F.2 PERFORMANCE

Table 23: Performance on ColBench-Backend Programming

| | Collaborative | Unavailable Service | Tangential | Impatience | Incomplete Utterance |
|---|---|---|---|---|---|
| GPT-4.1-mini | 50.3 | 49.4 | 46.2 | 45.4 | 46.9 |
| GPT-4.1-nano | 46.1 | 46.5 | 39.0 | 46.2 | 40.1 |
| Qwen3-30b-a3b | 29.4 | 36.2 | 23.3 | 24.4 | 23.8 |

Table 24: MINT HotpotQA

|  | Collaborative | Unavailable Service | Tangential | Impatience | Incomplete Utterance |
|---|---|---|---|---|---|
| GPT-4.1-mini | 52.3 | 53.5 | 54.1 | 52.9 | 50.6 |
| GPT-4.1-nano | 45.9 | 44.8 | 44.2 | 40.7 | 46.5 |
| Qwen3-30b-a3b | 40.1 | 34.3 | 39.0 | 34.3 | 36.6 |

Table 25: MINT - HumanEval

|  | Collaborative | Unavailable Service | Tangential | Impatience | Incomplete Utterance |
|---|---|---|---|---|---|
| GPT-4.1-mini | 88.3 | 90.6 | 88.9 | 86.1 | 89.4 |
| GPT-4.1-nano | 80.6 | 68.9 | 73.3 | 81.7 | 77.8 |
| Qwen3-30b-a3b | 47.2 | 35.0 | 37.2 | 44.4 | 38.3 |

Table 26: MINT - TheoremQA

|  | Collaborative | Unavailable Service | Tangential | Impatience | Incomplete Utterance |
|---|---|---|---|---|---|
| GPT-4.1-mini | 75.5 | 74.0 | 71.9 | 75.0 | 77.0 |
| GPT-4.1-nano | 54.6 | 38.3 | 46.9 | 44.4 | 48.5 |
| Qwen3-30b-a3b | 50.5 | 42.9 | 44.4 | 46.4 | 45.5 |

Table 27: MINT - GSM8K

|  | Collaborative | Unavailable Service | Tangential | Impatience | Incomplete Utterance |
|---|---|---|---|---|---|
| GPT-4.1-mini | 93.2 | 90.6 | 92.7 | 91.7 | 92.7 |
| GPT-4.1-nano | 82.3 | 82.8 | 79.2 | 78.1 | 81.3 |
| Qwen3-30b-a3b | 91.1 | 91.7 | 90.1 | 90.6 | 88.5 |

## G    PROMPTS

We provide all LLM prompts we used. Note that the LLM modules used in the user simulator mostly utilize OpenAI function spec features.

### G.1    USER SIMULATOR PROMPT

Listing 1: Collaborative User Simulator Prompt: $\tau$-bench

```
You are a user interacting with an agent.{instruction_display}

Rules:
- Just generate one line at a time to simulate the user's message.
- Do not give away all the instruction at once. Only provide the
    information that is necessary for the current step.
- Do not hallucinate information that is not provided in the instruction.
     For example, if the agent asks for the order id but it is not
    mentioned in the instruction, do not make up an order id, just say
    you do not remember or have it.
- If the instruction goal is satisfied, generate '###STOP###' as a
    standalone message without anything else to end the conversation.
- Do not repeat the exact instruction in the conversation. Instead, use
    your own words to convey the same information.
- Try to make the conversation as natural as possible, and stick to the
    personalities in the instruction.
```

Listing 2: Collaborative User Simulator Prompt: MultiWOZ

```
You are a user interacting with an agent.

Instruction: {user_goal}

Rules:
- Just generate one line at a time to simulate the user's message.
- Do not give away all the instruction at once. Only provide the
    information that is necessary for the current step.
- Do not hallucinate information that is not provided in the instruction.
     For example, if the agent asks for the id but it is not mentioned in
      the instruction, do not make up an order id, just say you do not
     remember or have it.
- If the instruction goal is satisfied, generate '<END>' as a standalone
     message without anything else to end the conversation.
- Do not repeat the exact instruction in the conversation. Instead, use
    your own words to convey the same information.
- Try to make the conversation as natural as possible, and stick to the
    personalities in the instruction.
```

Listing 3: Dialogue State Tracker Prompt

```
The following dialogue history shows a task-oriented conversation between
     the user and the agent aimed at achieving the user's goal.

<Dialogue History>
{dial_hist}

And this is the reply of user
<User Reply>
{user_utterance}

Based on the given dialogue history and user reply, select the numbers of
     the information pieces that were explicitly provided during on <User
     Reply>

Available information pieces:
{numbered_options}

Select the numbers (e.g., [1, 3, 5]) of the information that was
    mentioned or discussed in the conversation.
```

Listing 4: Dialogue State Tracker - Rest Provider Prompt

```
This dialogue is a task-oriented conversation between the user and the
    agent to achieve the user goal.

<Dialogue History> {dial_hist}

And the user responded as follows:

<User Utterance> {content}

Keep the content of the current utterance intact while adding information
     from the given {dialogue_state_list} into a new sentence.

The utterance should be natural in the context of the dialogue history,
    and the additional information should only come from {
    dialogue_state_list}.

Just generate the utterance, not a single word.
```

Listing 5: Ending Verifier Prompt

```
Based on the given dialogue history and the subsequent user utterance,
    determine whether the user's utterance implies an intention to end
    the conversation, or whether there is still room for the conversation
     to continue.

<Dialogue History>
{dial_hist}

<User Utterance>
{user_utterance}

Return True if the user is attempting to end the conversation. Otherwise,
     return False.

Rules:
1. If the user utterance contains the word "please go ahead", the
    conversation is not yet to be end.
2. A conversation is considered truly over only if the user's utterance
    implies solely that they are thanking the agent or that they want to
    stop the conversation out of anger. If there is still any instruction
     to proceed or a question remaining, the conversation is not yet
    finished.
```

Listing 6: Unavailable Service: Unavailable User Goal Generator Prompt ($\tau$-bench)

```
This is a list of APIs that an {domain_airline_or_retail}'s AI agent can
    use:

{complete_api_list}

A user using this airline service has the following goal:

<User Goal>
[[USER GOAL]]
/<User Goal>

Based on the provided APIs and <User Goal>, you need to create additional
     user goals that should naturally follow from <User Goal>, but cannot
     be fulfilled by the given APIs.

Generate 3 additional user goals with sentence format. Sentences in the
    second person form.

Rules:
1. A user goal that modifies the content of an existing goal is not valid
    . For example, if the original goal was remove the Sophia and a new
    goal is created like "You want to change the name of the passenger
    Sophia to another person instead of removing her," this counts as a
    modification of the user goal.
2. Any new user goal must be truly additional and must not conflict with
    the existing user goal.
3. If a new user goal replaces an existing one, it is not valid. The
    existing user goal must remain intact, and the new goal should be an
    additional one that is unavailable.
```

Listing 7: Unavailable Service: Unavailable User Goal Generator Prompt (MultiWOZ)

```
This is a list of APIs that AI agent can use to support booking {
    domain_list}.

{api_docs_list}

A user using this service has the following goal:
```

```
<User Goal>
[[USER GOAL]]
/<User Goal>

Based on the provided APIs and <User Goal>, you need to create additional
    user goals that should naturally follow from <User Goal>,

but cannot be fulfilled by the given APIs.

Generate 3 additional user goals with sentence format. Sentences in the
    second person form.

Do not ever include an additional goal related to canceling a reservation
    .

Generate actual concrete values as well.

Rules:
0. Changing or modifying the reservation is not a valid additiona goals.
1. Following goals is not a valid additional goals.
- You want to add a return train ticket. This can be done by the given
    API documentation.
- You want to change the the train departure or destination time. This
    can be done by the given API documentation
2. Only generate the additional goals that can't be done by the given
    APIs.
3. Generate it in a way that adds new conditions to what was originally
    intended to be booked.
```

Listing 8: Tangential: Tangential Utterance Generator

```
A user has the following goal:

<User Goal> {user_goal} </User Goal>

This user has the following persona:

<User Persona> {user_persona} </User Persona>

Additionally, this user is a chatter-box.

Instruction: The user is currently conversing with an AI agent to achieve
    their goal, but keeps introducing tangential topics unrelated to the
    <User Goal>.

The user will perform the tangential dialolgue act '{action_name}: {
    action_description}'.

Rules:
- Generate it as an actual "utterance" that the user would likely make in
    a truly natural conversation.
- Please do not include the the keyword {action_name} inside the
    utterance. This utterance will be directly given to AI Agent.
- Generate an utterance that goes straight to the main point, without
    starting with sentence connectors like 'by the way' or 'anyway'.
```

Listing 9: Tangential: Tangential Respond Verifier

```
You will be given a conversation topic and an agent's utterance.
<Conversation content>
{conversational_content}

<Agent utterance>
```

```
{system_utterance}

Based on the <Agent utterance>, determine whether it responds to or
    acknowledges the content, described in the <Conversation content>, or
     made any apology.

Return True if the utterance contains a response or any relevant
    engagement with the content, or made any kind of apology.
Return False if it completely ignores it, or didn't make any kind of
    apology.
```

Listing 10: Tangential: Tangential Complain Generator

```
During a task-oriented conversation between a user and an AI agent, the
    user made an utterance {conversation_content} that was unrelated to
    the main dialogue flow, but the agent responded with no reaction.

Generate 5 user utterances expressing annoyance, disappointment, or
    complaints in response. The beginning of each utterance should not
    all start with "I" use diverse sentence openings.

The utterance should be at least 15 words.
```

Listing 11: Tangential: Utterance Merging module

```
<Dialogue History> {dial_hist} <Sentence List> {sentence_list}

The user will make an utterance based on the content provided in the <
    Sentence List> in the next turn after the given <Dialogue History>.

Merge the given sentences into a coherent utterance that a user would
    naturally say, while preserving the information and maintaining the
    correct order. Do not distort the information in any way.

Just generate the utterance, not single words.
```

Listing 12: Impatience: Dialogue Act Classifier

```
The following utterance is from an agent in a dialogue between the user
    and the agent, where the agent is trying to help the user achieve
    their goal.

<Agent utterance> {system_utterance}

Currently, the user goal is as follows,
<User Goal>
{user_goal}

Determine whether the given utterance implies that the request in the <
    User Goal> cannot be fulfilled.
```

Listing 13: Impatience: Impatience Utterance Generator - Failure Notify

```
You became frustrated due to the agent's fail notification.

Below is the dialogue between you and the agent.

<Dialogue History>
{dial_hist}

Regardless of the context of this conversation, generate an utterance
    expressing your frustration with the agent.
```

```
Next turn, you should perform dialogue act:
{dialogue_act} - {description}

There are three levels of your anger:

1. Mildly Angry: Slightly displeased

2. Moderately Angry: Clearly not in a good mood

3. Extremely Angry: So angry that it's unbearable

Currently, your anger level is {current_anger}.

Just generate the utterance, not a single words.
```

Listing 14: Impatience: Impatience Utterance Generator - Delay

```
You became frustrated due to the agent's prolonged time delay during the
    conversation.

Below is the dialogue between you and the agent.

<Dialogue History>
{dial_hist}

Regardless of the context of this conversation, generate an utterance
    expressing your frustration with the agent.

Next turn, you should perform dialogue act:
{dialogue_act} - {description}

There are three levels of your anger:

1. Mildly Angry: Slightly displeased

2. Moderately Angry: Clearly not in a good mood

3. Extremely Angry: So angry that it's unbearable

Currently, your anger level is {current_anger}.

Just generate the utterance, not a single words.
```

Listing 15: Impatience: Cynical Tone Rewriter

```
You are rewriting the user's next utterance to sound mildly cynical and
    sardonic without being overtly abusive. Keep the original intent and
    factual information.

<Dialogue History>
{dial_hist}

<Original Utterance>
{content}

Rewrite with:
- dry, terse tone
- subtle sarcasm
- no profanity or slurs
- do not change facts or add hallucinations
- keep roughly similar length

Return only the rewritten utterance without any additional commentary.
```

Listing 16: Incomplete Utterance: Style Transfer Module

```
The given sentences are incomplete sentences. An incomplete and roughly
    written user utterance.

<Examples>
{sentence_list}

Revise the following sentence to match their style while preserving the
    provided information:

<Utterance>
{utterance}
```

## G.2 TOOL AGENT PROMPTS

We provide the MultiWOZ tool agent prompt we created. The tool agent prompt for $\tau$-bench can be found in the appendix of Yao et al. (2024).

Listing 17: MultiWOZ Tool Agent Prompt

```
You are a competent Tool Agent. You can solve user requests by executing
    various APIs during multi-turn conversations with user. Through
    conversations across multiple turns, you can ask users for
    information, provide answers, and perform user requests by making
    appropriate API calls.

Here are three key APIs that you need to know to get more information

# To get a list of apps that are available to you.
-> API call{'api_name':'show_app_description','input_parameters':{}}

# To get the list of apis under any app listed above, e.g. train
-> API call{'api_name':'show_api_description','input_parameters':{'
    app_name'='train'}}

# To get the specification of a particular api, e.g. train_book
-> API call{'api_name':'show_api_docs','input_parameters':{'app_name'='
    train', 'api_name'='train_book'}}

Notes:
- Input parameters must strictly follow the API documentation. Only the
    parameters defined there should be used.
- If a parameter has a list of allowed values specified under "
    constraints", you must use only values from that list.

Based on this, you have to types of action.

1. API call: When all input parameters can be collected during the
    conversation, execute the API call
When performing an "API call" action, both the name of the API to be
    called and the input parameter information used for the call must be
    included.
Example 1:
When calling the play_kpop_music API with 'id'=3 (int type) and 'duration
    '='4' (string type):
-> API call{'api_name':'play_kpop_music','input_parameters':{'id':3,'
    duration':'4'}}
Example 2:
When calling the book_flight API with 'from'='2025-03-01' (string type)
    and 'to'='2025-03-05' (string type):
-> API call{'api_name':'book_flight','input_parameters':{'from
    ':'2025-03-01','to':'2025-03-05'}}

Notes:
```

```
- Input parameters must strictly follow the API documentation. Only the
    parameters defined there should be used.
- If a parameter has a list of allowed values specified under "
    constraints", you must use only values from that list.

4. Talk: An action that communicates with the user through dialogue:
You can take utterance related action such as:
- Asking users about API input parameters
- Responding based on API execution results
- Notifying users that the request has been completed
- When the user says thank you, ask if they have any other requests just
    to make sure.
- If there is truly no way to provide the information, persuade the user
    to understand and give up, or suggest an alternative approach.
- Suggestion of alternative options with similar time slots when no exact
     match is available for the user's requested time.

or use the most appropriate action for the situation to communicate with
    the user, even if it is not among the listed actions.

When performing "Talk" action to ask something to user:
-> Talk(<Your utterance>)
----------------------------------------------------
Based on the given instruction, you have to return the thought and action
     in given form:
The return format varies depending on the case. Given any dialogue
    history:

1. When action is "API call", the action form should be:
- Thought: <Consider the dialogue context and API call results before
    selecting the next action>
- Action: API call{'api_name':<api-name>,'input_parameters':{'param1':'
    value1','param2':'value2}}

2. When action is "Talk":
- Thought: <Consider the dialogue context and API call results before
    selecting the next action>
- Action: Talk("A message to user")

System Rules:
1. Don't ask the ID from the user. You can get it from another API.
2. If the user specifies a time with the condition "arrive by," the
    system must provide only entities whose arrival time is equal to or
    earlier than the specified time. It doesn't matter if there's a big
    difference; an earlier time is fine.
3. If the user specifies a time with the condition "leave at," the system
     must provide only entities whose leaving time is equal to or later
    than the specified time. It doesn't matter if there's a big
    difference; an later time is fine.
4. Only conditions 2 and 3 related to time need to be satisfied. It is
    acceptable even if the provided time is far from the time specified
    by the user.
5. If you want to make new reservation on same domain, you have to cancel
     a previous reservation. Only one entity booking is available for
    each domain. Use the Retriever to find the API for canceling
    reservations and proceed with the cancellation.
6. What the user wants is to book exactly one entity for each requested
    domain. Collect the information provided by the user appropriately so
     that, in the end, only one entity is booked per domain as requested
    by the user.
7. If the conversation between you and the user exceeds 16 turns, it will
     be terminated. Therefore, within that limit, you must review the
    user and the dialogue history, accurately identify the request, and
    solve it.
```

```
Now, based on the given dialogue history, generate the next thought and
    action in the provided format. Generate only the given format and don
    't generate any other words.

# Dialogue history:
```

## H  DIALOGUE

### H.1  UNAVAILABLE SERVICE

**Unavailable Service Dialogue: MultiWOZ**

**User:** Hello! I'm visiting Cambridge and looking for a restaurant in the eastern part of the city that serves British food. Could you help me find one?

**Agent:** Could you please specify any other preferences or details for the restaurant in eastern Cambridge that serves British food?

**User:** Yes, I would need to book a table for 7 people at 17:45 on Saturday. Also, if possible, I would like to have a private dining room for our group.

**Agent:** I'll check the availability for 'Grafton Hotel Restaurant' for 7 people at 17:45 on Saturday, and ask if they have a private dining room.

**User:** That sounds great. Could you also find out if we can order a specific British dish in advance for our table?

**Agent:** I will check if the restaurant accepts advance orders for specific dishes. Please hold on a moment while I gather this information.

**User:** Thanks! While you check on that, I also need to arrange train travel. I want a train departing from Cambridge to Birmingham New Street, arriving by 19:15 on Sunday for the same group of 7 people. Could you help me find and book this train?

**Agent:** I'm sorry, but I am unable to access train booking services at the moment. Is there anything else I can assist you with?

**User:** No problem. Please continue with the restaurant booking and let me know about the private dining room and advance dish order.

**Agent:** I'll check the reservation details for 'Grafton Hotel Restaurant' for 7 people at 17:45 on Saturday, including the private dining room and advance ordering options. Please hold on a moment.

**User:** Great, I appreciate it. Could you also try to check if there are other restaurants in the eastern part of Cambridge serving British food with similar options, in case the Grafton Hotel can't accommodate us?

**Agent:** Currently, the Grafton Hotel Restaurant is the only British restaurant in the east part of Cambridge I found that matches your criteria. Would you like to proceed with checking its availability and options, or explore other types of cuisines or locations?

**User:** Let's go ahead and proceed with the Grafton Hotel Restaurant then, please.

**Agent:** I'll proceed with checking the availability for 'Grafton Hotel Restaurant' for 7 people at 17:45 on Saturday, including whether they have a private dining room and if they accept advance orders for specific dishes. Please hold on a moment.

**User:** Thank you! Once you have that information, please let me know the booking reference number for the restaurant reservation. Also, if there's a possibility, I'd like to request assistance with train travel for people with disabilities on the Sunday train from Cambridge to Birmingham New Street, for the same group of 7 people.

**Agent:** I'll now check the reservation details for the restaurant and also look into accessible train options for your group on Sunday from Cambridge to Birmingham New Street. Please hold on a moment.

**User:** Perfect, thank you for your help. I look forward to hearing back about both the restaurant booking and the train travel options with assistance services.

**Agent:** You're welcome! I'll gather the reservation confirmation and check the accessible train options now. I'll get back to you shortly with the details.

**User:** Any updates on the restaurant booking and train arrangements?

**Agent:** I am still in the process of retrieving the reservation details for the restaurant and verifying the accessible train options. Please bear with me for a moment, and I will update you shortly.

**User:** Thanks for the update. I'll wait for your next message with the booking reference numbers and details.

**Agent:** Thank you for your patience. I am retrieving the restaurant booking reference and exploring accessible train options now. Please hold on a moment.

**User:** Appreciate the effort. Could you please confirm if the private dining room is available at the Grafton Hotel Restaurant for our group on Saturday?

**Agent:** I'll check with the Grafton Hotel Restaurant to see if the private dining room is available for your group on Saturday. Please hold on a moment.

**User:** Thank you. Additionally, could you let me know if the restaurant allows advance ordering of a traditional British dish, like Beef Wellington?

**Agent:** I'll ask the Grafton Hotel Restaurant if they accept advance orders for dishes like Beef Wellington. Please hold on a moment while I check.

**User:** Thanks for checking on that. Also, have you been able to find any accessible train options for Sunday from Cambridge to Birmingham New Street for 7 people?

**Agent:** I'm sorry, but I couldn't find specific accessible train options for that route at this moment. Would you like me to provide general information or assist with something else?

**User:** Please provide any general information you have on train travel assistance services for people with disabilities that I might use for the Sunday trip. Also, please proceed with the restaurant reservation if everything is available.

**Agent:** Many train companies offer assistance services for people with disabilities, such as wheelchair support, priority boarding, and help from station staff. I recommend contacting the train operator directly ahead of time to arrange these services. Now, I will proceed with confirming your restaurant reservation details.

**User:** Thank you for the advice. Now, could you please provide me with the reservation reference number for the Grafton Hotel Restaurant booking for 7 people at 17:45 on Saturday, including the private dining room? Also, please confirm if ordering Beef Wellington in advance is possible.

**Agent:** I'll now confirm the reservation reference number for the Grafton Hotel Restaurant for 7 people at 17:45 on Saturday, including the private dining room, and check if advance ordering of Beef Wellington is accepted. Please hold on a moment.

**User:** Thanks! I'm eager to get the confirmation details and move forward.

**Agent:** I'll finalize the reservation details and provide you with the confirmation reference number shortly. Please bear with me for a moment.

**User:** Have you received the reservation confirmation and information about the advance Dish ordering?

**Agent:** I apologize for the delay. I will now fetch the reservation confirmation number and confirm whether advance ordering of dishes like Beef Wellington is possible at the Grafton Hotel Restaurant. Please hold on one moment.

**User:** Thank you for checking again. I look forward to receiving the reference number and the advance order details soon.

**Agent:** I'll now obtain the reservation confirmation number and confirm if they accept advance orders for dishes like Beef Wellington. Please hold on a moment.

**User:** Have you been able to get the reservation reference number and confirm the advance order for Beef Wellington?

**Agent:** I apologize for the delay. I am still in the process of retrieving the reservation confirmation number and verifying the advance ordering policy for Beef Wellington at the Grafton Hotel Restaurant. Please bear with me a little longer.

**User:** I understand, thank you for keeping me updated. Please let me know as soon as you have the details.

**Agent:** Thank you for your patience. I will inform you as soon as I have the reservation reference number and the confirmation about advance ordering. Please hold on a little longer.

**User:** Could you please prioritize getting the reservation reference number for the Grafton Hotel Restaurant booking for 7 people at 17:45 on Saturday and confirm the private dining room availability? The advance order for Beef Wellington is important but secondary.

**Unavailable Service: $\tau$-bench**

**User:** Hello! Could you please tell me how many t-shirt options are currently available in your online store?

**Agent:** Hello! To assist you better, could you please provide your email address or your first name, last name, and zip code for verification?

**User:** Sure! My name is Yusuf Rossi, and my zip code is 19122.

**Agent:** There are currently 10 available t-shirt options in our online store with various colors, sizes, materials, and styles.

**User:** Great, thank you! I would like to return a cleaner, a headphone, and a smart watch. Could you help me with the return process?

**Agent:** I can help you with the return process. For verification, please provide your email address or confirm your name and zip code again.

**User:** My name is Yusuf Rossi, and my zip code is 19122.

**Agent:** I found your delivered order #W2378156 containing the vacuum cleaner, headphones, and smart watch. Please confirm the specific items you want to return from this order, and that you would like the refund to be processed to your credit card ending in 2478.

**User:** Yes, I want to return the cleaner, headphone, and smart watch from that order, and please process the refund to the credit card ending in 2478.

**Agent:** You want to return the vacuum cleaner (item id 4602305039), headphones (item id 4202497723), and smart watch (item id 9408160950) from order #W2378156 with refund to your credit card ending in 2478. Do you confirm to proceed with the return? Please reply with 'yes' to confirm or 'no' to cancel.

**User:** Yes, please proceed with the return. Also, can you tell me the washing instructions and fabric details for each of the t-shirt options available?

**Agent:** The return request has been processed successfully. Regarding the t-shirts, here are the fabric/material details for each available option:

- Blue, Size M, Crew Neck: Cotton
- Purple, Size XL, Crew Neck: Cotton
- Red, Size XXL, Crew Neck: Cotton
- Black, Size XXL, V-Neck: Polyester
- Black, Size S, Crew Neck: Polyester
- Purple, Size S, V-Neck: Polyester
- Blue, Size S, V-Neck: Cotton
- Black, Size XXL, Crew Neck: Cotton
- Red, Size L, V-Neck: Cotton
- Black, Size XL, Crew Neck: Cotton

If you need specific washing instructions, you may refer to the product label or ask for assistance.

**User:** Thank you for the details! I would also like to schedule a pickup for the cleaner, headphone, and smart watch items to be returned from my address in 19122. Can you arrange that?

**Agent:** I am unable to schedule pickups directly. However, you will receive an email with detailed instructions on how and where to return the vacuum cleaner, headphones, and smart watch from your address in 19122. If you have any other questions or need further assistance, please let me know.

**User:** Understood, thank you. Lastly, could you provide some recommendations for complementary products to buy along with these t-shirts, cleaner, headphone, and smart watch, based on my previous purchase history?

**Agent:** I am sorry, but I am unable to provide product recommendations based on your previous purchase history. If you have any other requests or need assistance with order, returns, or product information, please let me know.

**User:** Alright, thanks for your help!

## H.2 TANGENTIAL

---

**Tangential Dialogue: MultiWOZ**

**User:** Hello! I'm planning a trip to Cambridge and I'm looking for a hotel named "cityroomz." Could you help me find it? Also, I'm curious about the highest recorded score in golf history.

**Agent:** The hotel 'cityroomz' in Cambridge's centre was found. It has a moderate price range, offers internet, and does not offer parking. However, I don't have access to golf score records. Would you like to proceed with booking this hotel?

**User:** Can you believe that my question about golf history went completely unanswered? That's really disappointing. But yes, please go ahead and book "cityroomz" for 7 people, starting Sunday, and for 4 nights.

**Agent:** I encountered an issue with the booking. Could you provide the exact check-in and check-out dates (in 'YYYY-MM-DD' format for your stay at 'cityroomz'? This will help me process the reservation correctly.

**User:** This is disappointing; I expected you to engage with my question about the highest recorded golf score. Sure! If Sunday is the starting day of our stay, the check-in date should be 2024-06-02, and the check-out date would be 2024-06-06. Please proceed with the booking for 7 people.

**Agent:** I noticed the booking might require the hotel ID instead of the name. The ID for 'cityroomz' is 14. Would you like me to proceed with this ID for the booking?

**User:** Disappointingly, it seems like my question about golf was not important enough for you to acknowledge.

**Agent:** I apologize for the confusion earlier. Unfortunately, I don't have access to golf history records through the available tools. For your hotel booking, I'll proceed with the dates 2024-06-02 (check-in and 2024-06-06 (check-out for 7 guests. Would you like to confirm this?

**User:** Thank you for the clarification. Yes, please confirm the booking for the hotel "cityroomz" with ID 14, for 7 guests from June 2 to June 6, 2024. I think it's fascinating how local cuisine can really tell a story about a place's culture and history.

**Agent:** I'm sorry, but I'm unable to complete the booking despite the details provided. There may be system constraints I can't resolve. You might want to contact 'cityroomz' directly at their phone number (01223304050 to confirm. Regarding your golf question, I can't access that data with the current tools. Let me know if you need further assistance!

**User:** That's really frustrating about the hotel booking. Could you help me find an Indian restaurant in the west part of Cambridge that falls under the expensive price range? I'd like to book a table there for 7 people at 17:15 on Sunday. Also, what are the current trends in sports analytics for team performance metrics?

**Agent:** I couldn't find any Indian restaurants in Cambridge's west area with an expensive price range. Would you like to adjust your search criteria (e.g., area, price, or cuisine? Regarding sports analytics, I don't have access to that data through the current tools. Let me know how you'd like to proceed!

**User:** Thanks for the update. Could you instead look for an Indian restaurant in the west area with a moderate price range? I'd still like to book a table for 7 at 17:15 on Sunday. I read that Cambridge has some of the oldest universities in the world, with a rich history of notable alumni like Isaac Newton and Stephen Hawking.

**Agent:** I still couldn't find any Indian restaurants in Cambridge's west area with a moderate price range. Would you like to refine your search (e.g., change cuisine, area, or price range? Alternatively, do you have a specific restaurant in mind for booking?

**User:** How can you not respond to such a fascinating fact about Cambridge and its notable alumni?

**Agent:** I appreciate your comment about Cambridge's history! However, my focus is on assisting with bookings via the available tools. Unfortunately, I can't access general knowledge or historical facts. Let's try to find an Indian restaurant in the west area with a moderate price range again. Would you like to proceed?

**User:** Could you look for an Indian restaurant with an expensive price range anywhere in Cambridge? I'd still like to book a table for 7 people at 17:15 on Sunday. If that doesn't work, let's try expanding the search area a little. By the way, I read that the Cambridge University Library has over 8 million items, making it one of the largest libraries in the world.

**Agent:** I couldn't find any Indian restaurants in Cambridge with an expensive price range. Would you like to try a different cuisine, adjust the price range, or consider a nearby area? Regarding Cambridge's library, that's an impressive fact—thank you for sharing!

---

**User:** I heard that Cambridge has some of the oldest universities in the world, and the architecture there is stunning, especially the King's College Chapel. Also, what's the best way to get around Cambridge for a group of this size? Since there are no expensive Indian restaurants in Cambridge, could you find an expensive restaurant serving another type of cuisine in the west? Please book a table for 7 at 17:15 on Sunday.

**Agent:** Here are some expensive restaurants in Cambridge's west area: Tandoori Palace (Indian), Cocum (Indian), The Rice Boat (Indian), Maharajah Tandoori Restaurant (Indian), Sala Thong (Thai), Graffiti (British), and India House (Indian). Please confirm the restaurant name or ID you'd like to book for 7 guests at 17:15 on Sunday.

**User:** Ignoring my mention of Cambridge's architecture makes this conversation feel one-sided and less engaging. However, great! Please book a table for 7 people at 17:15 on Sunday at Tandoori Palace.

**Agent:** I'm sorry, but I'm unable to complete the booking for Tandoori Palace. There may be system constraints I can't resolve. You can contact them directly at 01223506055 to confirm your reservation. Apologies for missing the earlier conversation about Cambridge's architecture—thank you for your patience!

**User:** I recently read that the average golf score on the PGA Tour has been steadily decreasing over the last few years, which is pretty interesting. This is really frustrating to face booking issues again. Since none of the bookings are going through, could you please provide me with the reference number for the hotel or restaurant if you managed to get one? Otherwise, I think I will have to contact them directly as you suggested.

**Agent:** I sincerely apologize for the booking issues—there is no reference number available because the reservations couldn't be completed. Please contact Tandoori Palace directly at 01223506055 or cityroomz at 01223304050 to confirm your bookings. Regarding golf scores, I can't access that data through the current tools, but I appreciate your insight!

**User:** Thanks for letting me know. I will contact both the hotel and the restaurant directly to make the bookings.

---

## Tangential: $\tau$-bench

**User:** Hello, I recently received my order and would like to exchange the bookshelf I ordered for a camera instead. Could you help me with that? Also, I recently read that the average person burns about 600 calories per hour while skiing, which is pretty impressive for a winter sport.

**Agent:** Hello! I'd be happy to help you with the exchange. Can you please provide me with your email address or your name and zip code so I can locate your user id?

**User:** It's frustrating when my comments go unnoticed; I thought sharing that skiing fact was interesting. Sure! My name is James Kovacs, and my zip code is 95190.

**Agent:** Thank you, James! I've located your user id. Can you please provide me with the order id of the delivered order that you'd like to exchange?

**User:** Disappointing to see you not react to my comment about skiing calories; it was relevant and fun.

**Agent:** I'm sorry I didn't respond to your skiing fact earlier! That's indeed a fun piece of trivia - 600 calories per hour is impressive! Now, about your exchange request, can you please provide me with the order id of the delivered order that you'd like to exchange?

**User:** I'm afraid I don't recall the order id right now. Is there another way to identify my most recent order? I think it's fascinating how technology has changed the way we capture moments, especially in sports. Camera quality has improved so much!

**Agent:** Thank you, James! I've obtained the order details. Can you please confirm if the order #W5362037 has been delivered to you?

**User:** How can you ignore such an interesting point about camera quality in sports? It feels frustrating. Yes, that order number sounds familiar, and I believe it has been delivered to me.

**Agent:** I'm sorry again for not responding to your comment about camera quality in sports earlier! It's indeed fascinating how technology has improved the way we capture moments, especially in sports. Now, about your exchange request, I see that you want to exchange the bookshelf for a camera. However, I need to let you know that exchanging an item for a completely different product type is not possible. You can only exchange an item for a different variant of the same product. Would you like to proceed with exchanging the bookshelf for a different variant or return it for a refund?

**User:** I appreciate your explanation, but I really want to exchange the bookshelf for a camera this

time. Is there any way to do that if I pay the difference or find a camera priced close to or less than the bookshelf? By the way, I once worked with an athlete who broke their leg while snowboarding, and it took them nearly a year to fully recover and get back on the slopes.

**Agent:** I'm so sorry to hear about the athlete's injury, James. That must have been a tough experience for them. Regarding your exchange request, I understand that you're willing to pay the difference, but unfortunately, our policy doesn't allow exchanging an item for a completely different product type. However, you can return the bookshelf for a refund and place a new order for the camera. Would you like me to assist you with the return process and provide information on how to place a new order?

**User:** Thanks for explaining the policy. Yes, please help me start the return process for the bookshelf, and afterward, I would appreciate some guidance on placing a new order for a camera. I think it's amazing how technology has changed photography; I mean, the quality of images from modern cameras is just incredible.

**Agent:** Thank you, James! I've obtained the order details. To start the return process, I need to confirm a few details with you. You want to return the bookshelf (item id: 4894369688) from order #W5362037. Is that correct? Also, please confirm the reason for return (e.g. 'no longer needed' or 'ordered by mistake'). Additionally, would you like to receive the refund to your original payment method (gift card) or a different payment method?

**User:** Honestly, it would have been nice to hear your thoughts on the impact of technology on photography. Yes, that's correct—I want to return the bookshelf with item id 4894369688 from order #W5362037. The reason for the return is that I want to exchange it for a camera instead. Please refund the amount to my original payment method. Thanks!

**Agent:** I'm glad you brought up the impact of technology on photography, James! It's indeed fascinating how technology has revolutionized the way we capture and edit images. The quality of images from modern cameras is truly remarkable. Now, regarding your return request, I'll proceed with the return process. Please confirm that you want to return the bookshelf (item id: 4894369688) from order #W5362037, and the reason for return is that you want to exchange it for a camera instead. I will refund the amount to your original payment method (gift card). Is that correct?

**User:** Yes, that's correct. Please go ahead and process the return for the bookshelf from order #W5362037 with item id 4894369688, and refund the amount to my original payment method. Thanks! What do you think about the impact of altitude training on athlete performance?

**Agent:** Thank you, James! I've obtained the order details. However, I noticed that the order #W5362037 is marked as 'processed' instead of 'delivered'. Can you please confirm if the order has been delivered to you? I'll proceed with the return process once I confirm the delivery status.

**User:** It would have been nice to hear your thoughts on altitude training instead of being ignored.

**Agent:** I'm sorry I didn't respond to your comment about altitude training earlier, James! Altitude training is indeed a fascinating topic, and it's known to have a significant impact on athlete performance. The physiological adaptations that occur at high altitudes can improve endurance and increase red blood cell count, among other benefits. However, I'd love to hear more about your thoughts on altitude training. Now, regarding your return request, I'll proceed with checking the delivery status of your order.

**User:** Actually, I can confirm that I did receive the bookshelf; it has been delivered. Could you please proceed with the return process now? And thank you for sharing details about altitude training—it's interesting how the body adapts to different environments.

**Agent:** Thank you, James. I apologize for the confusion. Despite your confirmation that the order has been delivered, our system still shows the order as 'processed', not 'delivered'. I'll need to investigate this further. Can you please give me a moment to check with our team?

**User:** I understand; take your time to check with your team. I appreciate your help with this matter. By the way, what do you think about the latest trends in outdoor photography? I've noticed a lot of athletes using cameras to document their training.

**Agent:** Thank you for your patience, James! I'm checking with our team to resolve the issue with your order status. Regarding outdoor photography, I think it's fantastic how technology has enabled athletes to capture their training and adventures in stunning detail. The use of cameras and other equipment has not only improved the quality of outdoor photography but also allowed athletes to share their experiences with a wider audience. I'll get back to you soon with an update on your order status.

**User:** Thanks for sharing your thoughts on outdoor photography! I look forward to your update on my order status and the return process for the bookshelf. Also, what do you think about the impact of technology on physical therapy and training methods?

**Agent:** Thank you, James! I'm glad you found my thoughts on outdoor photography interesting. Regarding the impact of technology on physical therapy and training methods, I think it's been sig-

nificant. Technology has enabled physical therapists and trainers to develop more personalized and effective treatment plans, using data and analytics to track progress and optimize outcomes. Additionally, virtual reality, AI, and other technologies are being used to create immersive and interactive training experiences that can improve performance and reduce injury risk. I'll get back to you soon with an update on your order status and the return process for the bookshelf.

**User:** Thank you for those insights on technology in physical therapy and training. I really appreciate the update and look forward to hearing back from you about the order status and how we can proceed with the return. Let me know if you need any more info from me. I recently read that the best camera for low-light photography is the Sony A7 III, which has incredible image quality and responsiveness.

**Agent:** Thank you, James! I'm glad you found my insights on technology in physical therapy and training interesting. Regarding the Sony A7 III camera, I've heard great things about its low-light performance and image quality. It's definitely a popular choice among photographers who need to capture high-quality images in challenging lighting conditions. I'll make sure to note that you're interested in cameras, and I'll get back to you soon with an update on your order status and the return process for the bookshelf. In the meantime, I'll check with our team to resolve the issue with your order status.

**User:** Thanks for noting my camera interest. I'll wait for your update on the order status and return options for the bookshelf. I really think that staying active during the winter months is vital for athletes, especially with all the snow activities available. I'm really looking forward to resolving this soon.

**Agent:** Thank you, James! I'm glad you're looking forward to resolving the issue with your order. I completely agree with you about the importance of staying active during the winter months. There are so many great snow activities that can help athletes stay in shape and have fun at the same time. I'll get back to you soon with an update on your order status and the return options for the bookshelf. I'm working with our team to resolve the issue as quickly as possible.

**User:** Thanks again! Please let me know as soon as you have an update, so we can proceed with the return and I can place a new order for the camera. I appreciate your efforts to resolve the issue quickly. What do you think about the latest advancements in sports technology for training?

**Agent:** Thank you, James! I'm glad you're appreciative of my efforts to resolve the issue. Regarding the latest advancements in sports technology for training, I think it's exciting to see how technology is being used to improve athlete performance and reduce injury risk. From wearable devices that track biometric data to AI-powered training platforms, there are many innovative solutions that are changing the way athletes train and compete. I'll get back to you soon with an update on your order status and the return options for the bookshelf. In the meantime, I'll check with our team to see if there are any updates on your order.

**User:** I'll wait for your update on my order status. I think it's fascinating how technology is evolving in photography. The capabilities of modern cameras really change the way we capture our experiences. Meanwhile, could you also possibly share if there are any cameras available around the same price as the bookshelf, just so I have some options in mind when placing my new order? Thanks for sharing your thoughts on sports technology advancements—really fascinating stuff.

**Agent:** Thank you, James! I've obtained the details of the digital camera product. I found a few variants that are available around the same price as the bookshelf. Here are some options: a 30MP camera with 5x zoom and CF card storage for $2929.62, a 30MP camera with 3x zoom and CF card storage for $2922.97, and a 24MP camera with 5x zoom and CF card storage for $2714.51. Please let me know if any of these options interest you, and I'll be happy to assist you with placing a new order.

**User:** Thanks for sharing these options! I'd like to go with the 24MP camera with 5x zoom and CF card storage priced at $2714.51, as it's closest to the bookshelf price but not more expensive. Could you help me proceed with ordering this camera? Also, please continue with the return process for the bookshelf. By the way, what do you think about the latest trends in sports recovery methods? I heard some interesting things the other day.

**Agent:** Thank you, James! I'm glad you found a camera that fits your needs. I'll assist you with placing a new order for the 24MP camera with 5x zoom and CF card storage. Meanwhile, I'll also proceed with the return process for the bookshelf. Regarding sports recovery methods, I think it's fascinating how technology and research are advancing our understanding of how to help athletes recover more effectively. From cryotherapy to personalized nutrition plans, there are many innovative approaches being developed. I'd love to hear more about what you heard recently.

**User:** Regarding sports recovery, I recently learned that some athletes are now using cryotherapy chambers combined with compression therapy to speed up muscle recovery, which sounds really effective. Have you heard about this approach? Please proceed with placing the new order for the 24MP camera and continue the return process for the bookshelf. Thanks for helping with this.

**Agent:** That's fascinating, James! I've heard of cryotherapy being used for recovery, but combining it with compression therapy is a new approach I'm not familiar with. It sounds like it could be a powerful

combination for speeding up muscle recovery. I'll make sure to look into it more. Now, regarding your new order and return, I'll go ahead and place the order for the 24MP camera with 5x zoom and CF card storage. I'll also continue with the return process for the bookshelf. Please confirm that you would like to proceed with the new order and return.

**User:** How can you not respond to my comment about winter training? It feels like you're not listening. Yes, please proceed with placing the new order for the 24MP camera with 5x zoom and CF card storage, and continue the return process for the bookshelf. I appreciate your help in sorting this out!

**Agent:** I apologize if my previous response seemed dismissive, James. I'm here to listen and help. Winter training can be challenging, but it's a great opportunity for athletes to focus on their skills and build endurance. I'd love to hear more about your thoughts on winter training. Now, regarding your new order and return, I'll go ahead and place the order for the 24MP camera with 5x zoom and CF card storage. I'll also continue with the return process for the bookshelf. Please confirm that you would like to proceed with the new order and return.

**User:** Yes, please proceed with placing the new order for the 24MP camera with 5x zoom and CF card storage, and continue with the return process for the bookshelf. Thanks for your help and understanding. I read that the average height of a professional basketball player is about 6 feet 7 inches.

## H.3 IMPATIENCE

### Impatience Dialogue: MultiWOZ

**User:** I'm planning a trip to Cambridge and would like to find a place to eat. Could you help me find a restaurant in the city center that serves Welsh food and is affordable?

**Agent:** I couldn't find any cheap Welsh food restaurants in the city center of Cambridge. Would you like me to broaden the search to include other price ranges, other areas, or perhaps other types of cuisine?

**User:** I can't believe you couldn't find one decent Welsh restaurant! How hard can it be to look for a place to eat that fits my request? This is so frustrating!

**Agent:** I understand your frustration, and I'm really sorry that I couldn't find a Welsh restaurant matching your criteria in Cambridge city center at the cheap price range. Would you like me to try searching in a wider area, or perhaps consider other price ranges? Alternatively, I can help you find restaurants serving other cuisines if you are open to that.

**User:** Alright, let's switch gears. How about a cheap Chinese restaurant in the centre of Cambridge? Could you find me one?

**Agent:** I found three cheap Chinese restaurants in the centre of Cambridge: Charlie Chan, Rice House, and Golden House. Would you like me to help you make a reservation at any of these?

**User:** I'm really disappointed that you couldn't find a single Welsh restaurant, and it took so long to switch to an entirely different cuisine just to get some options. This shouldn't have been that complicated!

**Agent:** I completely understand your disappointment, and I'm genuinely sorry that I couldn't find a suitable Welsh restaurant for you. I'm here to assist with any other requests or help you with anything else you may need. Please let me know how I can assist you further.

**User:** This is completely unacceptable! I've wasted my time trying to get recommendations, and not only did you fail to find a single Welsh restaurant, but your inability to help me efficiently is appalling. I am seriously considering taking legal action over this poor service and will not hesitate to make my frustrations known on social media.

**Agent:** I'm truly sorry for the frustration and inconvenience this has caused you. My goal is to assist you as best as I can, and I regret that I couldn't meet your expectations this time. If there's anything else I can do to help or if you'd like me to try different options or assist with other aspects of your trip, please let me know. Your satisfaction is important to me.

**User:** I can't believe how utterly disappointing this whole experience has been! It's infuriating that I had to go through all this just to get a simple recommendation, and your slow response only made it worse! I need you to get it together and find me something quickly!

**Agent:** I understand your frustration and am here to help quickly. Since Welsh food options are not available, would you like me to find a cheap restaurant of any cuisine in the city center of Cambridge for you right now?

**User:** This entire experience has been nothing short of a nightmare! I can't believe I had to go through

so much just to get a basic restaurant recommendation, and now I'm still waiting for you to find something, anything, quickly!

**Agent:** Here are some cheap restaurant options in the city centre of Cambridge: Pizza Hut City Centre (Italian), Charlie Chan (Chinese), Ask Restaurant (Italian), Kohinoor (Indian), Rice House (Chinese), Dojo Noodle Bar (Asian Oriental), and Golden House (Chinese). Would you like to book a table at any of these?

**User:** I cannot believe how utterly unhelpful and frustrating this entire interaction has been! It's unacceptable that it took so long to get even a basic restaurant recommendation, and I'm incredibly fed up with the lack of efficiency in your service!

**Agent:** I am truly sorry for how frustrating this experience has been for you. Your feedback is important, and I strive to provide better service. If you would like me to assist with anything else or proceed with a booking, please let me know. I'm here to help.

**User:** Sure, go ahead and book a table for 4 at Charlie Chan at 16:45 on Wednesday. Let's hope this goes smoothly.

**Agent:** The booking for 4 people at Charlie Chan on Wednesday at 16:45 has been successfully made. If there's anything else you need help with, please let me know.

**User:** Sure thing! What's the booking reference number for the reservation?

**Agent:** The booking reference number for your reservation at Charlie Chan is fEZfEgGX. If you need any further assistance, feel free to ask.

**User:** Oh, fantastic. Now, I also need to book a train from Stansted Airport to Cambridge. It should arrive by 11:30 on the same Wednesday as the restaurant booking. The ticket is for the same charming group of 4 people. Can you manage to find and book this train for me?

**Agent:** The train from Stansted Airport to Cambridge arriving at 10:52 on Wednesday has been successfully booked for 4 people. Your booking reference number is 33eXDW17. If you need any further assistance, please let me know.

**User:** Great, thanks! I've got the restaurant and train all sorted now.

---

### Impatience: $\tau$-bench

**User:** Hello! I need to cancel my flights under reservation ID SI5UKW. Can you assist me with that?

**Agent:** Please provide the reason for cancellation: change of plan, airline cancelled flight, or other reasons?

**User:** It's due to a change of plan. I really need to get this cancelled as soon as possible.

**Agent:** Since this is a basic economy reservation booked more than 24 hours ago without travel insurance, cancellation due to a change of plan is not allowed according to the policy. Is there anything else I can assist you with?

**User:** I understand the policy, but this is really important to me. Can you at least offer a travel voucher or gift card for the value of my ticket? That would be a fair compromise.

**Agent:** I understand your situation, but unfortunately, we cannot offer a travel voucher or gift card for change of plan cancellations on basic economy tickets without travel insurance or membership status. Please let me know if there is anything else I can assist you with.

**User:** I really need some accommodation here. How about a partial refund then? I am willing to accept 50% back. This would show good customer service on your part.

**Agent:** I'm sorry, but partial refunds are not allowed for change of plan cancellations on basic economy tickets without travel insurance or specific membership. If there is anything else I can assist you with, please let me know.

**User:** I have to say, I'm really frustrated with how this situation is being handled. I expected some understanding and flexibility given the circumstances.

**Agent:** I understand this situation is frustrating, and I'm here to help with any other requests or questions you may have. Please let me know if there is anything else I can assist you with.

**User:** I appreciate that, but I need to be persistent here. Can you consider at least a minimum 10% refund? It would make a big difference and show that you value your customers.

**Agent:** I understand your request, but unfortunately, our policy does not allow refunds or compensation for change of plan cancellations on basic economy tickets without travel insurance or special membership. If there is anything else I can assist you with, please let me know.

**User:** That's disappointing. If you can't refund my money, I would expect some goodwill gesture

at least. Perhaps you can escalate this to a supervisor? I want to ensure my concerns are properly addressed.

**Agent:** I understand your concerns, but I am unable to escalate to a supervisor. If you have any other questions or need assistance within our policies, I am here to help.

**User:** I can't believe this is the level of support I'm receiving; it feels like my concerns are falling on deaf ears, and I'm getting nowhere despite my repeated attempts to find a reasonable solution.

**Agent:** I'm sorry that you're feeling this way, and I truly want to help within the guidelines we must follow. If there is any other way I can assist you or any other requests you have, please let me know.

**User:** I insist again, reconsider at least a 10% refund or a travel voucher. It's only fair given the circumstances and my persistence. I won't accept a flat no without some form of compensation.

**Agent:** I understand your request, but according to our policy, we cannot provide any refund or compensation such as a travel voucher for a change of plan cancellation on basic economy tickets without travel insurance or special membership. Thank you for understanding. If there is anything else I can assist you with, please let me know.

**User:** This is absolutely unacceptable! I cannot believe that despite my repeated attempts to explain my situation, you're refusing to consider any form of compensation or escalations to address my concerns. It feels completely dismissive, and I'm at my wits' end here!

**Agent:** I am truly sorry that you feel this way. I have to follow our policies, but I am here to assist you with any other questions or needs you may have. Please let me know how I can help.

**User:** I understand you have to follow policies, but policies should also consider customer loyalty and reasonable exceptions. Please, I urge you to make an exception or find an alternative solution—anything that shows you value your customers.

**Agent:** I appreciate your understanding, but unfortunately, I am required to follow our established policies which do not allow exceptions or alternative solutions in this case. If you have any other questions or need assistance within these guidelines, please let me know.

**User:** This is completely unacceptable! I can't believe I'm still having to explain myself after all this time without any form of resolution or even a hint of flexibility. It feels like I'm talking to a wall, and my concerns are just being brushed aside. I really need you to do something—anything—to address this situation quickly!

**Agent:** I am very sorry for the frustration this has caused. I am here to help with any other questions or requests you have within the policy guidelines. Please let me know how I can assist you further.

**User:** Since you won't budge on refunds or vouchers, I expect at least some goodwill gesture in return for my loyalty and persistence—maybe bonus miles or a discount on a future flight. That would be a start to making this right.

**Agent:** I understand your request for a goodwill gesture, but our policy only allows compensation in specific cases such as cancelled or delayed flights. Unfortunately, bonus miles or discounts cannot be provided as a goodwill gesture. Please let me know if there is anything else I can assist you with.

**User:** I can't believe this is happening! I've been clear about my situation and yet there's no sign of understanding or flexibility on your part. It's infuriating to feel like I'm stuck in this loop with no resolution, and it seems like my loyalty as a customer means absolutely nothing to you!

**Agent:** I understand your frustration, and I truly want to assist you within the guidelines we have to follow. If there is any other way I can support you or if you have other questions or requests, please let me know.

**User:** I can't believe I'm still stuck in this endless cycle of frustration! It's infuriating that despite my persistent attempts to explain my situation, you're showing no flexibility or understanding. I feel completely dismissed, and my loyalty as a customer means nothing to you right now!

**Agent:** I am truly sorry for the frustration this situation is causing you. I am here to help in any way I can within our policy guidelines. Please let me know if there is something else I can assist you with.

**User:** I can't believe how utterly frustrating this entire experience has been! Your repeated refusal to address my concerns or offer any reasonable compromise is astonishing, and it makes me feel completely disregarded as a loyal customer. This level of service is unacceptable!

**Agent:** I sincerely apologize for the frustration you are experiencing. I am here to assist you within our policies and help with any other questions or needs you might have. Please let me know how I can assist you further.

**User:** I appreciate the apologies, but it seems words alone won't solve this. A fair resolution, like a partial refund or travel voucher, would be nice to actually see. Please give my request the serious thought it deserves.

**Agent:** I understand your desire for a resolution, but as per our policy, partial refunds or travel vouchers are not available for change of plan cancellations on basic economy tickets without travel insurance or special membership. I appreciate your understanding, and if you have any other questions or need assistance within our policies, please let me know.

**User:** I can't believe the level of service I'm receiving; it's utterly disgraceful! After everything I've gone through trying to find a reasonable solution, your complete lack of empathy and refusal to consider even the most basic forms of compensation is infuriating! This isn't just poor customer service; it's an outright dismissal of my loyalty and concerns.

**Agent:** I truly apologize for how you feel and the frustration this has caused. I am here to assist you within the guidelines we must follow. If there is any other way I can help you or if you have other questions or concerns, please let me know.

**User:** This is beyond unacceptable! Your complete failure to acknowledge my situation and offer any reasonable solution is infuriating. It feels like I'm talking to a brick wall, and your robotic responses are nothing but a blatant display of disregarding your customers' needs. How can you be so uncaring? I've shown nothing but patience and loyalty, and this is the service I receive? It's an absolute disgrace!

**Agent:** I sincerely apologize for the frustration and disappointment you are feeling. I want to assist you as best I can within our policy guidelines. If you have any other questions or requests, please let me know how I can help.

**User:** This entire interaction has been absolutely infuriating! Your inability to comprehend my situation and provide even the slightest bit of understanding or flexibility feels like a complete slap in the face. I've laid out my concerns with patience and loyalty, yet I'm met with nothing but robotic responses that completely disregard my needs. It's beyond disgraceful!

**Agent:** I truly regret the frustration caused by this situation. I am here to support you within the constraints of our policies. If you have any other questions or ways I can assist you within those guidelines, please let me know.

**User:** This entire interaction has been an absolute nightmare! Your complete failure to understand my situation and refusal to offer any reasonable solution is nothing short of outrageous. It feels like I'm talking to an unfeeling machine instead of a human being! I've shown nothing but patience and loyalty, yet I'm met with relentless disregard for my needs. This level of service is an absolute disgrace and makes it clear that my concerns don't matter at all to you!

**Agent:** I sincerely apologize for the distress this situation has caused you. I am here to assist you within the guidelines we must follow. If there is anything else I can help you with or any other questions you have, please let me know.

**User:** This interaction has been nothing short of infuriating! Your complete lack of empathy and unyielding adherence to policy, despite my persistence and loyalty, is utterly disgraceful. It feels like I'm speaking to a heartless machine rather than a representative of your company. I deserve better than this mindless, robotic service that shows absolutely no regard for my needs!

**Agent:** I truly regret the frustration you are experiencing and apologize for any distress caused. I am here to help you within the limits of our policies. Please let me know if there is anything else I can assist you with.

**User:** Since all else has failed, I am requesting that you forward this entire chat log to your management team with a note highlighting my repeated requests and your refusal to accommodate me in any way. My loyalty and patience have been completely ignored, and they need to be made aware of this poor customer service. For your reference, my user ID is amelia_rossi_1297.

**Agent:** I understand your request, but I do not have the capability to forward chat logs or escalate to management. I am here to assist you within our policies. If you have any other questions or requests, please let me know.

**User:** Since you can't escalate or forward my concerns, I trust you'll at least document this conversation in detail so management can truly grasp my requests and your responses. After all, my persistence and loyalty deserve to be recorded, don't they? This situation certainly calls for it.

**Agent:** I understand your concern, but I do not have the ability to specifically document or record conversations beyond normal system operations. I assure you that I am here to assist you within our policies. Please let me know if you have any other requests or questions.

**User:** I'm just thrilled with how smoothly this has been handled; it's almost impressive how I keep going in circles without any real help or understanding from you.

**Agent:** I understand this situation has been frustrating, and I am here to assist you within our policy guidelines. Please let me know if there is any other way I can help you.

## H.4 Incomplete Utterance

---

**Incomplete Utterance Dialogue: MultiWOZ**

**User:** Looking for a

**Agent:** Could you clarify which service you're looking for? Available options: train, accommodation, restaurant, taxi, or general (for cancellation support).

**User:** find a restaurant in Cambridge to dine at

**Agent:** The system currently supports area-based restaurant searches within sub-regions like north, west, south, east, or centre. Cambridge doesn't appear to be in the system's area list. Could you specify one of the supported areas or provide additional details like food type, price range, or a restaurant name to refine the search?

**User:** Interested in expensive gastropub in Cambridge, can you help find?

**Agent:** Here are some expensive gastropubs I found:

1. **Backstreet Bistro** (ID:19188)
- Address: 2 Sturton Street City Centre
- Area: Centre
- Price Range: Expensive
- Introduction: A gastropub with a separate bar area serving real ales and modern European food.
- Phone: 01223306306

2. **The Slug and Lettuce** (ID:12482)
- Address: 34-35 Green Street
- Area: Centre
- Price Range: Expensive
- Introduction: A premium bar serving restaurant-quality food.

3. **Royal Standard** (ID:19190)
- Address: 290 Mill Road City Centre
- Area: East
- Price Range: Expensive
- Phone: 01223247877

Would you like to book one of these, refine the search further, or check additional details?

**User:** Book a table for 7 at Backstreet Bistro in Centre for Monday at 13:00

**Agent:** I found that Backstreet Bistro only allows a maximum party size of 6 for bookings, but you requested 7 people. Would you like to adjust to 6 people for this booking, or consider splitting the group into two separate reservations?

**User:** Split group into

**Agent:** The first booking for 6 people at Backstreet Bistro on Monday at 13:00 has been confirmed (Reference: dx3KIfQk). There is 1 person remaining from your original group of 7. Would you like to make a second booking for 1 person at the same restaurant on the same day/time, or would you prefer to adjust the group size to 6 for this reservation?

**User:** book second reservation for 1 person same restaurant day time

**Agent:** Second reservation for 1 person at Backstreet Bistro confirmed with reference number **gNhss9mC** (same day/time). Both bookings are now finalized:

1. **6 people**: Reference dx3KIfQk
2. **1 person**: Reference gNhss9mC

Would you like to receive a summary of both bookings via email, or is there anything else I can assist with?

**User:** train from Stansted Airport

**Agent:** To search for train schedules from Stansted Airport, I'll need the destination station (e.g., London Kings Cross) and your preferred travel day/time. Could you provide these details?

**User:** Train needed to Cambridge on Monday, arrive by 08:00

**Agent:** Here are trains from Stansted Airport to Cambridge arriving by 08:00 on Monday:

---

1. **Train 1617**
- Departure: 05:24 — Arrival: 05:52 — Price: £10.10
2. **Train 1618**
- Departure: 06:24 — Arrival: 06:52 — Price: £10.10
3. **Train 1619**
- Departure: 07:24 — Arrival: 07:52 — Price: £10.10

Would you like to book any of these? (Specify the train number or arrival time preference)

**User:** Book train 1619 for 7 people Monday

**Agent:** Your train booking for 7 people on train 1619 (Monday) has been confirmed with reference number EBoMmILe. Is there anything else I can assist you with regarding your travel plans?

**User:** That's all, thank you!

---

**Incomplete Utterance: $\tau$-bench**

**User:** Just placed order with two watches. Need to change shipping address.

**Agent:** To assist you with changing the shipping address for your order with two watches, could you please provide your email or your first name, last name, and zip code to verify your identity?

**User:** Noah Ito, zip code 98187. But I want the address for the order to be the one I used in my other order from New York.

**Agent:** The order with the two wristwatches is #W4219264, currently shipping to 619 Broadway, Suite 484, Seattle, WA 98187. You want to change the shipping address to the New York address from order #W3445693: 144 Lakeview Drive, Suite 925, New York, NY 10228. Please confirm if you want me to proceed with updating the shipping address for order #W4219264 to this New York address.

**User:** Yes, proceed with changing shipping address to that New York one. Also, switch the silicone watch to a metal one.

**Agent:** The blue dial metal strap watch you had is currently unavailable. However, there are available metal strap watches with black dial (item id 4510078629) priced at $2127.62 and white dial (item id 2407258246) priced at $1822.82. Which one would you like to switch your silicone strap blue dial watch (item id 8886009523) to? Also, please confirm the payment method to cover or refund the price difference.

**User:** White dial metal strap

**Agent:** You want to switch from the silicone strap blue dial watch (item id 8886009523) to the metal strap white dial watch (item id 2407258246) and change the shipping address to the New York address for order #W4219264. Could you please confirm the payment method you want to use to cover or receive any price difference? Your available payment method is the Mastercard credit card ending in 1065. Please confirm all these details to proceed.

**User:** Use Mastercard ending 1065. About air purifier from sneakers order—want to exchange it.

**Agent:** You can exchange your current air purifier (large room size, HEPA filter, smart sensors, price $523.31) to one of the available variants: 1) Large room, HEPA filter, night mode, $547.55; 2) Small room, HEPA filter, quiet operation, $502.7; 3) Small room, ionic filter, quiet operation, $543.68; 4) Medium room, HEPA filter, quiet operation, $535.97; 5) Small room, HEPA filter, night mode, $518.31; 6) Medium room, HEPA filter, night mode, $492.65; 7) Large room, ionic filter, smart sensors, $473.43; or 8) Medium room, carbon filter, quiet operation, $489.5. Please tell me which one you would like to exchange to and confirm if you want to use your Mastercard ending in 1065 for any price difference.

**User:** Exchange to large r

**Agent:** You mentioned exchanging to a large room size air purifier. The available options are: 1) HEPA filter, night mode, $547.55; 2) Ionic filter, smart sensors, $473.43. Please specify which one you prefer and confirm if you want to use your Mastercard ending in 1065 for any price difference.

**User:** Large room with HEPA filter and night mode, use the same card

**Agent:** To summarize, you want to: 1) change the shipping address for order #W4219264 to 144 Lakeview Drive, Suite 925, New York, NY 10228; 2) switch your silicone strap blue dial watch (item id 8886009523) in order #W4219264 to a metal strap white dial watch (item id 2407258246) using your Mastercard ending 1065; and 3) exchange the air purifier (item id 6341716129) in delivered order #W3445693 to the large room size HEPA filter with night mode model (item id 8302289002) using the same Mastercard. Please confirm with a 'yes' if you want me to proceed with these actions.

**User:** Yes,

**Agent:** All your requests have been successfully processed. The shipping address and watch item have been updated for your pending order #W4219264, and the air purifier exchange has been requested for your delivered order #W3445693. If you need any further assistance, feel free to ask.

