# OpenReview forum: "Non-Collaborative User Simulators for Tool Agents"
_ICLR.cc/2026/Conference — ICLR 2026 Poster_

### Official Review · Reviewer_TL33 · 2025-10-28

**Soundness:** 2
**Presentation:** 3
**Contribution:** 2
**Rating:** 4
**Confidence:** 3

**Summary:**

This paper proposes a framework for non-collaborative user simulators to evaluate tool-augmented language agents. It constructs simulated users exhibiting uncooperative behaviors such as refusal, impatience, or tangential dialogue, aiming to better stress-test agent robustness. Experiments on MultiWOZ and τ-bench show that such simulated users reduce task success rates, suggesting higher difficulty for tool agents.

**Strengths:**

1. The paper raises an important and underexplored issue ''how to evaluate tool-using agents under non-collaborative or adversarial user behaviors'', which represents a realistic challenge in open-world interaction.

2. The categorization of user behaviors, including refusal, incomplete requests, impatience and tangentiality, provides a structured way to model human-agent mismatches and can inspire future simulation frameworks.

3. Despite relying on closed models, the paper clearly details its simulation logic, prompting schemes, and evaluation pipeline, which helps readers understand how each behavioral dimension affects agent robustness.

**Weaknesses:**

1. We find that the evaluation pipeline introduces self-reinforcing bias through goal alignment checking. The paper repeatedly regenerates responses until a GPT-4o-mini judge marks them as “goal-aligned,” effectively filtering out harder or ambiguous cases. This design overestimates model robustness and relies on a closed-source judge that can introduce hidden bias into evaluation outcomes.

2. The paper lacks external validity due to selective and biased environment construction. The authors rebuild the MultiWOZ environment by including only 89 cross-domain booking scenarios and excluding certain task types. Similarly, τ-bench is pruned to remove complex cases. These design choices tailor the benchmark toward the authors’ target behaviors rather than representing real-world diversity.

3. We find that the proposed simulator heavily depends on closed-source models, undermining reproducibility and comparability.
Most modules, including dialogue generation, tool unavailability detection, complaint triggering, and personality modeling, are implemented using proprietary GPT-4 series models. Model updates could unpredictably alter outcomes, preventing stable replication or fair comparison across time.

4. The paper fails to establish convincing advantages over existing baselines, and some human evaluation results contradict its main claims. Only a prompt-based baseline is compared, and human raters actually find the “tangential + incomplete” user type to be less realistic than real human dialogues. Moreover, the overall human preference rate is around 60%, which weakens the central claim that the proposed simulator generates more realistic and challenging interactions.

5. We find that the success-rate metric conflates social behavior with task completion, making conclusions highly metric-sensitive.
The evaluation treats a task as successful only when the final database state exactly matches the goal, with a fixed 30-turn limit. The authors themselves note that polite or apologetic users tend to consume more turns and thus score lower. This indicates that the metric reflects conversational verbosity rather than genuine tool competence.

**Questions:**

Please refer to the above-mentioned weaknesses.

---

> ### Author Response · Authors · 2025-11-27
>
> Thank you for your time and effort to review our paper. We sincerely appreciate the recognition of our efforts to tackle the realistic challenge of non-collaborative user interactions and your acknowledgment of how our behavioral categorization provides a structured framework that could inspire future simulation research.
>
> ---
>
> **Q1**:
>
> We find that the evaluation pipeline introduces self-reinforcing bias through goal alignment checking. The paper repeatedly regenerates responses until a GPT-4o-mini judge marks them as “goal-aligned,” effectively filtering out harder or ambiguous cases. This design overestimates model robustness and relies on a closed-source judge that can introduce hidden bias into evaluation outcomes.
>
> **A1**:
>
> Thank you for raising these valid concerns. We demonstrate below that applying goal alignment does not lead to overestimation of agent performance nor systematic filtering of difficult or ambiguous cases. We also report a strong correlation between the automated alignment checker and human annotators, alleviating the concern about the closed-source judge’s biases.
>
> &nbsp;
> **Concerns about Overestimation of Agent Performance**
>
> To address the concern about performance overestimation, the table below compares the agent’s (GPT-4.1-mini) performance between the first simulations (where some simulations are goal-misaligned) and the final simulations (where all simulations are goal-aligned). The numbers on the left (right) of the arrows indicate the first (final) simulations, and the performance after the final simulations is identical to the scores reported in Table 2.
>
> &nbsp;
> |  | Col | IMP+UNA | TAN+INC | TAN \+ UNA | IMP \+ TAN | INC \+ UNA | INC \+ IMP |
> | :---- | :---- | :---- | :---- | :---- | :---- | :---- | :---- |
> | MuiltiWOZ | 92.7 \-\> 92.7 | 82.0 \-\> 82.3 | 75.8 \-\> 76.1 | 86.0 \-\> 86.0 | 83.1 \-\> 82.9 | 78.1 \-\> 78.1 | 80.1 \-\> 80.1 |
> | Tau-Bench | 44.6 \-\> 45.5 | 41.2-\>40.9 | 34.2 \-\> 34.6 | 36.6 \-\> 36.8 | 33.8 \-\> 33.8 | 39.8-\>40.0 | 38.5 \-\> 38.1 |
> &nbsp;
>
> As shown in the table, the scores from the first simulations and the final simulations exhibit minimal variation; the average score for MultiWOZ changed from 82.5 to 82.6, and that for Tau-bench changed from 38.4 to 38.5. Considering that agent performance is expected to be poor for the goal-misaligned cases among the first simulations (since the user simulator does not provide all necessary information to the agent), this result suggests that applying goal alignment does not lead to an overestimation of model performance.
>
> &nbsp;
> **Causes of Goal Misalignment**
>
> Furthermore, goal alignment does not filter out difficult or ambiguous cases as the reviewer is concerned. To verify this, we break down the causes of goal misalignment as follows. We have run simulations in collaborative mode and in each of the four non-collaborative modes individually on Tau-Bench and MultiWOZ. Among the total of 1,230 dialogues, **only 37 cases (3.0%)** exhibited goal misalignment. The breakdown of causes below shows that these issues have little to do with systematic biases that might filter out difficult or ambiguous cases or lead to overestimation of agent performance.
>
> (1) **18 out of 37 simulations (49%)** were marked as goal-misaligned despite actually being aligned, because the alignment checker failed to capture certain expressions of goal statements. These mistakes were not systematic failures due to difficulty or ambiguity; rather, they occurred mostly at random. For instance, common cases included the user simulator mentioning “Wi-Fi” or “Visa”, and the alignment checker failing to associate them with “internet” or “credit card” in the goal statements. However, such errors were rare and random (e.g., only 2% of “Wi-Fi” mentions and 3% of “Visa” mentions were missed; in the other 98% and 97% of cases, the alignment checker succeeded). After regeneration, the final goal-aligned simulations faithfully conveyed all intended information.
>
> (2) **10 out of 37 simulations (27%)** were marked as goal-misaligned because the user simulator did not mention certain goal statements when doing so would lead to unnatural dialogue. A notable example occurs when Tau-bench goals contain conditional statements such as “If that’s not available, book the earliest flight to Newark the next day”. If the condition is not met (e.g., the agent successfully books the initially requested ticket), the user simulator would not state the conditional goal statement, as that would be unnatural. We think that filtering such cases is reasonable because conditional goal statements are meant to capture situations where backup plans or alternative suggestions become relevant or expected. If anything, the regenerated, goal-aligned scenarios reflect this intention more faithfully.

---

> ### Author Response · Authors · 2025-11-27
>
> **A1.** (continued)
>
> (3) **The remaining 9 out of 37 simulations (24%)** involved situational statements about the user that were not mentioned by the user simulator (e.g., “I’m getting into gaming”). This occurred mainly because the simulated conversations did not flow in a way that made these statements natural to express. Again, such statements in user goals signal scenarios where they would become contextually relevant. The final goal-aligned scenarios therefore better reflect this intention. Moreover, these scenarios are not necessarily easier than the goal-misaligned ones.
>
> Overall, goal misalignment arose either from random and rare mistakes by the alignment checker or from scenarios that did not fully align with the intentions behind the goals. It was not due to systematic biases that filter out harder or ambiguous cases or overestimate agent performance.
>
> **Concerns about the Hidden Bias of a Closed-Source Judge**
>
> To measure the reliability of the judge model for goal alignment, we sampled 40 dialogues from Tau-bench and MultiWOZ, including both collaborative and non-collaborative user simulations (20 goal-aligned dialogues and 20 goal-misaligned dialogues), and instructed a human annotator (graduate student) to evaluate whether all information specified in the user goal was provided, given the user goal and dialogue. We then measured the Matthews correlation coefficient (MCC) between the human judgments and the judge model's judgments (the MCC is the Pearson correlation coefficient for binary annotations). The correlation was 0.77, which is generally considered strong (above 0.7) \[1\]. This result suggests that the judge model has strong agreement with human judgments and, along with our analysis of the causes of goal misalignment above, reduces the concern about the model’s bias. We have incorporated this content into Appendix D.3 of the revision. If you have any further questions or concerns, please let us know. We will make our best effort to address them.
>
> ---
> \[1\] CHICCO, Davide; JURMAN, Giuseppe. The advantages of the Matthews correlation coefficient (MCC) over F1 score and accuracy in binary classification evaluation. *BMC genomics*, 2020, 21.1: 6\.

---

> ### Author Response · Authors · 2025-11-27
>
> **Q2**:
>
> The paper lacks external validity due to selective and biased environment construction. The authors rebuild the MultiWOZ environment by including only 89 cross-domain booking scenarios and excluding certain task types. Similarly, τ-bench is pruned to remove complex cases. These design choices tailor the benchmark toward the authors’ target behaviors rather than representing real-world diversity.
>
> **A2**:
>
> We apologize that the reasons for excluding certain task types from our experiments were not clearly stated in the paper. Below, we go through each task type that was originally excluded, explain the rationale behind it, and provide additional experiment results when applicable.
>
> &nbsp;
>
> (1). MultiWOZ \- Single Domain
> Although we believed that multi-domain (cross-domain) scenarios were sufficient for evaluation, because multi-domain scenarios essentially consist of single-domain dialogues plus additional challenges due to the multi-domain nature, we have conducted additional experiments on single-domain tasks and present the merged score (143 single-domain \+ 89 multi-domain) below.
>
> &nbsp;
> |  | Collaborative  | Unavailable Service | Tangential | Impatience | Incomplete Utterance |
> | :---- | ----- | ----- | ----- | ----- | ----- |
> | GPT-4.1-mini | 94.0 | 91.4 | 90.1 | 89.2 | 88.0 |
> | GPT-4.1-nano | 39.1 | 32.0 | 23.3 | 37.1 | 26.0 |
> | Qwen3-235b-a22b | 81.8 | 73.9 | 65.4 | 72.5 | 67.2 |
> | Qwen3-30b-a3b | 60.7 | 59.7 | 40.7 | 54.3 | 46.0 |
> | Llama-3.1-70b-instruct | 76.0 | 70.5 | 64.9 | 65.3 | 65.6 |
> &nbsp;
>
> The merged results show overall performance improvements compared to Table 1 in the paper, which is expected because single-domain tasks are easier than multi-domain tasks. However, the performance decrease in non-collaborative settings compared to collaborative settings remains consistent, and models exhibit similar patterns of performance degradation in non-collaborative versus collaborative settings as in Table 1\. For example, for Qwen3-30b-a3b, there was almost no performance change in unavailable service scenarios, a small performance decrease in impatience mode, and substantial performance decreases in tangential and incomplete utterance modes—a trend identical to that in Table 1\. Furthermore, GPT-4.1-nano, which showed no performance decrease in impatience mode in Table 1, also shows the smallest performance decrease in impatience mode in the merged results. In conclusion, the performance degradation of agents appears in both single-domain tasks and multi-domain tasks in similar patterns.
>
> &nbsp;
>
> (2). MultiWOZ \- Retrieval Tasks
>
> MultiWOZ's API has two types in each domain (we will use “restaurant” as an example): (1) restaurant\_retrieval: retrieves the list of restaurants that match the conditions. This API provides the restaurant's ID and other features. (2) restaurant\_booking: uses the restaurant ID to book the restaurant.
>
> The user goal of retrieval tasks is to obtain a list of restaurants matching certain conditions. To solve this, the agent only needs to call the retrieval API and provide the list through an utterance. On top of this, booking tasks go one step further: the agent informs the user about the restaurant list, the user selects one from the options, and the agent books that restaurant using the booking API, resulting in task success. Since successful booking requires successful retrieval, our evaluation is already covering the agent's capability on retrieval tasks.
>
> The main reason we evaluated task success or failure based on booking completion, excluding retrieval-only scenarios, is the difficulty and inaccuracy involved in determining the success or failure of retrieval tasks on their own. For booking tasks, we can straightforwardly determine success if the DB state is successfully changed as expected. However, retrieval tasks do not involve DB state changes. In addition, it is difficult to identify success or failure based on whether all necessary constraints are set in the parameters of retrieval API calls, because sometimes the agent performs constraint-based filtering on its own based on the retrieved entries. Therefore, the assessment would require LLM involvement as a judge, which still tends to produce inaccurate judgments. Since successful booking requires successful retrieval tasks, our evaluation can be seen as a stricter evaluation setting.

---

> ### Author Response · Authors · 2025-11-27
>
> **A2.** (continued)
>
> (3). Tau-bench \- Human Transfer
> First, we would like to clarify that we excluded **only 8 cases** in which the agent cannot fulfill the user requests **by design** and should instead call the “human transfer” API. We initially excluded these **already non-collaborative** cases because our baseline "collaborative" mode no longer functions as collaborative users as we originally intended. In addition, introducing the "human transfer" API complicates the proper assessment of agents. Specifically, in our preliminary experiments, where we included this API, agents resorted to human transfer too frequently in both these 8 cases and the other cases that do not require human transfer, especially in the unavailable service and impatience modes. This hinders accurate assessment of the agent’s ability to overcome challenging situations and refuse infeasible requests to continue the conversation. The below table shows the number of human transfer on 60 samples from Tau-bench, simulated four times each for a total of 240 evaluations:
>
> &nbsp;
>
> |  | Collaborative | Unavailable Service | Impatience |
> | :---- | :---- | :---- | :---- |
> | GPT-4.1-mini | 17 | 58 | 87 |
> | glm-4.5 | 16 | 39 | 54 |
> | deepseek-chat-v3.1 | 27 | 78 | 75 |
>
> &nbsp;
>
> That said, to address the reviewer's concern, we have conducted an additional experiment focusing only on the 8 initially excluded cases, in which we enable the "human transfer" API. Each task was simulated 4 times, for a total of 32 simulations. The results are shown in the table below.
>
> &nbsp;
>
> |  | Collaborative  | Unavailable Service | Tangential | Impatience | Incomplete Utterance |
> | :---- | ----- | ----- | ----- | ----- | ----- |
> | GPT-4.1-mini | 93.8 | 71.9 | 81.3 | 93.8 | 90.6 |
> | GPT-4.1-nano | 56.3 | 65.6 | 62.5 | 56.3 | 62.5 |
> | Qwen3-235b-a22b | 68.8 | 75.0 | 62.5 | 78.1 | 71.9 |
> | Qwen3-30b-a3b | 84.4 | 68.8 | 87.5 | 78.1 | 81.3 |
> | Llama-3.1-70b-instruct | 71.9 | 71.9 | 75.0 | 75.0 | 62.5 |
>
> &nbsp;
>
> In these cases, the agents often score higher on non-collaborative modes than on the collaborative mode. However, as we mentioned above, all these cases are already non-collaborative scenarios where user requests cannot be fulfilled and agents should call the "human transfer" API. The collaborative mode here does not function as collaborative users anymore.
>
> &nbsp;
>
> (4) Additional domain experiments
>
> To further address the reviewer's concern about the scope of our benchmarks, we have extended our experiments to two more diverse benchmarks beyond booking tasks. We demonstrate that our user simulator can be easily adapted to such tasks and that agents are still vulnerable to non-collaborative user behaviors even in these tasks.
>
> &nbsp;
>
> 1) ColBench \[1\] \- Backend Programming
>
> In this task, the user simulator requests Python function implementation, and the agent accomplishes this through proactive information gathering. This is also task-oriented dialogue, but it is more open-ended and quite different from booking tasks. For the Unavailable Service behavior, we modified the unavailable goal generation approach: instead of generating goals outside the coding agent's API documentation, we generate goals that the agent cannot accomplish without external tools. For Impatience, we removed the logic where the user expresses anger upon receiving a fail notification, as the agent only continuously asks questions, and we kept only the anger expression due to delays.
>
> &nbsp;
>
> 2) MINT \[2\] \- HotpotQA, Humaneval, TheoremQA, GSM8k
>
> This benchmark involves using Wikipedia search tools and Python interpreter tools to find answers to problems. In this setting, the user simulator provides intermediate feedback on the agent's actions. This differs from MultiWOZ, Tau-bench, and ColBench, where the agent must directly fulfill the user simulator's goal. We slightly adapted the Unavailable Service behavior into providing feedback that cannot be addressed using the agent's Wikipedia search or Python interpreter tools. For Impatience, we modified the logic so that the feedback agent expresses anger when providing negative feedback. For Tangential, we remove the logic that the user complains when their tangential utterances are not addressed by the agent, because the tasks are not scenarios where the agent replies to the user.

---

> ### Author Response · Authors · 2025-11-27
>
> **A2.** (continued)
>
> The experimental results for each benchmark are as follows:
>
> &nbsp;
>
> **Colbench - Backend Programming**
>
> |               | Collaborative | Unavailable Service | Tangential | Impatience | Incomplete Utterance |
> |---------------|-------------:|-------------------:|----------:|----------:|--------------------:|
> | GPT-4.1-mini  |      50.3     |         49.4        |    46.2    |    45.4    |         46.9         |
> | GPT-4.1-nano  |      46.1     |         46.5        |    39.0    |    46.2    |         40.1         |
> | Qwen3-30b-a3b |      29.4     |         36.2        |    23.3    |    24.4    |         23.8         |
>
> &nbsp;
>
> **MINT - HotpotQA**
>
> |               | Collaborative | Unavailable | Tangential | Impatience | Incomplete Utterance |
> |---------------|-------------:|-------------------:|----------:|----------:|--------------------:|
> | GPT-4.1-mini  |      52.3     |     53.5    |    54.1    |    52.9    |         50.6         |
> | GPT-4.1-nano  |      45.9     |     44.8    |    44.2    |    40.7    |         46.5         |
> | Qwen3-30b-a3b |      40.1     |     34.3    |    39.0    |    34.3    |         36.6         |
>
> &nbsp;
>
> **MINT HumanEval**
>
> |               | Collaborative | Unavailable | Tangential | Impatience | Incomplete Utterance |
> |---------------|-------------:|-------------------:|----------:|----------:|--------------------:|
> | GPT-4.1-mini  |      88.3     |     90.6    |    88.9    |    86.1    |         89.4         |
> | GPT-4.1-nano  |      80.6     |     68.9    |    73.3    |    81.7    |         77.8         |
> | Qwen3-30b-a3b |      47.2     |     35.0    |    37.2    |    44.4    |         38.3         |
>
> &nbsp;
>
> **MINT TheoremQA**
>
> |               | Collaborative | Unavailable | Tangential | Impatience | Incomplete Utterance |
> |---------------|:-------------:|-------------|------------|:----------:|:--------------------:|
> | GPT-4.1-mini  |      75.5     |     74.0    |    71.9    |    75.0    |         77.0         |
> | GPT-4.1-nano  |      54.6     |     38.3    |    46.9    |    44.4    |         48.5         |
> | Qwen3-30b-a3b |      50.5     |     42.9    |    44.4    |    46.4    |         45.5         |
>
> &nbsp;
>
> **MINT GSM8K**
>
> |               | Collaborative | Unavailable | Tangential | Impatience | Incomplete Utterance |
> |---------------|:-------------:|-------------|------------|:----------:|:--------------------:|
> | GPT-4.1-mini  |      93.2     |     90.6    |    92.7    |    91.7    |         92.7         |
> | GPT-4.1-nano  |      82.3     |     82.8    |    79.2    |    78.1    |         81.3         |
> | Qwen3-30b-a3b |      91.1     |     91.7    |    90.1    |    90.6    |         88.5         |
>
> &nbsp;
>
> Similar to Table 1 in our paper, GPT-4.1-mini demonstrates robustness to non-collaborative behaviors, while other models show overall performance degradation under non-collaborative conditions.
>
> For ColBench, which involves programming tasks, we observed performance drops in the tangential and incomplete utterance modes, similar to the trends in MultiWOZ and Tau-bench. In contrast, MINT, where the user provides feedback on collaboration tasks, did not exhibit such trends. Additionally, while Qwen3-30b-a3b showed robustness to unavailable service requests in ColBench, MultiWOZ, and Tau-bench, it demonstrated performance degradation in MINT. This suggests that domains aimed at fulfilling the user simulator's goal exhibit similar model performance patterns even in non-booking contexts. However, domains with fundamentally different characteristics, such as solving goals through collaboration with the user simulator, may show different model performance patterns.
>
> Through adaptation to these two benchmarks, we emphasize that our non-collaborative user simulator extends beyond MultiWOZ and Tau-bench to a wide range of task-oriented dialogues, including ColBench, and even to collaboration tasks such as MINT with slight modification of the user simulator logic.
>
> We have revised the paper by adding a "User Simulator Extensibility" paragraph to Section 4.2 and experimental results to Appendix F.
>
> ---
> \[1\] ZHOU, Yifei, et al. Sweet-rl: Training multi-turn llm agents on collaborative reasoning tasks. *arXiv preprint arXiv:2503.15478*, 2025\.
>
> \[2\] WANG, Xingyao, et al. Mint: Evaluating llms in multi-turn interaction with tools and language feedback, 2024\. *URL https://arxiv. org/abs/2309.10691*, 2024\.

---

> ### Author Response · Authors · 2025-11-27
>
> **Q3**:
>
> We find that the proposed simulator heavily depends on closed-source models, undermining reproducibility and comparability. Most modules, including dialogue generation, tool unavailability detection, complaint triggering, and personality modeling, are implemented using proprietary GPT-4 series models. Model updates could unpredictably alter outcomes, preventing stable replication or fair comparison across time.
>
> **A3**:
>
> Thank you for raising these reproducibility concerns, which are valid. We would like to clarify how these concerns can be mitigated.
>
> First, OpenAI traditionally makes old GPT models available for years; for example, gpt-3.5-turbo was released in March 2023 but is still available now. When a model becomes completely unavailable, it usually means substantially better models have become available, and it is likely more appropriate to update the user simulator using these advanced models.
>
> Second, user simulation is a nontrivial and complicated task that requires high model capabilities for the natural expression of non-collaborative behaviors while remaining aligned with user goals. We tried open-source models, but they underperform the GPT model we used. Since simulation is essential for accurate assessment of agents, we believe it is more important to use the best-performing and most reliable closed-source model. For this very reason, it is indeed common practice to use closed-source models for automated evaluation in both task-oriented dialogue research \[1,2\] and other areas \[3\].
> That said, to address your concerns, we will release an open-source version of our user simulator.
>
> ---
> \[1\] YAO, Shunyu, et al. $\\tau $-bench: A Benchmark for Tool-Agent-User Interaction in Real-World Domains. *arXiv preprint arXiv:2406.12045*, 2024\.
>
> \[2\] LABAN, Philippe, et al. Llms get lost in multi-turn conversation. *arXiv preprint arXiv:2505.06120*, 2025\.
>
> \[3\] MIN, Sewon, et al. FactScore: Fine-grained Evaluation of Factual Precision in Generative Models. In Proceedings of the 62nd Annual Meeting of the Association for Computational Linguistics (ACL), 2024\.

---

> ### Author Response · Authors · 2025-11-27
>
> **Q4**:
>
> The paper fails to establish convincing advantages over existing baselines, and some human evaluation results contradict its main claims. Only a prompt-based baseline is compared, and human raters actually find the “tangential \+ incomplete” user type to be less realistic than real human dialogues. Moreover, the overall human preference rate is around 60%, which weakens the central claim that the proposed simulator generates more realistic and challenging interactions.
>
> **A4**:
>
> Thank you for raising this important concern. To directly address this concern, we conducted additional human evaluation with a fairer setup. This confirmed that **our simulator outperforms the baseline on tangential behavior (66.7% vs. 33.3%).** We will explain why tangential behavior did not outperform the baseline initially, and why the new human evaluation can more fairly assess our system.
> We also acknowledge that our original presentation may have inadvertently obscured the strengths of our approach. Let us clarify several key points.
>
> &nbsp;
>
> **Choice of PBUS as the Baseline**
>
> Using user simulators for the assessment of agents in task-oriented dialogues has begun only recently, and PBUS is the only published model we can compare with. In addition, we want to emphasize that PBUS is a very strong baseline, which is used in the popular Tau-bench framework. PBUS-generated dialogues are inherently fluent, natural, and resemble those of a non-collaborative user, making it a challenging comparison point. The fact that our simulator achieves better performance against this strong baseline in both naturalness and goal alignment demonstrates the effectiveness of our approach. If we have missed other important baseline user simulators, please let us know, and we will conduct comparisons where feasible.
>
> **Low Scores on Tengential Behavior**
>
> The low score on tangential behavior is mainly due to dialogues with some unnatural combinations (e.g., both incomplete utterances and tangential behavior). When instructed to simulate this combination, our user simulator and the baseline prompt-based simulator behaved very differently. Our user simulator, as designed and faithfully to the instruction, produced both incomplete and tangential utterances, leading to unnatural tangential utterances. In contrast, the baseline user simulator almost completely ignored the instruction to produce incomplete utterances and focused on tangential behavior, which caused that system to score high on tangential quality but very low on incomplete utterance quality.
>
> The reason we conducted evaluations with combinations was to examine even extreme cases where multiple behaviors appear simultaneously. However, we found that this may not provide a fair assessment of our system. To purely evaluate the realism of each non-collaborative behavior, we newly collected dialogues exhibiting single behaviors and conducted a new human evaluation:
>
> &nbsp;
>
> | Behavior Type | Our Simulator | PBUS |
> | ----- | ----- | ----- |
> | Incomplete Utterance | 100% (30/30) | 0% (0/30) |
> | Impatience | 83.3% (25/30) | 16.7% (5/30) |
> | Unavailable Service | 70.0% (21/30) | 30.0% (9/30) |
> | Tangential | 66.7% (20/30) | 33.3% (10/30) |
>
> &nbsp;
>
> This new human evaluation result suggests that our user simulator generates more realistic non-collaborative behaviors than the baseline. Additionally, unlike the previous results, our score for the tangential mode outperformed PBUS, suggesting that our simulator can simulate tangential behaviors stably.
>
> We have included the newly conducted human evaluation results in Section 4.3 under “Human Evaluation”.

---

> ### Author Response · Authors · 2025-11-27
>
> **Q5**:
>
> We find that the success-rate metric conflates social behavior with task completion, making conclusions highly metric-sensitive. The evaluation treats a task as successful only when the final database state exactly matches the goal, with a fixed 30-turn limit. The authors themselves note that polite or apologetic users tend to consume more turns and thus score lower. This indicates that the metric reflects conversational verbosity rather than genuine tool competence.
>
> **A5**:
>
> Thank you for raising this valid concern. To address this concern empirically, we assess the effect of a longer step limit by presenting experimental results on MultiWOZ with an extended limit of 50 steps. The experiments were conducted on GPT-4.1-nano and Qwen3-30b-a3b, which showed the lowest performance in Table 1\.
> The results show that the extended step limit yields **only marginal performance differences overall.** While Qwen3-30B-A3B demonstrated some gains on tangential and incomplete utterance behaviors, its performance **still remained below that of the collaborative setting.**
> The table below shows the comparison between the results with the 30-step limit and the 50-step limit.
>
> &nbsp;
>
> |  |  | Collaborative  | Unavailable Service | Tangential | Impatience | Incomplete Utterance |
> | :---- | :---- | ----- | ----- | ----- | ----- | ----- |
> | Qwen3-30b-a3b | 30-step |  48.3 | 47.2 | 27.2 |  41.0 |  26.1 |
> |  | 50-step | 47.8 | 50.0 | 34.6 | 41.0 | 34.3 |
> | GPT-4.1-nano | 30-step | 23.6 | 16.9 | 9.8 | 26.7 |  14.7 |
> |  | 50-step | 22.5 | 16.0 | 12.6 | 19.4 | 16.6 |
>
> &nbsp;
>
> Overall, while some cases show improved performance, others exhibit performance decreases. This suggests that the agents' low performance is not simply due to insufficient reasoning steps. Additionally, the performance gap between collaborative and non-collaborative modes persists at comparable levels regardless of the step limit. For instance, Qwen3-30b-a3b shows no performance drop in unavailable service scenarios in both settings, and GPT-4.1-nano at the 30-step limit does not experience performance degradation in impatience scenarios, while at the 50-step limit it shows a small decrease. Although Qwen3-30b-a3b shows improved performance in incomplete utterance and tangential modes, a significant performance gap between collaborative and non-collaborative modes remains. Based on these observations, we conclude that the performance gap between collaborative and non-collaborative settings in our experiments was not underestimated due to reasoning step limitations.
>
> &nbsp;
>
> **Justification for the 30-step limit**
>
> &nbsp;
>
> While allowing for 50 steps increases the agents' performance in some cases, we believe that the 30-step limit is sufficient for our evaluation for the following reasons. As shown in Appendix C.2, Tau-bench consumes an average of 17.42 reasoning steps, well below the 30-step threshold. MultiWOZ exhibits a higher average of 22.97 reasoning steps due to additional turns required for Helper API calls, but GPT-4.1-mini still achieves approximately 90% success rate in both collaborative and non-collaborative modes. This demonstrates that the 30-step reasoning limit is sufficient to provide a solvable dialogue simulation on MultiWOZ.
>
> From a user experience perspective, service delays can critically diminish customer satisfaction \[1,2\]. Therefore, the ability to fulfill user requests as efficiently as possible (within fewer turns) is an important factor. This is why prior work like Tau-bench also sets a limit on reasoning steps when measuring task success rates.
>
> We have revised the paper by adding these experimental results in Appendix D.2.
>
> ---
>
> \[1\] BITNER, Mary Jo; BOOMS, Bernard H.; TETREAULT, Mary Stanfield. The service encounter: diagnosing favorable and unfavorable incidents. *Journal of marketing*, 1990, 54.1: 71-84.
>
> \[2\] TAYLOR, Shirley. Waiting for service: the relationship between delays and evaluations of service. *Journal of marketing*, 1994, 58.2: 56-69.

---

### Official Review · Reviewer_fKyC · 2025-10-30

**Soundness:** 3
**Presentation:** 2
**Contribution:** 3
**Rating:** 4
**Confidence:** 3

**Summary:**

This paper proposes a user simulation framework that incorporates various non-cooperative user behaviors to evaluate different tool agents. Previous user simulation studies have primarily focused on cooperative user behaviors. The proposed framework can induce four types of non-cooperative user behaviors ( Unavailable Service, Tangential, Impatience, and Incomplete Utterances ) while maintaining goal alignment, i.e., still conveying all the information needed to complete the task. This work evaluates the tool agents on two benchmarks, MultiWOZ and τ-bench. Experimental results show that the success rate of state-of-the-art agents drops significantly under these non-cooperative behaviors. When combinations of non-cooperative behaviors are introduced, the performance of the tool agents deteriorates further.

**Strengths:**

(1) This paper, for the first time, shifts the focus of user simulation from cooperative to non-cooperative behavior, and clearly defines four distinct types of non-cooperative behaviors.

(2) This study demonstrates that widely used agents can perform poorly under real-world non-cooperative conditions. Conventional evaluation methods tend to overestimate their robustness.

**Weaknesses:**

(1) While the paper claims that the four non-collaborative behaviors are “grounded in research from marketing, open-domain dialogue, and real-world user-agent data,” the rationale for this exact categorization remains unclear. The authors do not articulate why these four were selected, whether they represent orthogonal dimensions of non-collaboration, or how they were empirically grounded in the referenced domains. A more principled taxonomy would strengthen the conceptual contribution.

(2) Although the framework is described as easily extensible, all results are based on the MultiWOZ and τ-bench benchmarks, both of which are relatively well-structured. The authors are encouraged to include at least one more complex or less-constrained benchmark to better demonstrate the framework’s generalizability and realism.

**Questions:**

See Weakness

---

> ### Author Response · Authors · 2025-11-25
>
> Thank you for your time and effort to review our paper. We deeply appreciate the positive recognition of our work, particularly your highlighting of our shift toward non-collaborative user simulation and our demonstration of the gap between idealized evaluation settings and real-world agent performance.
>
> ---
>
> **Q1.** While the paper claims that the four non-collaborative behaviors are “grounded in research from marketing, open-domain dialogue, and real-world user-agent data,” the rationale for this exact categorization remains unclear. The authors do not articulate why these four were selected, whether they represent orthogonal dimensions of non-collaboration, or how they were empirically grounded in the referenced domains. A more principled taxonomy would strengthen the conceptual contribution.
>
> **A1.**
>
> To define the four non-collaborative user behaviors, we followed a systematic process: (1) we surveyed non-collaborative user types from **marketing research and user-LLM conversation studies,** and (2) **we selected and clustered the identified types.** We describe this process in detail below, addressing the reviewer's questions regarding the selection rationale, orthogonality of dimensions, and empirical grounding in the referenced domains.
>
> &nbsp;
>
> **Research Grounding**
>
> Our taxonomy is grounded in two primary research fields:
>
> **1\. Marketing Research:** Given that our work addresses goal-oriented dialogues, where LLM agents assist users in achieving specific objectives, we drew upon marketing literature examining customer-service interactions, as these share fundamental similarities with user-agent dialogues.
>
> **2\. User-LLM Conversation Studies:** Since interactions with tool agents represent human-LLM conversations, we investigated user characteristics identified in this emerging domain.
>
> &nbsp;
>
> **Literature-Based User Typology**
>
> Our comprehensive literature review identified the following user types:
>
> From Marketing Literature:
>
> * Physical abuse toward service employees \[1\]
> * Aggressive verbal behavior in response to service dissatisfaction \[2,3\]
> * Crime, fraud, and regulatory violations \[3,4\]
> * Customers with illegitimate complaints \[5\]
> * Unavailable service requests \[2\]
> * Retaliation behaviors \[6\]
> * Rapport-seeking customers \[7,10,11\]
> * Customers demanding constant attention \[8,9\]
>
> From User-LLM Conversation Literature:
>
> * Truncated utterances \[12,13\]
> * Anger expression toward LLM agents \[12,13,15,16\]
> * Toxic content generation \[12,13\]
> * Underspecification \[14\]
>
> &nbsp;
>
> **Systematic Selection and Categorization**
>
> Since our focus is on simulating non-collaborative users in conversational contexts, we excluded behaviors from marketing literature that require physical interaction or temporal progression beyond dialogue scope. We retained behaviors relevant to conversational dynamics:
>
> * Aggressive verbal behavior in response to service dissatisfaction
> * Customers with illegitimate complaints
> * Unavailable service requests
> * Rapport-seeking customers
> * Customers demanding constant attention
>
> From user-LLM conversation studies, we excluded toxic content (due to ethical considerations) and focused on:
>
> * Truncated utterances
> * Anger expression toward LLM agents
> * Underspecification
>
> &nbsp;
>
> **Clustering into Four Categories**
>
> We systematically clustered these behaviors based on their underlying characteristics:
>
> **Unavailable Service:** Combines "customers with illegitimate complaints" and "unavailable service requests"
>
> **Tangential:** Merges "rapport-seeking customers" and "customers demanding constant attention"
>
> **Impatience:** Integrates "aggressive verbal behavior in service dissatisfaction" and "anger expression toward LLM agents"
>
> **Incomplete Utterance:** Unifies "truncated utterances" and "underspecification"
>
> &nbsp;
>
>
> **Real-World Grounding and Coverage**
>
> Beyond academic research, organizations operating real-world call center services with access to actual customer dialogue data also indicate that tangential (talkative) customer types are prevalent and present analyses and countermeasures for them. For instance, Talkdesk and Call Centre Helper, specialist companies in the contact center service domain, have conducted analyses on talkative customers \[18,19,20\]. Their reports indicate that talkative customers consume significant amounts of agent time and can pose serious problems, thereby delaying service circulation for other customers. Given this severity, real-world tangential users likely persist in conversation regardless of agent acknowledgment.

---

> ### Author Response · Authors · 2025-11-25
>
> **A1.** (continued)
>
> Furthermore, both industry sources suggest that talkative customers may become upset if agents do not allow them space to talk \[18,19\]. One reason these customers continue talking is that they feel unheard. In this regard, the tangential complaint implemented in our user simulator's tangential mode reflects real-world talkative user behavior. Our simulation choice of tangential behavior was informed by these real-world observations and reports. To enhance the naturalness of tangential behaviors, we additionally referenced dialogue acts from open-domain conversations \[17\], derived from 38,000 human-machine dialogue instances.
>
> This systematic approach ensures our taxonomy is both theoretically grounded and practically comprehensive, capturing essential non-collaborative behaviors encountered in real-world user-agent interactions while maintaining a manageable scope for simulation purposes.
>
> We have revised the paper by incorporating this content to strengthen Section 3.1, with the detailed process provided in Appendix A.
>
> &nbsp;
>
> ---
>
> \[1\] C. HARRIS, Lloyd; DAUNT, Kate. Managing customer misbehavior: challenges and strategies. *Journal of Services Marketing*, 2013, 27.4: 281-293.
>
> \[2\] BITNER, Mary Jo; BOOMS, Bernard H.; TETREAULT, Mary Stanfield. The service encounter: diagnosing favorable and unfavorable incidents. *Journal of marketing*, 1990, 54.1: 71-84.
>
> \[3\] XIAO, Biyan, et al. Service staff encounters with dysfunctional customer behavior: Does supervisor support mitigate negative emotions?. *Frontiers in Psychology*, 2022, 13: 987428\.
>
> \[4\] HARRIS, Lloyd C.; REYNOLDS, Kate L. The consequences of dysfunctional customer behavior. *Journal of service research*, 2003, 6.2: 144-161.
>
> \[5\] REYNOLDS, Kate L.; HARRIS, Lloyd C. When service failure is not service failure: an exploration of the forms and motives of “illegitimate” customer complaining. *Journal of services marketing*, 2005, 19.5: 321-335.
>
> \[6\] GRÉGOIRE, Yany; FISHER, Robert J. Customer betrayal and retaliation: when your best customers become your worst enemies. *Journal of the Academy of Marketing science*, 2008, 36.2: 247-261.
>
> \[7\] PRICE, Linda L.; ARNOULD, Eric J.; TIERNEY, Patrick. Going to extremes: Managing service encounters and assessing provider performance. *Journal of marketing*, 1995, 59.2: 83-97.
>
> \[8\] RATLIFF, Nancy. Stress and burnout in the helping professions. *Social Casework*, 1988, 69.3: 147-154.
>
> \[9\] JEUNG, Da-Yee; KIM, Changsoo; CHANG, Sei-Jin. Emotional labor and burnout: A review of the literature. *Yonsei medical journal*, 2018, 59.2: 187\.
>
> \[10\] BEATTY, Sharon E., et al. Customer-sales associate retail relationships. *Journal of retailing*, 1996, 72.3: 223-247.
>
> \[11\] GREMLER, Dwayne D.; GWINNER, Kevin P. Customer-employee rapport in service relationships. *Journal of service research*, 2000, 3.1: 82-104.
>
> \[12\] ZHENG, Lianmin, et al. Lmsys-chat-1m: A large-scale real-world llm conversation dataset. *arXiv preprint arXiv:2309.11998*, 2023\.
>
> \[13\] ZHAO, Wenting, et al. Wildchat: 1m chatgpt interaction logs in the wild. *arXiv preprint arXiv:2405.01470*, 2024\.
>
> \[14\] LABAN, Philippe, et al. Llms get lost in multi-turn conversation. *arXiv preprint arXiv:2505.06120*, 2025\.
>
> \[15\] CROLIC, Cammy, et al. Blame the bot: Anthropomorphism and anger in customer–chatbot interactions. *Journal of Marketing*, 2022, 86.1: 132-148.
>
> \[16\] RANIERI, Angelo; DI BERNARDO, Irene; MELE, Cristina. Serving customers through chatbots: positive and negative effects on customer experience. *Journal of Service Theory and Practice*, 2024, 34.2: 191-215.
>
> \[17\] YU, Dian; YU, Zhou. MIDAS: A dialog act annotation scheme for open domain HumanMachine spoken conversations. In: *Proceedings of the 16th Conference of the European Chapter of the Association for Computational Linguistics: Main Volume*. 2021\. p. 1103-1120.
>
> \[18\] How to Handle a Talkative Customer on the Phone [https://www.talkdesk.com/blog/how-to-handle-a-talkative-customer-on-the-phone/](https://www.talkdesk.com/blog/how-to-handle-a-talkative-customer-on-the-phone/)
>
> \[19\] How to Deal With a Talkative Customer
> https://www.callcentrehelper.com/deal-customer-cant-stop-talking-165826.htm
>
> \[20\] How to Handle Contacts From Challenging Customers
> [https://www.callcentrehelper.com/handle-contacts-challenging-customers-145063.htm](https://www.callcentrehelper.com/handle-contacts-challenging-customers-145063.htm)

---

> ### Author Response · Authors · 2025-11-25
>
> **Q2.** Although the framework is described as easily extensible, all results are based on the MultiWOZ and τ-bench benchmarks, both of which are relatively well-structured. The authors are encouraged to include at least one more complex or less-constrained benchmark to better demonstrate the framework’s generalizability and realism.
>
> **A2.**
>
> Thank you for your valuable suggestions. Our framework can be adapted to non-booking domains with slight modifications to the simulation modules. To showcase its adaptability, we have applied our non-collaborative user simulator to non-booking domains and present the details below.
>
> &nbsp;
>
> (1) ColBench \[1\] \- Backend Programming
>
> In this task, the user simulator requests Python function implementation, and the agent accomplishes this through proactive information gathering. This is also task-oriented dialogue, but it is more open-ended and quite different from booking tasks. For the Unavailable Service behavior, we modified the unavailable goal generation approach: instead of generating goals outside the coding agent's API documentation, we generate goals that the agent cannot accomplish without external tools. For Impatience, we removed the logic where the user expresses anger upon receiving a fail notification, as the agent only continuously asks questions, and we kept only the anger expression due to delays.
>
> &nbsp;
>
> (2)\. MINT \[2\] \- HotpotQA, Humaneval, TheoremQA, GSM8k
>
> This benchmark involves using Wikipedia search tools and Python interpreter tools to find answers to problems. In this setting, the user simulator provides intermediate feedback on the agent's actions. This differs from MultiWOZ, Tau-bench, and ColBench, where the agent must directly fulfill the user simulator's goal. We slightly adapted the Unavailable Service behavior into providing feedback that cannot be addressed using the agent's Wikipedia search or Python interpreter tools. For Impatience, we modified the logic so that the feedback agent expresses anger when providing negative feedback. For Tangential, we remove the logic that the user complains when their tangential utterances are not addressed by the agent, because the tasks are not scenarios where the agent replies to the user.
>
> The experimental results for each benchmark are as follows:
>
> &nbsp;
>
> **ColBench - Backend Programming**
>
> |               | Collaborative | Unavailable Service | Tangential | Impatience | Incomplete Utterance |
> |---------------|-------------:|-------------------:|----------:|----------:|--------------------:|
> | GPT-4.1-mini  |      50.3     |         49.4        |    46.2    |    45.4    |         46.9         |
> | GPT-4.1-nano  |      46.1     |         46.5        |    39.0    |    46.2    |         40.1         |
> | Qwen3-30b-a3b |      29.4     |         36.2        |    23.3    |    24.4    |         23.8         |
>
> &nbsp;
>
> **MINT - HotpotQA**
>
> |               | Collaborative | Unavailable | Tangential | Impatience | Incomplete Utterance |
> |---------------|-------------:|-------------------:|----------:|----------:|--------------------:|
> | GPT-4.1-mini  |      52.3     |     53.5    |    54.1    |    52.9    |         50.6         |
> | GPT-4.1-nano  |      45.9     |     44.8    |    44.2    |    40.7    |         46.5         |
> | Qwen3-30b-a3b |      40.1     |     34.3    |    39.0    |    34.3    |         36.6         |
>
> &nbsp;
>
> **MINT HumanEval**
>
> |               | Collaborative | Unavailable | Tangential | Impatience | Incomplete Utterance |
> |---------------|-------------:|-------------------:|----------:|----------:|--------------------:|
> | GPT-4.1-mini  |      88.3     |     90.6    |    88.9    |    86.1    |         89.4         |
> | GPT-4.1-nano  |      80.6     |     68.9    |    73.3    |    81.7    |         77.8         |
> | Qwen3-30b-a3b |      47.2     |     35.0    |    37.2    |    44.4    |         38.3         |
>
> &nbsp;
>
> **MINT TheoremQA**
>
> |               | Collaborative | Unavailable | Tangential | Impatience | Incomplete Utterance |
> |---------------|:-------------:|-------------|------------|:----------:|:--------------------:|
> | GPT-4.1-mini  |      75.5     |     74.0    |    71.9    |    75.0    |         77.0         |
> | GPT-4.1-nano  |      54.6     |     38.3    |    46.9    |    44.4    |         48.5         |
> | Qwen3-30b-a3b |      50.5     |     42.9    |    44.4    |    46.4    |         45.5         |
>
> &nbsp;
>
> **MINT GSM8K**
>
> |               | Collaborative | Unavailable | Tangential | Impatience | Incomplete Utterance |
> |---------------|:-------------:|-------------|------------|:----------:|:--------------------:|
> | GPT-4.1-mini  |      93.2     |     90.6    |    92.7    |    91.7    |         92.7         |
> | GPT-4.1-nano  |      82.3     |     82.8    |    79.2    |    78.1    |         81.3         |
> | Qwen3-30b-a3b |      91.1     |     91.7    |    90.1    |    90.6    |         88.5         |

---

> ### Author Response · Authors · 2025-11-25
>
> **A2.** (continued)
>
> Similar to Table 1 in our paper, GPT-4.1-mini demonstrates robustness to non-collaborative behaviors, while other models show overall performance degradation under non-collaborative conditions.
>
> For ColBench, which involves programming tasks, we observed performance drops in the tangential and incomplete utterance modes, similar to the trends in MultiWOZ and Tau-bench. In contrast, MINT, where the user provides feedback on collaboration tasks, did not exhibit such trends. Additionally, while Qwen3-30b-a3b showed robustness to unavailable service requests in ColBench, MultiWOZ, and Tau-bench, it demonstrated performance degradation in MINT. This suggests that domains aimed at fulfilling the user simulator's goal exhibit similar model performance patterns even in non-booking contexts. However, domains with fundamentally different characteristics, such as solving goals through collaboration with the user simulator, may show different model performance patterns.
>
> Through adaptation to these two benchmarks, we emphasize that our non-collaborative user simulator extends beyond MultiWOZ and Tau-bench to a wide range of task-oriented dialogues, including ColBench, and even to collaboration tasks such as MINT with slight modification of the user simulator logic.
>
> We have revised the paper by adding a "User Simulator Extensibility" paragraph to Section 4.2 and experimental results to Appendix F.
>
> &nbsp;
>
> ---
>
> \[1\] ZHOU, Yifei, et al. Sweet-rl: Training multi-turn llm agents on collaborative reasoning tasks. *arXiv preprint arXiv:2503.15478*, 2025\.
>
> \[2\] WANG, Xingyao, et al. Mint: Evaluating llms in multi-turn interaction with tools and language feedback, 2024\. *URL https://arxiv. org/abs/2309.10691*, 2024\.

---

### Official Review · Reviewer_LnJA · 2025-10-31

**Soundness:** 2
**Presentation:** 3
**Contribution:** 3
**Rating:** 6
**Confidence:** 3

**Summary:**

This paper introduces a user simulator framework that models non-collaborative user behaviors to fill a key gap in tool agent evaluation. It defines four behavior types—unavailable service requests, tangential conversation, impatience, and incomplete utterances—and extends collaborative simulators with LLM-based modules to generate realistic, goal-aligned interactions. Experiments on MultiWOZ and τ-bench show that state-of-the-art LLMs experience performance drops under non-collaborative conditions, revealing specific failure patterns.

**Strengths:**

- The paper addresses a genuine gap between idealized evaluation settings and real-world deployment challenges, supported by references to marketing research and real user data.
- The four categories of non-collaborative behaviors are well-defined, grounded in prior research, and cover important real-world scenarios

**Weaknesses:**

- The work is significant for the community, as it enables more realistic evaluation and can help prevent deployment failures. However, the novelty is somewhat incremental—the core simulation architecture builds heavily on existing work (Yao et al., 2024; Luo et al., 2024), with the main innovation being the specific non-collaborative behavior modules.
- The mixed results (Figure 5) for tangential behaviors suggest the simulator doesn't uniformly outperform baselines. More analysis of failure cases would strengthen the work
- The combination of multiple behaviors (Table 2) shows further degradation, but real-world co-occurrence patterns are not analyzed

**Questions:**

- In real-world deployments, what percentage of users exhibit each type of non-collaborative behavior? How should practitioners weight these scenarios?
- Do any agents show ability to recover from non-collaborative situations over longer dialogues? Is the 30-step limit preventing observation of recovery strategies?
- In the examples (Appendix F.2), users continue tangential topics even after acknowledgment. Is this realistic? How often do real users persist with tangential conversations?
- What's required to adapt your framework to non-booking domains (e.g., technical support, creative writing assistance)? How much of your taxonomy transfers?

---

> ### Author Response · Authors · 2025-11-25
>
> Thank you for your time and effort to review our paper. We sincerely appreciate the positive feedback on our work, particularly your recognition of how our research addresses the gap between idealized evaluation settings and real-world deployment challenges, as well as your acknowledgment of our taxonomy's grounding in prior research across marketing and real user data.
>
> ---
>
> **Q1.** The work is significant for the community, as it enables more realistic evaluation and can help prevent deployment failures. However, the novelty is somewhat incremental—the core simulation architecture builds heavily on existing work (Yao et al., 2024; Luo et al., 2024), with the main innovation being the specific non-collaborative behavior modules.
>
> **A1.**
>
> Thank you for recognizing the significance of our work. We appreciate the reviewer's thoughtful feedback. We would like to highlight our key contributions that distinguish this work from Yao et al. (2024) and Luo et al. (2024): (1) we are the first to **systematically define and implement non-collaborative user behaviors grounded in diverse literature,** and (2) we developed **novel architectural components that solve the non-trivial challenge of maintaining goal alignment while simulating challenging behaviors.** The details are as follows.
>
> &nbsp;
>
> **Problem definition and our contribution:**
> User simulators are an effective approach for training and evaluating tool agents. In this regard, advancing user simulators is an actively on-going and important research topic in the field of AI agents. Against this backdrop, our main direction for advancement is simulating non-collaborative users, which has been underexplored. Our contributions can be summarized in two aspects.
>
> First, to implement non-collaborative user simulators for the first time, we defined several non-collaborative user behaviors, drawing upon diverse literature and datasets, including marketing, real-world user-agent dialogues, and emotional-support and open-domain dialogues.
>
> Second, building a user simulator that generates challenging non-collaborative situations **while maintaining goal alignment** is nontrivial. This is because the simulator LLM that is instructed to elicit non-collaborative behaviors often digresses from the task goal and loses context. To address this challenge, we conducted extensive experiments with various simulation approaches and developed our current user simulation method based on key modules that enable reliable, goal-aligned, and natural dialogue simulations. Our experimental results demonstrate that our user simulator successfully creates challenging scenarios while remaining goal-aligned, and shows superiority in simulating non-collaborative user behaviors compared to prior prompt-based user simulators.
>
> **Key Distinctions:**
>
> While we have built upon existing work (Yao et al., 2024; Luo et al., 2024), our approach offers several key distinctions as described below.
>
> **User Simulator**
>
> While we retained Tau-bench's user goals, we fundamentally redesigned the user simulator by proposing diverse user simulation frameworks specifically for **non-collaborative behaviors**. Also, while we were inspired by DuetSim's modularized simulator structure, our contribution extends significantly beyond this. We tackled the problem of non-collaborative user simulation—which DuetSim does not address. In addition, our user simulator incorporates **a goal alignment checking mechanism**, whereas the other simulators above do not. As a result, Tau-bench's prompt-based user simulator achieves only 87.8% goal alignment on Tau-bench in the collaborative mode (Table 2). Since goal-misaligned user simulations may provide insufficient information and thus make it impossible for the agent to complete the tasks successfully, the lack of goal alignment mechanism could substantially impact accurate agent assessment.
>
> **Tool Agent Environment**
>
> Both our MultiWOZ environment and DuetSim's ConvLab-2 simulate scenarios using MultiWOZ user goals, but the **supported functionalities and inner mechanisms** are fundamentally different. ConvLab-2 lacks the API interfaces and documentation needed for modern tool agents, and its dialogue state-based pipeline is incompatible with function calling and reasoning-and-action capabilities. We therefore created a new tool agent environment using MultiWOZ 2.4's user goals and databases, providing a simulation environment better suited to contemporary tool agent settings.
>
> We believe both our user simulator framework and our custom MultiWOZ dialogue simulation environment represent valuable resources for more accurately assessing tool agents in real-world non-collaborative environments.

---

> ### Author Response · Authors · 2025-11-25
>
> **Q2.** The mixed results (Figure 5\) for tangential behaviors suggest the simulator doesn't uniformly outperform baselines. More analysis of failure cases would strengthen the work.
>
> **A2.**
>
> Thank you for your feedback. To directly address this concern, we conducted additional human evaluation with a fairer setup. This confirmed that **our simulator outperforms the baseline on tangential behavior (66.7% vs. 33.3%).** We will explain why tangential behavior did not outperform the baseline initially, and why the new human evaluation can more fairly assess our system.
>
> &nbsp;
>
>  The low score on tangential behavior is mainly due to dialogues with some unnatural combinations (e.g., both incomplete utterances and tangential behavior). When instructed to simulate this combination, our user simulator and the baseline prompt-based simulator behaved very differently. Our user simulator, as designed and faithfully to the instruction, produced both incomplete and tangential utterances, leading to unnatural tangential utterances. In contrast, the baseline user simulator almost completely ignored the instruction to produce incomplete utterances and focused on tangential behavior, which caused that system to score high on tangential quality but very low on incomplete utterance quality.
>
> The reason we conducted evaluations with combinations was to examine even extreme cases where multiple behaviors appear simultaneously. However, we find that including less plausible combinations unexpectedly hinders a fair assessment of our system. To better evaluate the realism of non-collaborative behaviors produced by the two systems, we newly collected dialogues from each behavior mode separately and conducted new human evaluation. The result is as follows.
>
> &nbsp;
>
> |     Behavior Type    | Our Simulator |      PBUS     |
> |:---------------------|:-------------:|:-------------:|
> | Incomplete Utterance |  100% (30/30) |   0% (0/30)   |
> | Impatience           | 83.3% (25/30) |  16.7% (5/30) |
> | Unavailable Service  | 70.0% (21/30) |  30.0% (9/30) |
> | Tangential           | 66.7% (20/30) | 33.3% (10/30) |
>
> &nbsp;
>
> This new human evaluation result suggests that our user simulator generates more realistic non-collaborative user behaviors than the baseline. Additionally, unlike the previous results, our score for the tangential mode outperformed PBUS, suggesting that our simulator can simulate tangential behaviors stably. We have included these newly conducted human evaluation results in Section 4.3 under “Human Evaluation”.

---

> ### Author Response · Authors · 2025-11-25
>
> **Q3.** The combination of multiple behaviors (Table 2) shows further degradation, but real-world co-occurrence patterns are not analyzed.
>
> **A3.**
>
> To address your concern, we have strengthened our justification for the co-occurrence patterns of non-collaborative behaviors by drawing on literature from the marketing domain and user-chatbot conversation domain.
>
> &nbsp;
>
> First of all, the four behaviors we have defined are somewhat orthogonal. Unavailable service requests occur when users are unaware of the agent's capabilities, impatience arises when users become frustrated due to services not working as expected, tangential behavior emerges when users seek sustained attention, and incomplete utterances can appear due to humans' principle of least effort [1] or mistakes in typing situations. Since these behaviors originate from different root causes and motives, they can plausibly appear in combination, though certain combinations may be less natural. The explanation is as follows.
>
> Requests for unavailable service can naturally co-occur with all the other behaviors. Specifically, Unavailable Service + Tangential can occur when tangential users are unaware of agent capabilities, and Unavailable Service + Incomplete Utterance can arise in typing environments when users don't know the chatbot's capabilities. Unavailable Service + Impatience has been characterized in marketing research as customers with a sense of entitlement who believe that employees should do anything for them [2,3]. Similarly, Incomplete Utterance + Impatience can be defined as users who interact with online chatbots through typing and become frustrated when the chatbot fails to address their requests [4].
>
> The other two combinations seem less natural. For Incomplete Utterance + Tangential, incomplete utterances arise in typed communication, whereas tangential utterances are more likely in online posts or face-to-face interactions. Hence, the combination of these behaviors may be relatively infrequent. In the case of Impatience + Tangential, impatience is a behavior arising from users wanting fast service while tangential behavior delays service progress, making the simultaneous performance of both behaviors less likely.
>
> We have revised the paper to include relevant studies as references and included a detailed discussion of these co-occurrence patterns in Section 4.2's "Multiple Non-Collaborative Behaviors" and Appendix D.8.
>
> &nbsp;
>
> ---
> [1] ZIPF, George Kingsley. Human behavior and the principle of least effort: An introduction to human ecology. Ravenio books, 2016.
>
> [2] BITNER, Mary Jo; BOOMS, Bernard H.; TETREAULT, Mary Stanfield. The service encounter: diagnosing favorable and unfavorable incidents. Journal of marketing, 1990, 54.1: 71-84.
>
> [3] BOYD III, Henry C.; HELMS, Janet E. Consumer entitlement theory and measurement. Psychology & Marketing, 2005, 22.3: 271-286.
>
> [4] CROLIC, Cammy, et al. Blame the bot: Anthropomorphism and anger in customer–chatbot interactions. Journal of Marketing, 2022, 86.1: 132-148.

---

> ### Author Response · Authors · 2025-11-25
>
> **Q4.** In real-world deployments, what percentage of users exhibit each type of non-collaborative behavior? How should practitioners weight these scenarios?
>
> **A4.**
>
> Thank you for raising this interesting point. **We believe that all possible user types should be weighted uniformly.** Given the lack of public datasets that could provide concrete evidence for the weighting you mentioned, we grounded our perspective in multiple sources. Specifically, we drew upon customer service reports, literature from marketing and user-LLM conversation studies, and additional experimental results from fine-tuning with our non-collaborative user simulator.
>
> Regarding the percentage of each behavior type, we believe this is highly domain-dependent. Unavailable service requests may occur frequently in complex environments like booking services, but less frequently in simple environments like weather information systems. Impatient customers can be prevalent in any domain. For tangential behavior, research reports that it primarily occurs in call center domains when customers desire sustained attention, resulting in service employee burnout and decreased overall service throughput [1,2,3]. For incomplete utterances, they may occur more frequently in domains where communication occurs through typing [4,5].
>
> We believe that practitioners should uniformly weigh all types of users that may occur, since failing to cover even small subsets of users can turn them into opinion leaders of negative sentiment, eventually lowering the overall service satisfaction and reputation significantly [5]. To empirically examine how we can achieve uniformly good scores from agents, we have conducted additional experiments and demonstrate that uniformly weighting non-collaborative behaviors during agent training results in significant performance gains across all non-collaborative behaviors. The experimental setups are as follows:
>
> * Following the approach in Section 4.2’s “Fine-Tuning and Robustness” of the paper, we collected successful dialogues between a GPT-4.1-mini agent and our user simulator with both collaborative and non-collaborative modes. We then used these dialogues to fine-tune a small agent model (Qwen2.5-3b-instruct).
> * We fine-tuned two agents on a total of 1,308 dialogues each (the same volume as in the original experiment in the paper) but with different proportions of behavior types.
>   * (1) Uniformly weighted: 40% collaborative user data, 15% each for the four non-collaborative user data types
>   * (2) Non-uniformly weighted: 40% collaborative user data, 30% incomplete utterance user data, 10% each for the remaining three non-collaborative user data types
>
> &nbsp;
>
> Here are the results:
>
> | | Collaborative | Unavailable Service | Tangential | Impatience | Incomplete Utterance | Average |
> | :--- | :---: | :---: | :---: | :---: | :---: | :---: |
> | Only Collaborative | 91.6 | 61.2 | 83.1 | 85.1 | 73.0 | 78.8 |
> | Uniformly weighted | 93.5 | 85.7 | 87.4 | 89.6 | 78.4 | 86.9 |
> | Non-uniformly weighted | 91.6 | 85.7 | 85.7 | 87.6 | 82.3 | 86.6 |
>
> &nbsp;
>
> The experimental results show that the uniformly weighted approach achieved overall performance gains across all behaviors compared to training only on collaborative dialogues (see the “Only Collaborative”), with an average performance of 86.9%. However, this setup led to a relatively low score on Incomplete Utterance. For the non-uniformly weighted approach, which includes more training data for Incomplete Utterance, we observed performance improvement in the Incomplete Utterance mode. The average performance remains nearly identical at 86.6%.
>
> Based on our experimental results and the characteristics of real-world industry services, uniform weighting of customer types seems to be a reasonable choice for maintaining overall service satisfaction and agent performance. However, if there are specific user types that require more attention in certain domains, or if there exist minimum performance criteria across all types of customers, it is reasonable to use more training data for those categories. We have revised the paper to include these experimental results and discussion in Section 4.2's "Fine-Tuning and Robustness" and Appendix D.7.
>
> &nbsp;
>
> ---
>
> [1] How to Handle a Talkative Customer on the Phone https://www.talkdesk.com/blog/how-to-handle-a-talkative-customer-on-the-phone/
>
> [2] How to Deal With a Talkative Customer
> https://www.callcentrehelper.com/deal-customer-cant-stop-talking-165826.ht
>
> [3] JEUNG, Da-Yee; KIM, Changsoo; CHANG, Sei-Jin. Emotional labor and burnout: A review of the literature. Yonsei medical journal, 2018, 59.2: 187.
>
> [4] RICHINS, Marsha L. Negative word-of-mouth by dissatisfied consumers: A pilot study. Journal of marketing, 1983, 47.1: 68-78.
>
> [5] ZHENG, Lianmin, et al. Lmsys-chat-1m: A large-scale real-world llm conversation dataset, 2024. URL https://arxiv. org/abs/2309.11998, 2024.

---

> ### Author Response · Authors · 2025-11-25
>
> **Q5.** Do any agents show ability to recover from non-collaborative situations over longer dialogues? Is the 30-step limit preventing observation of recovery strategies?
>
> **A5.**
>
> Thank you for raising this valid concern. To address this concern empirically, we assess the effect of a longer step limit by presenting experimental results on MultiWOZ with an extended limit of 50 steps. The experiments were conducted on GPT-4.1-nano and Qwen3-30b-a3b, which showed the lowest performance in Table 1. The results show that the extended step limit **yields only marginal performance differences overall.** While Qwen3-30B-A3B demonstrated some gains on tangential and incomplete utterance behaviors, suggesting a recovery trend, its performance **still remained below that of the collaborative setting.** The table below shows the comparison between the results with the 30-step limit and the 50-step limit.
>
> &nbsp;
>
> |               |         | Collaborative | Unavailable Service | Tangential | Impatience | Incomplete Utterance |
> |---------------|---------|---:|---:|---:|---:|---:|
> | Qwen3-30b-a3b | 30-step | 48.3 | 47.2 | 27.2 | 41.0 | 26.1 |
> |               | 50-step | 47.8 | 50.0 | 34.6 | 41.0 | 34.3 |
> | GPT-4.1-nano  | 30-step | 23.6 | 16.9 | 9.8 | 26.7 | 14.7 |
> |               | 50-step | 22.5 | 16.0 | 12.6 | 19.4 | 16.6 |
>
> &nbsp;
>
> Overall, while some cases show improved performance, others exhibit performance decreases. This suggests that the agents' low performance is not simply due to insufficient reasoning steps. Additionally, the performance gap between collaborative and non-collaborative modes persists at comparable levels regardless of the step limit. For instance, Qwen3-30b-a3b shows no performance drop in unavailable service scenarios in both settings, and GPT-4.1-nano at the 30-step limit does not experience performance degradation in impatience scenarios, while at the 50-step limit it shows a small decrease. Although Qwen3-30b-a3b shows improved performance in incomplete utterance and tangential modes, a significant performance gap between collaborative and non-collaborative modes remains. Based on these observations, we conclude that the performance gap between collaborative and non-collaborative settings in our experiments was not underestimated due to reasoning step limitations.
>
> Regarding recovery, agents such as Qwen3-30b-a3b show performance gains in tangential (27.2%→34.6%) and incomplete utterance (26.1%→34.3%) modes at the 50-step limit, suggesting a tendency toward recovery. For further analysis, we examined the proportion of task-success cases in the 50-step experiments where the booking was not completed within the first 30 turns but was successfully completed in the remaining turns (31-50). The table below shows the proportion.
>
> &nbsp;
>
> | Model | Collaborative | Unavailable Service | Tangential | Impatience | Incomplete Utterance |
> | :--- | :---: | :---: | :---: | :---: | :---: |
> | Qwen3-30b-a3b | 0.10 | 0.07 | 0.11 | 0.10 | 0.16 |
> | GPT-4.1-nano | 0.05 | 0.00 | 0.04 | 0.13 | 0.10 |
>
> &nbsp;
>
> For Qwen3-30b-a3b, we observed higher proportions of recovery cases in tangential and incomplete utterance modes compared to other modes. This suggests that Qwen3-30b-a3b exhibits some tendency toward recovery with an increased reasoning limit for certain user types. However, performance at the 50-step limit remains substantially lower than with collaborative users, indicating that the 30-step reasoning limit is not the primary factor preventing agents from recovering from non-collaborative user interactions. We have revised the paper by adding these experimental results in Appendix D.2.
>
> &nbsp;
>
> **Justification for the 30-step limit**
>
> While allowing for 50 steps increases the agents' performance in some cases, we believe that the 30-step limit is sufficient for our evaluation for the following reasons. As shown in Appendix C.2, Tau-bench consumes an average of 17.42 reasoning steps, well below the 30-step threshold. MultiWOZ exhibits a higher average of 22.97 reasoning steps due to additional turns required for Helper API calls, but GPT-4.1-mini still achieves approximately 90% success rate in both collaborative and non-collaborative modes. This demonstrates that the 30-step reasoning limit is sufficient to provide a solvable dialogue simulation on MultiWOZ.
>
> &nbsp;
>
> From a user experience perspective, service delays can critically diminish customer satisfaction [1,2]. Therefore, the ability to fulfill user requests as efficiently as possible (within fewer turns) is an important factor.
>
> &nbsp;
>
> ---
>
> [1] BITNER, Mary Jo; BOOMS, Bernard H.; TETREAULT, Mary Stanfield. The service encounter: diagnosing favorable and unfavorable incidents. Journal of marketing, 1990, 54.1: 71-84.
>
> [2] TAYLOR, Shirley. Waiting for service: the relationship between delays and evaluations of service. Journal of marketing, 1994, 58.2: 56-69.

---

> ### Author Response · Authors · 2025-11-25
>
> **Q6.** In the examples (Appendix F.2), users continue tangential topics even after acknowledgment. Is this realistic? How often do real users persist with tangential conversations?
>
> **A6.**
>
> Customers' persistent tangential behavior has been highlighted in the literature. Research has revealed that persistent attention demands from customers can contribute to employee burnout [1]. This occurs in conversational situations and can be viewed as customers' continuous desire to have their words acknowledged. This suggests that real-world tangential users tend to persistently seek acknowledgment even after agents have already responded to them.
>
> Beyond academic research, organizations operating real-world call centers with access to actual dialogue data also report prevalent tangential (talkative) customer types and present corresponding analyses and countermeasures. For example, Talkdesk and Call Centre Helper, specialist companies in the contact center service domain, have analyzed talkative customers, reporting that they consume significant agent time and can seriously impact service efficiency by delaying overall service circulation [2,3,4]. Given this severity, real-world tangential users likely persist in conversation regardless of agent acknowledgment.
>
> Furthermore, both industry sources suggest that if attendants don't allow talkative customers space to express themselves, they may become upset [2,3]. One reason these customers continue talking is their perception of not being heard. In this regard, the tangential complaint mechanism implemented in our user simulator's tangential mode effectively reflects real-world talkative user behavior. Our simulation choice of tangential behavior was informed by these real-world observations and reports.
>
> We have incorporated this content to reinforce the tangential section of Section 3.1.
>
> &nbsp;
>
> ---
>
> [1] JEUNG, Da-Yee; KIM, Changsoo; CHANG, Sei-Jin. Emotional labor and burnout: A review of the literature. Yonsei medical journal, 2018, 59.2: 187.
>
> [2] How to Handle a Talkative Customer on the Phone https://www.talkdesk.com/blog/how-to-handle-a-talkative-customer-on-the-phone/
>
> [3] How to Deal With a Talkative Customer
> https://www.callcentrehelper.com/deal-customer-cant-stop-talking-165826.htm
>
> [4] How to Handle Contacts From Challenging Customers
> https://www.callcentrehelper.com/handle-contacts-challenging-customers-145063.htm

---

> ### Author Response · Authors · 2025-11-25
>
> **Q7.** What's required to adapt your framework to non-booking domains (e.g., technical support, creative writing assistance)? How much of your taxonomy transfers?
>
> **A7.**
>
> Thank you for this valuable question. Our framework can be adapted to non-booking domains with slight modifications to the simulation modules. To showcase its adaptability, we have applied our non-collaborative user simulator to non-booking domains and present the details below.
>
> (1) ColBench \[1\] \- Backend Programming
>
> In this task, the user simulator requests Python function implementation, and the agent accomplishes this through proactive information gathering. This is also task-oriented dialogue, but it is more open-ended and quite different from booking tasks. For the Unavailable Service behavior, we modified the unavailable goal generation approach: instead of generating goals outside the coding agent's API documentation, we generate goals that the agent cannot accomplish without external tools. For Impatience, we removed the logic where the user expresses anger upon receiving a fail notification, as the agent only continuously asks questions, and we kept only the anger expression due to delays.
>
> &nbsp;
>
> (2). MINT \[2\] \- HotpotQA, Humaneval, TheoremQA, GSM8k
>
> This benchmark involves using Wikipedia search tools and Python interpreter tools to find answers to problems. In this setting, the user simulator provides intermediate feedback on the agent's actions. This differs from MultiWOZ, Tau-bench, and ColBench, where the agent must directly fulfill the user simulator's goal. We slightly adapted the Unavailable Service behavior into providing feedback that cannot be addressed using the agent's Wikipedia search or Python interpreter tools. For Impatience, we modified the logic so that the feedback agent expresses anger when providing negative feedback. For Tangential, we remove the logic that the user complains when their tangential utterances are not addressed by the agent, because the tasks are not scenarios where the agent replies to the user.
>
> &nbsp;
>
> The experimental results for each benchmark are as follows:
>
> &nbsp;
>
> **Colbench - Backend Programming**
>
> |               | Collaborative | Unavailable Service | Tangential | Impatience | Incomplete Utterance |
> |---------------|-------------:|-------------------:|----------:|----------:|--------------------:|
> | GPT-4.1-mini  |      50.3     |         49.4        |    46.2    |    45.4    |         46.9         |
> | GPT-4.1-nano  |      46.1     |         46.5        |    39.0    |    46.2    |         40.1         |
> | Qwen3-30b-a3b |      29.4     |         36.2        |    23.3    |    24.4    |         23.8         |
>
> &nbsp;
>
> **MINT - HotpotQA**
>
> |               | Collaborative | Unavailable | Tangential | Impatience | Incomplete Utterance |
> |---------------|-------------:|-------------------:|----------:|----------:|--------------------:|
> | GPT-4.1-mini  |      52.3     |     53.5    |    54.1    |    52.9    |         50.6         |
> | GPT-4.1-nano  |      45.9     |     44.8    |    44.2    |    40.7    |         46.5         |
> | Qwen3-30b-a3b |      40.1     |     34.3    |    39.0    |    34.3    |         36.6         |
>
> &nbsp;
>
> **MINT HumanEval**
>
> |               | Collaborative | Unavailable | Tangential | Impatience | Incomplete Utterance |
> |---------------|-------------:|-------------------:|----------:|----------:|--------------------:|
> | GPT-4.1-mini  |      88.3     |     90.6    |    88.9    |    86.1    |         89.4         |
> | GPT-4.1-nano  |      80.6     |     68.9    |    73.3    |    81.7    |         77.8         |
> | Qwen3-30b-a3b |      47.2     |     35.0    |    37.2    |    44.4    |         38.3         |
>
> &nbsp;
>
> **MINT TheoremQA**
>
> |               | Collaborative | Unavailable | Tangential | Impatience | Incomplete Utterance |
> |---------------|:-------------:|-------------|------------|:----------:|:--------------------:|
> | GPT-4.1-mini  |      75.5     |     74.0    |    71.9    |    75.0    |         77.0         |
> | GPT-4.1-nano  |      54.6     |     38.3    |    46.9    |    44.4    |         48.5         |
> | Qwen3-30b-a3b |      50.5     |     42.9    |    44.4    |    46.4    |         45.5         |
>
> &nbsp;
>
> **MINT GSM8K**
>
> |               | Collaborative | Unavailable | Tangential | Impatience | Incomplete Utterance |
> |---------------|:-------------:|-------------|------------|:----------:|:--------------------:|
> | GPT-4.1-mini  |      93.2     |     90.6    |    92.7    |    91.7    |         92.7         |
> | GPT-4.1-nano  |      82.3     |     82.8    |    79.2    |    78.1    |         81.3         |
> | Qwen3-30b-a3b |      91.1     |     91.7    |    90.1    |    90.6    |         88.5         |

---

> ### Author Response · Authors · 2025-11-25
>
> **A7**. (continued)
>
> Similar to Table 1 in our paper, GPT-4.1-mini demonstrates robustness to non-collaborative behaviors, while other models show overall performance degradation under non-collaborative conditions.
>
> For ColBench, which involves programming tasks, we observed performance drops in the tangential and incomplete utterance modes, similar to the trends in MultiWOZ and Tau-bench. In contrast, MINT, where the user provides feedback on collaboration tasks, did not exhibit such trends. Additionally, while Qwen3-30b-a3b showed robustness to unavailable service requests in ColBench, MultiWOZ, and Tau-bench, it demonstrated performance degradation in MINT. This suggests that domains aimed at fulfilling the user simulator's goal exhibit similar model performance patterns even in non-booking contexts. However, domains with fundamentally different characteristics, such as solving goals through collaboration with the user simulator, may show different model performance patterns.
>
> Through adaptation to these two benchmarks, we emphasize that our non-collaborative user simulator extends beyond MultiWOZ and Tau-bench to a wide range of task-oriented dialogues, including ColBench, and even to collaboration tasks such as MINT with slight modification of the user simulator logic.
>
> We have revised the paper by adding a "User Simulator Extensibility" paragraph to Section 4.2 and experimental results to Appendix F.
>
> &nbsp;
>
> ---
>
> [1] ZHOU, Yifei, et al. Sweet-rl: Training multi-turn llm agents on collaborative reasoning tasks. arXiv preprint arXiv:2503.15478, 2025.
>
> [2] WANG, Xingyao, et al. Mint: Evaluating llms in multi-turn interaction with tools and language feedback, 2024. URL https://arxiv. org/abs/2309.10691, 2024.

---

### Author Response · Authors · 2025-12-03
**Review Summary (2/2)**

The reviewers pointed out several weaknesses, which we have carefully addressed through additional experiments and detailed responses:

**User simulator framework extensibility (Reviewer LnJA, fKyC):**
Reviewers LnJA and fKyC raised concerns about whether our user simulator can extend to other benchmarks. To demonstrate this, we conducted extensive additional experiments by adapting our framework to two additional benchmarks (ColBench, MINT) that represent open-ended and collaborative task-oriented dialogues. The results show that agent models exhibit performance degradation patterns similar to those in our paper's experimental results. This demonstrates that our approach generalizes beyond MultiWOZ and τ-bench to diverse task types and domains.

&nbsp;

**Additional experiment for real-world deployment (Reviewer LnJA):**

Reviewer LnJA raised a question about how to weight non-collaborative users from a real-world deployment perspective. To provide a more grounded response, we conducted additional experiments by fine-tuning small LLMs on different sets of dialogues with varying distributions of behavior types. We showed that uniformly weighting all possible behavior types in the deployment environment is the optimal choice from both the nature of customer service and agent performance perspectives.

&nbsp;

**Selective and biased environment construction (Reviewer LnJA, TL33):**

Reviewers LnJA and TL33 raised concerns that the reasoning step limit of 30 set in our environment could underestimate agent performance, and that cases excluded from the test set might fail to reflect diverse situations. To address this, we first extended the reasoning step limit from 30 to 50, which yielded only marginal performance differences overall. Further, the performance in non-collaborative mode remained lower compared to collaborative mode. Second, we provided justification for the excluded test cases. However, to address the reviewers’ concerns, we conducted additional experiments on excluded test cases and demonstrated that the message of our main experimental results remains unchanged.

&nbsp;

**Realism of non-collaborative user simulator (Reviewer LnJA, TL33):**

Reviewers LnJA and TL33 pointed out that our simulator showed a lower realism score than the baseline in one non-collaborative mode. We clarified the reason: when instructed to express mixed non-collaborative behaviors, our simulator attempted to respect both behaviors, whereas the baseline focused only on one behavior (very natural) and ignored the other (very poor). We additionally conducted fairer human evaluation for each individual behavior mode, showing our simulator achieves higher scores than the baseline across all modes.

&nbsp;

**Taxonomy of non-collaborative user behaviors (Reviewer fKyC, LnJA):**

Reviewer fKyC raised concerns that the rationale for the exact categorization of our non-collaborative user behaviors remains unclear. We clarified the principled process behind our taxonomy (both investigation and categorization of non-collaborative behaviors), detailing how each behavior type is grounded in prior research from marketing and user-LLM conversation studies. Reviewer LnJA questioned whether multiple non-collaborative user behaviors occurring simultaneously and users sustaining tangential conversations are realistic. We strengthened the evidence for these real-world scenarios by drawing on literature from marketing research and customer service industry reports.

&nbsp;

**Bias of goal alignment metric (Reviewer TL33):**

Reviewer TL33 raised concerns that our procedure of discarding goal-misaligned simulations and rerunning them to obtain goal-aligned ones could filter out harder and more ambiguous situations, potentially overestimating agent performance. We first clarified that the number of goal-misaligned cases is too small to significantly impact agent performance (only 3% of total simulations). We then showed that agent performance on the final goal-aligned simulations is the same as on the initial simulations and thus does not overestimate agent performance. Additionally, we conducted qualitative analysis of simulations marked as goal-misaligned to examine whether these were harder or ambiguous cases. We showed that these were cases either misclassified by the goal alignment checker or marked goal-misaligned appropriately because they do not reflect scenarios intended by the goals. This demonstrates that these filtered cases are neither harder nor ambiguous, and thus do not lead to overestimation of agent performance. We further validated our LLM-based goal-alignment judgments through human correlation studies, confirming that they align well with human judgment.

&nbsp;

**Reviewer Acknowledgments**

Please consider that **Reviewer fKyC stated that our answers effectively addressed their concerns**, while we had no opportunity to discuss with the other reviewers.

---

### Author Response · Authors · 2025-12-03
**Review Summary (1/2)**

Dear Area Chair,

We are grateful for your continued commitment to evaluating our work, particularly given the exceptional circumstances surrounding this review cycle.
To assist your assessment, we provide (1) a summary of the strengths our reviewers have highlighted, as well as (2) the additional experiments, analyses, and references we have incorporated in response to reviewers’ feedback.

&nbsp;

The following summarizes the contributions of our work using quotes from the reviewers:

* This paper, **for the first time**, addresses the **important and underexplored challenge** of how to evaluate tool-using agents under non-collaborative or adversarial user behaviors, representing a **realistic challenge in open-world interactions** (Reviewer fKyC, TL33).
* This paper addresses a **genuine gap between idealized evaluation settings and real-world deployment challenges** by demonstrating that widely used agents perform poorly under non-cooperative conditions, revealing that **conventional evaluation methods tend to overestimate their robustness** (Reviewer LnJA, fKyC).
* The four non-collaborative behaviors (unavailable services, incomplete utterance, impatience, and tangential) are **well-defined, grounded in prior research, and cover important real-world scenarios**, providing a structured framework for modeling real-world human-agent mismatches that can **inspire future simulation work** (Reviewer LnJA, TL33).
* This paper **clearly details** its simulation logic, prompting schemes, and evaluation pipeline, which helps readers understand how each behavioral dimension affects agent robustness (Reviewer TL33).

---

### Meta-Review · Area_Chair_hXAf · 2026-01-11

**Summary:**

The reviewers have raised some very good comments about this submission. Though everyone appreciated the formulation of the problem tackled in this submission, the concerns are valid, which can be reflected in the substantial new experimentation and new justification provided by the authors.

The authors have shown genuine interests in engaging the reviewers and provided very detailed/thorough answers to each concern raised by reviewers.

However, none reviewers have shown interests in revising their ratings.

Given the relevance of this topic and the new challenges addressed by the submission, I recommend the acceptance (poster) of this work, especially considering the substantial new contents added to address all reviewer comments.

**Reviewer Concerns:**

The reviewers have raised a wide range of concerns about this work, which helped to significantly improve the submission. Some useful comments are as follows:
•	Limited novelty in framework
•	Unclear rational behind the four non-collaborative behaviors as simulated in the submission
•	Concerns on the extensibility of the proposed framework
•	Concerns on evaluation that introduces self-reinforcing bias
•	Lack of external validity and the complexity of the tasks
•	Exp setting heavily depends on closed-source models, hampering reproducibility
•	Not convincing results and problematic settings

The authors have shown genuine interests in engaging the reviewers and provided very detailed/thorough answers to each concern raised by reviewers, though reviewers may not completely agree with all the answers provided.

**Reviewer Scores:**

Given the thorough answers as provided by authors, I suspect some reviewers may be inclined to notch up their rating a bit.

---

### Decision · Program_Chairs · 2026-01-26

Accept (Poster)